# Trading-off price for data quality to achieve fair online allocation

**Mathieu Molina**
Inria, FairPlay Team, Palaiseau, France
mathieu.molina@inria.fr

**Nicolas Gast**
Univ. Grenoble Alpes, Inria, CNRS
Grenoble INP, LIG, 38000 Grenoble, France
nicolas.gast@inria.fr

**Patrick Loiseau**
Inria, FairPlay Team, Palaiseau, France
patrick.loiseau@inria.fr

**Vianney Perchet**
CREST, ENSAE, Palaiseau, France
Criteo AI Lab, Paris, France
vianney.perchet@normalesup.org

## Abstract

We consider the problem of online allocation subject to a long-term fairness penalty. Contrary to existing works, however, we do not assume that the decision-maker observes the protected attributes—which is often unrealistic in practice. Instead they can purchase data that help estimate them from sources of different quality; and hence reduce the fairness penalty at some cost. We model this problem as a multi-armed bandit problem where each arm corresponds to the choice of a data source, coupled with the online allocation problem. We propose an algorithm that jointly solves both problems and show that it has a regret bounded by $\mathcal{O}(\sqrt{T})$. A key difficulty is that the rewards received by selecting a source are correlated by the fairness penalty, which leads to a need for randomization (despite a stochastic setting). Our algorithm takes into account contextual information available before the source selection, and can adapt to many different fairness notions. We also show that in some instances, the estimates used can be learned on the fly.

## 1 Introduction

We consider the problem of online allocation with a long-term fairness penalty: A decision maker interacts sequentially with different types of users with the global objective of maximizing some cumulative rewards (e.g., number of views, clicks, or conversions); but she also has a second objective of taking globally "fair" decisions with respect to some protected/sensitive attributes such as gender or ethnicity (the exact concepts of fairness will be discussed later). This problem is important because it models a large number of practical online allocation situations where the additional fairness constraint might be crucial. For instance, in online advertising (where the decision maker chooses to which users an ad is shown), some ads are actually positive opportunities such as job offers, and targeted advertising has been shown to be prone to discrimination Lambrecht & Tucker (2019); Speicher et al. (2018); Ali et al. (2019). Those questions also arise in many different fields such as workforce hiring (Dickerson et al., 2018), recommendation systems (Burke, 2017), or placement in refugee settlements (Ahani et al., 2021).

There has been a flourishing trend of research addressing fairness constraints in machine learning—see e.g., (Chouldechova, 2017; Hardt et al., 2016; Kleinberg & Raghavan, 2018; Barocas et al., 2019; Emelianov et al., 2020; Molina & Loiseau, 2022)—and in sequential decision-making problems—see e.g., (Joseph et al., 2016; Jabbari et al., 2017; Heidari & Krause, 2018)—as algorithmic decisions have real consequences on the lives of individuals, with unfortunate observed discrimination (Buolamwini

37th Conference on Neural Information Processing Systems (NeurIPS 2023).

& Gebru, 2018; Larson et al., 2016; Dastin, 2018). In online allocation problems, general long-term constraints have been studied for instance by Agrawal & Devanur (2015) who maximize the utility of the allocation while ensuring the feasibility of the average allocation, or by Balseiro et al. (2020) who show how to handle hard budget constraints for online allocation problems. More directly related to fairness, Nasr & Tschantz (2020) formulate the problem of fair repeated auctions with a hard constraint on the difference between the number of ads shown to each group. Finally, in recent works, Balseiro et al. (2021); Celli et al. (2022) consider the online allocation problem where a non-separable penalty related to fairness is suffered by the decision maker at the end of the decision-making process instead of a hard constraint—these works are the closest to ours.

All the aforementioned papers, unfortunately, assume that the protected attributes (which define the fairness constraints) are observed before taking decisions. In practice, it is often not the case—for instance to respect users' privacy (Guardian, 2023)—and this makes it challenging to satisfy fairness constraints (Lipton et al., 2018). In online advertising for example, the decision-maker typically has access to some public "contexts" on each user, from which she could try to infer the value of the attribute; but it was shown that the amount of noise can be prohibitive and therefore ensuring that a campaign reaches a non-discriminatory audience is non-trivial Gelauff et al. (2020).

**Our contribution.** In this paper, we consider the online allocation problem under long-term fairness penalty, in the practical case where the protected attributes are not observed. Instead, we consider the case where the decision-maker can pay to acquire more precise information on the attributes (beyond the public context), either by directly compensating the user (the more precise the information on the attribute, the higher the price) or by buying additional data to some third parties data-broker.[1] Using this extra information, she should be able to estimate more precisely, and thus sequentially reduces, the unfairness of her decisions while keeping a high cumulative net reward. The main question we aim at answering is *how should the decision maker decide when, and from which source (or at what level of precision), to buy additional data in order to make fair optimal allocations?*

Compared to the closest existing works (Balseiro et al., 2021; Celli et al., 2022) which study online allocation with a long term fairness penalty, the main novelty in the setting we examine is two-fold: we allow for uncertainty on the attributes of each individual, and more importantly we consider *jointly* the fair online allocation problem with a source selection problem. Consequently, we present the efficient Algorithm 1 that tackles both of these challenges concurrently. This algorithm combines a dual gradient descent for the fair allocation aspect and a bandit algorithm for the source selection part. The final performance of an algorithm is its net cumulative utility (rewards minus costs of buying extra information) penalized by its long-term unfairness; that is quantified by the "regret": the difference between this performance and the one of some benchmarking "optimal" algorithm. We show that Algorithm 1 has a sub-linear regret bound under some stochastic assumptions. Notably, the performance achieved by Algorithm 1 using randomized source selection is strictly better than when using a single fixed source, because of the interaction through the fairness penalty—a key difference with standard bandit settings. On a more technical level we show how one can model the randomness and estimates for the protected attributes, how to bound the fair dual parameters which is crucial in order to use adversarial bandit techniques, and how to combine the analysis of the primal and dual steps of the algorithm.

There are many different definitions of group fairness that can be studied (e.g., demographic parity, equal opportunity, etc.). Instead of focusing on a specific one, we consider a generic formulation that can be instantiated to handle most of those different concepts (see Section 2.1 and Appendix A). We also discuss in Section 4.1 how to adapt our algorithm to different fairness penalties. This gives a higher level of generality to our results compared to existing approaches that can handle fewer fairness criteria (e.g., Celli et al. (2022)).

For the sake of clarity, we expose our key results in a simple setting. In particular, we assume binary decisions and a linear utility, and we assume that the expected utility conditional on the context is known. All these assumptions can be relaxed. In particular, we can learn the utilities (see Section 4.2). We can handle also more general decision variables and utility forms. This allows in particular to tackle problems of matching and auctions (albeit with some restrictions), we discuss that in Appendices B and C.

---

[1]This setting includes as special cases the extremes where no additional information can be bought and where the full information is available.

**Related work.** The problem of online fair allocation is closely related to online optimization problems with constraints, which is studied in a few papers. For instance, bandit problems with knapsack constraints where the algorithm stops once the budget has been depleted have been studied by Badanidiyuru et al. (2018); Agrawal & Devanur (2016). Li & Ye (2021); Li et al. (2022) consider online linear programs, with stochastic and adversarial inputs. Liu et al. (2021) deal with linear bandits and general anytime constraints, which can be instantiated as fairness constraints. More recently Castiglioni et al. (2022a,b) propose online algorithms for long-term constraints in the stochastic and adversarial cases with bandit or full information feedback. Some papers take into account soft long-term constraints (Agrawal & Devanur, 2015; Jenatton et al., 2016), and more recently in (Balseiro et al., 2021; Celli et al., 2022) where the long-term constraint can be instantiated as a fairness penalty—we also adopt a soft constraint. We depart from this literature by considering the case where the protected attributes (based on which fairness is defined) is not observed. We consider the case where the decision-maker can buy additional information to estimate it (which adds considerable technical complexity), but even in the case where no additional information can be bought our work extends that of (Balseiro et al., 2021; Celli et al., 2022).

As mentioned above, the fairness literature usually assumes perfect observation of the protected attributes yet noisy realizations of protected attributes or limited access to them to measure and compute fair machine learning models has also been considered (Lamy et al., 2019; Celis et al., 2021; Zhao et al., 2022). In some cases, the noise may come from privacy requirements, and the interaction between those two notions has been studied (Jagielski et al., 2019; Chang & Shokri, 2021), see (Fioretto et al., 2022) for a survey. There are also works on data acquisition, which is similar to purchasing information from different sources of information; e.g., Chen et al. (2018); Chen & Zheng (2019) study mechanisms to acquire data so as to estimate some of the population's statistics. They use a mechanism design approach where the cost of data is unknown and do not consider fairness (or protected attributes). However, none of these approaches can handle sequential decision problems with fairness constraints or penalties (and choosing information sources).

## 2 Preliminaries

### 2.1 Model and assumptions

We present here a simpler model, and later-on discuss possible extensions. Consider a decision maker making sequential allocation decisions for a known number of $T$ users (or simply stages) in order to maximize her cumulative rewards. The user $t$ has some protected attributes $a_t \in \mathcal{A} \subset \mathbb{R}^d$, that is not observed before taking a decision $x_t$. On the other hand, the decision-maker first observes some public context $z_t \in \mathcal{Z}$, where $\mathcal{Z}$ is the finite set of all possible public contexts and she has the possibility to buy additional information. There are $K$ different sources for additional information and choosing the source $k_t \in [K]$ has a cost of $p_{k_t}$, but it provides a new piece of information, which together with the public information is summarized in the random variable $c_{tk_t} = (z_t, \text{data from } k_t)$. Based on this, the decision $x_t \in \{0, 1\}$ can be made; this corresponds to include, or not, user $t$ in the cohort (for instance, to display an ad or not). Including user $t$ generates some reward/utility $u_t$, which might be unknown to the decision maker (as it may depend on the private attribute), but can be estimated using the different contexts.

To fix ideas, we show how this model applies to two examples: Imagine an advertiser aiming to display ads to an user, able to see some bare-bone information through cookies, such as which website was previously visited ($z_t$). Based on this information, they can decide whether or not to buy additional information ($c_{tk}$) from different data brokers that collect user activity. For example, if they observe that the user has browsed clothing stores, they might opt to acquire data containing purchase details from this website. This enables them to estimate the user's gender ($a_t$) based on the type of clothing bought. Now consider the problem of fairly relocating refugees to different cities. When the organization in charge of resettlement receives a resettlement case ($z_t$), it can either decide to directly assign the refugee to a specific city, or to conduct an additional costly investigation (which might involve a third party watch-dog) to get more information ($c_{tk}$) on some protected attributes of interest such as wealth or age ($a_t$), which might have been intentionally misreported.

We assume that the global objective is to maximize the sum of three terms: the cumulative rewards of all selected users, minus the costs of the additional information bought, minus some unfairness penalty, represented by some function $R(.)$. Denoting by $\boldsymbol{k} = (k_1, \cdots, k_T)$ and $\boldsymbol{x} = (x_1, \cdots, x_T)$

the sources and allocations selected during the $T$ rounds, the total utility of the decision maker is then

$$\mathcal{U}(\boldsymbol{k}, \boldsymbol{x}) = \sum_{t=1}^{T} u_t x_t - \sum_{t=1}^{T} p_{k_t} - TR\Big(\frac{1}{T}\sum_{t=1}^{T} a_t x_t\Big). \tag{1}$$

The penalty function $R(.)$ is a convex penalty function that measures the fairness cost of the decision-making process, due to the unbalancedness of the selected users at the end of the $T$ rounds. It can be used to represent statistical parity (Kamishima et al., 2011), as a measure of how far the allocation is from this fairness notion. This fairness penalty $R(.)$ is also used in Balseiro et al. (2021); Celli et al. (2022). In fact, the objective (1) is equal the one used in these papers minus the cost of additional information bought, $\sum_t p_{k_t}$.

**Knowns, unknowns and stochasticity**  We assume that users are independent and identically distributed (i.i.d.), in the sense that the whole vectors $(z_t, u_t, a_t, c_{t1} \ldots c_{tK})$ are i.i.d., drawn from some underlying unknown probability distribution. While this may be a strong assumption, some applications such as online advertising correspond to large $T$ but to a short real time-frame, hence incurring very little variation in the underlying distribution. The prices $p_k$ and the penalty function $R(.)$ are known beforehand. As mentioned several times, the only feedback received is $c_{tk}$, after selecting source $k$, and this should be enough to estimate $u_t$ and $a_t$. We therefore assume that the conditional expectations $\mathbb{E}[u_t \mid c_{tk_t}]$ and $\mathbb{E}[a_t \mid c_{tk_t}]$ are known. The rationale behind this assumption is that these conditional expectations have been learned from past data.

**Penalty examples and generalizations**  A typical example for $a_t$ is the case of one-hot encoding: there are $d$ protected categories of users and $a_t$ indicates the category of user $t$. For simplicity, assume that $d = 2$, then the quantity $\sum_t a_{ti} x_t$ is the number of users of type $i \in \{1, 2\}$ that have been selected. The choice of $TR(\sum_{t=1}^{T} a_t x_t / T) = |\sum_t a_{t1} x_t - \sum_t a_{t2} x_t|$ amounts to penalizing the decision maker proportional to the absolute difference of users in both groups. This generic setting can also model other notions of fairness, such as Equality of Opportunity (Hardt et al., 2016), by choosing other values for $a_t$ and $R(.)$, see examples and discussion in Appendix A.

Similarly, the choice of $x_t \in \{0, 1\}$ can be immediately generalized to any finite decision set or even continuous compact one (say, the reward at stage $t$ would then be $\mathbf{u}_t^\top x_t$ for some vector $\mathbf{u}_t$), which makes it possible to handle problems such as bipartite matching. Instead of deriving a linear utility from selecting an user, general bounded upper semi-continuous (u.s.c.) utility functions can also be treated, and can be used to instantiate auctions mechanism (with some limitations detailed in Appendix C). We also explain how to relax the assumption that $\mathbb{E}[u_t \mid c_{tk_t}]$ is known in Section 4.2, by deriving an algorithm that actually learns it in an online fashion, following linear contextual bandit techniques (Abbasi-yadkori et al., 2011).

We show in Section 3.5 that the assumption that $\mathcal{Z}$ is finite can be relaxed if all conditional distributions depend smoothly on $z$, following techniques from Perchet & Rigollet (2013).

**Mathematical assumptions**  We shall assume $|u_t| \leq \bar{u}$, for all $t$, and that $\|a\|_2 \leq 1$ for all $a \in \mathcal{A}$. We make, for now, no structural assumption on the variables $c_{tk}$. We mention here that the decision maker has to choose a single source at each stage, but this is obviously without loss of generality (by adding void or combination of sources).

We define $\Delta = \mathrm{Conv}(\mathcal{A} \cup \{0\})$, where $\mathrm{Conv}$ is the closed convex hull of a set. Since $\mathcal{A}$ is compact, the set $\Delta$ is also convex and compact. The penalty function $R : \Delta \to \mathbb{R}$ is a proper closed convex function that is $L$-Lipschitz continuous for the Euclidean norm $\|\cdot\|_2$. While the convexity assumption is pretty usual, the Lipschitzness assumption is rather mild as $\Delta$ is convex and compact. Nevertheless, interesting non-Lipschitz functions, such as the Kullback-Leibler divergence, can be modified to respect these assumptions (see Celli et al. (2022)).

## 2.2  Benchmark and regret

A usual measure of performance for online algorithms is the regret that compares the utility obtained by an online allocation to the one obtained by an oracle that knows all parameters of the problem, yet not the realized sequence of private attributes. We denote the performance of this oracle by OPT:

$$\mathrm{OPT} = \max_{\boldsymbol{h} \in ([K]^{\mathcal{Z}})^T} \mathbb{E}\left[ \max_{\boldsymbol{x} \in \{0,1\}^T} \mathbb{E}\left[\mathcal{U}(\boldsymbol{k}, \boldsymbol{x}) \mid c_{1k_1}, \ldots, c_{Tk_T}\right] \right], \tag{2}$$

where the conditional expectation indicates that the oracle first chooses a contextual policy $h_t$ for all users that specifies which source $k_t = h_t(z_t)$ to select as a function of the variables $z_t$. It then observes all contexts $c_{tk_t}$ and makes for all $t$ the decisions $x_t$ based on that.

Denoting ALG the expected penalized utility of an online algorithm, its expected regret is:

$$\mathrm{Reg} = \mathrm{OPT} - \mathbb{E}[\mathcal{U}(\boldsymbol{k}, \boldsymbol{x})] = \mathrm{OPT} - \mathrm{ALG}\,. \tag{3}$$

We remark that the benchmark of Equation (2) allows choosing different sources of information for different users with the same public information $z_t$. As such, it differs from classical benchmarks in the contextual bandit literature that compare the performance of an algorithm to the best static choice of arm per context and whose performance would be

$$\mathrm{static-OPT} := \max_{h \in [K]^{\mathcal{Z}}} \mathbb{E}[\max_{\boldsymbol{x} \in \{0,1\}^T} \mathbb{E}[\mathcal{U}(\boldsymbol{k}, \boldsymbol{x}) | c_{1k_1}, \dots, c_{Tk_T}]], \tag{4}$$

where the $h \in [K]^{\mathcal{Z}}$ policy that maps the public information to a source selection is the same for all users.

The benchmark (4) is the typical benchmark in contextual multi-armed bandits, as the global impact of decisions at different epochs and for different contexts are independent. However, this is no longer the case with the unfairness penalty $R(.)$ that requires coupling all decisions:

**Proposition 2.1.** *There exist an instance of the problem and a constant $b > 0$ such that for all $T$:*

$$\mathrm{static-OPT} + bT < \mathrm{OPT}\,.$$

This result shows that an algorithm that only tries to identify the best source will have a linear regret compared to OPT. This indicates that the problem of source selection and fairness are strongly coupled, even without public information available, and cannot be solved through some sort of two-phase algorithm where each problem is solved separately. The proof of this result is presented in Appendix F.1. In this paper, our primary emphasis lies in the examination of the performance disparity between an online algorithm and the offline optimum. Nevertheless, we provide supplementary experiments in Appendix F.3 that investigate how variations in the prices $p_k$ and the penalty $R$ impact the solution of the offline optimum.

## 3 Algorithm and Regret Bounds

In this part, we present our online allocation algorithm and its regret bound. For clarity of exposition, we first present the algorithm in the case $|\mathcal{Z}| = 1$, i.e., without public information available (and thus we remove $z_t$ from the algorithm). The extension to $|\mathcal{Z}| > 1$ uses similar arguments and is discussed in Section 3.5.

### 3.1 Overview of the algorithm

Algorithm 1 devised to solve this problem is composed of two parts: a bandit algorithm for the source selection, and a gradient descent to adjust the penalty regularization term. This requires a dual parameter $\lambda_t \in \mathbb{R}^d$ that is used to perform the source selection as well as the allocation decision $x_t$.

The intuition is the following: the performance of each source is evaluated through some "dual value" for a given dual parameter $\lambda$. The optimal primal performance is equal to the dual value when it is minimized in $\lambda$, because $R$ is convex and randomized combinations of sources is allowed thus there is no duality gap. Hence the dual value of each source is iteratively evaluated in order to select the best source, and simultaneously minimize the dual value of the selected source through $\lambda$, so that the source selected is indeed optimal.

**Bandit part.** For the source selection, we use the EXP3 algorithm (see Chapter 11 of Lattimore & Szepesvári (2020)) on a virtual reward that depends on the dual parameter. Given a dual parameter $\lambda \in \mathbb{R}^d$ and a context $c_{tk}$, we define the virtual reward as

$$\varphi(\lambda_t, c_{tk}, k) = \max\left(\mathbb{E}[u_t \mid c_{tk}] - \langle \lambda_t, \mathbb{E}[a_t \mid c_{tk}]\rangle, 0\right) - p_k, \tag{5}$$

where $\langle \lambda_t, \mathbb{E}[a_t \mid c_{tk}]\rangle$ denotes the scalar product between $\lambda_t$ and $\mathbb{E}[a_t \mid c_{tk}]$. To compute this expectation, one needs to know the quantities $p_k$, $\mathbb{E}[u_t|c_{tk}]$ and $\mathbb{E}[a_t|c_{tk}]$.

To apply EXP3, a key property is to ensure that the virtual rewards are bounded, which requires $\lambda_t$ to remain bounded. As we show in Lemma 3.2, this is actually guaranteed by the design of the gradient descent on $\lambda$. This lemma implies that there exists $m \in \mathbb{R}^+$ such that $|\varphi(\lambda_t, c_{tk})| \leq m$ for all $t$ and $k$. Let us denote by $\pi_{tk}$ the probability that source $k$ is chosen at time $t$ and $k_t \sim \pi_t$. We define the importance-weighted unbiased estimator vector $\hat{\varphi}(\lambda, c_t) \in \mathbb{R}^K$ where each coordinate $k' \in [K]$ is:

$$\hat{\varphi}(\lambda, c_{tk}, k, k') = m - \mathbb{1}[k' = k]\frac{m - \varphi(\lambda, c_{tk}, k)}{\pi_{tk}}.$$

Using this unbiased estimator, we can apply the EXP3 algorithm to this virtual reward function.

**Gradient descent part.** Once the source $k_t$ for user $t$ is chosen and $c_{tk_t}$ is observed, we can compute the decision $x_t$ (see (6)). This $x_t$ is then used in (8)-(9) to perform a dual descent step on the multiplier $\lambda_t$. Although using a dual descent step is classical, our implementation is different because we need to guarantee that the values of $\lambda_t$ remain bounded for EXP3. To do so, we modify the geometry of the convex optimization sub-problem by considering a set of allocation targets larger than the original $\Delta$. For $\delta \in \Delta$, we define the set $\Delta_\delta$ as the ball of center $\delta$ and radius $\mathrm{Diam}(\Delta)$. This ball contains $\Delta$: $\Delta \subset \Delta_\delta$.

Algorithm 1 uses any extension of $R$ to $\bar{\Delta} = \cup_{\delta \in \Delta} \Delta_\delta$ that is convex and Lipschitz-continuous, for instance, the following one (see Lemma D.2):

$$\bar{R}(\delta) = \inf_{\delta' \in \Delta} \{R(\delta') + L\|\delta - \delta'\|_2\},$$

that has the same Lipschitz-constant $L$ as $R$ (which is the best Lipschitz-constant possible).

## 3.2 Algorithm and implementation

Combining these different ideas leads to Algorithm 1. This algorithm maintains a dual parameter $\lambda_t$ that encodes the history of unfairness that ensures that the fairness penalty $R(.)$ is taken into account. This dual parameter $\lambda_t$ is used in (6) to compute the allocation $x_t$ and in (7) to choose the source of information. The dual update (8)-(9) guarantees that we take $R(.)$ into account.

---

**Algorithm 1** Online Fair Allocation with Source Selection

---

**Input:** Initial dual parameter $\lambda_0$, step sizes $\eta$ and $\rho$, cumulative estimated rewards $S_0 = 0 \in \mathbb{R}^K$.
**for** $t \in [T]$ **do**

    Draw a source $k_t \sim \pi_t$ where $\pi_{tk} = \exp(\rho S_{t-1,k})/\sum_{l=1}^K \exp(\rho S_{t-1,l})$, and observe $c_{tk_t}$.
    Compute the allocation for user $t$:

$$x_t = \begin{cases} 1 & \text{if } \mathbb{E}[u_t \mid c_{tk_t}] \geq \langle \lambda_t, \mathbb{E}[a_t \mid c_{tk_t}] \rangle \\ 0 & \text{otherwise.} \end{cases} \tag{6}$$

    Update the estimated rewards sum and sources distributions for all $k \in [K]$:

$$S_{tk} = S_{(t-1)k} + \hat{\varphi}(\lambda_t, c_{tk}, k_t, k), \tag{7}$$

    Let $\delta_t = x_t \mathbb{E}[a_t \mid c_{tk_t}]$. Compute the dual fairness allocation target and update the dual parameter

$$\gamma_t = \arg\max_{\gamma \in \Delta_{\delta_t}}\{\langle \lambda_t, \gamma \rangle - \bar{R}(\gamma)\}, \tag{8}$$

$$\lambda_{t+1} = \lambda_t - \eta(\gamma_t - \delta_t). \tag{9}$$

**end for**

---

An interesting property of the dual gradient descent is that it manages to provide a good fair allocation while updating the source selection parameters simultaneously. Indeed if $\lambda$ were fixed, then we could solve the source selection part through $\varphi$ and a bandit algorithm. However, both the $\lambda_t$ and the $\pi_t$ change over time, which may hint at the necessity of a two-phased algorithm as the combination of these two problems generates non-stationarity for both the dual update and also for the bandit problem. The dual gradient descent manages to combine both updates in a single-phased algorithm.

The different assumptions imply that the decision maker has access to the $\mathbb{E}[a_t|c_{tk}]$ and to $\mathbb{E}[u_t \mid c_{tk}]$ for any possible context value $c_{tk}$. Such values could be indeed estimated from offline data. The knowledge of such values is sufficient to compute the allocation $x_t$ in (6), the virtual value estimation $\hat{\varphi}$ of (7), or to compute $\delta_t$. Once these values are computed, the only difficulty is to solve (8). In some cases, it might be solved analytically. Otherwise, it can also be solved numerically as it is an (a priori low-dimensional) convex optimization problem. Overall this is an efficient online algorithm which only uses the current algorithm parameters $\lambda_t, \pi_t$, and current context $c_{tk_t}$.

### 3.3 Regret bound

We emphasize that Algorithm 1 uses randomization among the different sources of information and does not aim to identify the best source. As shown in Proposition 2.1, this is important because using the best source of information can be strictly less good that using randomization. Moreover, it simplifies the analysis because it convexifies the set of strategies. This means that Sion's minimax theorem can be applied to some dual function, which allows for $\lambda_t$ and $\pi_t$ to be updated simultaneously. If one would try to target the best static source of information (static-OPT), one would need to determine the optimal dual parameter $\lambda_k$ of each source. This would lead to an algorithm that is both more complicated and less efficient (because of Proposition 2.1).

The following theorem shows that Algorithm 1 has a sub-linear regret of order $O(\sqrt{T})$. This regret bound is comparable to those in Balseiro et al. (2021); Celli et al. (2022) but we handle the much more challenging case of having multiple sources of information, and imperfect information about $a_t$ and $u_t$.

**Theorem 3.1.** *Assume that Algorithm 1 is run with the parameters* $\eta = L/(2\operatorname{Diam}(\Delta)\sqrt{T})$, $m = \bar{u} + L + \max_k |p_k| + 2\eta \operatorname{Diam}(\Delta)$, $\rho = \sqrt{\log(K)/(TKm^2)}$, *and that* $\lambda_0 \in \partial R(0)$, *the subgradient of $R$ at 0. Then the expected regret of the algorithm is upper bounded by:*

$$\operatorname{Reg} \leq 2((L + \bar{u} + \max_k |p_k|)\sqrt{K\log(K)} + L\sqrt{d} + L\operatorname{Diam}(\Delta))\sqrt{T} + 2L\sqrt{K\log(K)}.$$

Note that this regret bound is tight: when $R = 0$ the problem we consider reduces to a $K$-armed bandit with bounded rewards $\mathbb{E}[\max(\mathbb{E}[u_t \mid c_{t,k}], 0) - p_k]$ for arm $k \in [K]$, which has a $\Omega(\sqrt{T})$ regret lower bound (Lattimore & Szepesvári, 2020). It is of the same order in $T$ as our regret upper bound. Remark that $R = 0$ implies that $L = 0$ and we do recover the regret bound for the EXP3 algorithm. Similarly if $K = 1$ the bandit regret contribution disappears. In our analysis, the regret due to the interaction between the bandit and the online fair allocation is $L\sqrt{K\log(K)T}$.

The time-horizon dependent parameters used in Algorithm 1 can be adapted to obtain an anytime algorithm. While using a doubling trick directly for $\eta$ is not possible as some protected attribute would already be selected when restarting the algorithm, we can use an adaptive learning rate of $\eta_t = \mathcal{O}(1/\sqrt{t})$. Indeed due to the boundedness of the $(\lambda_t)_{t \in T}$ (Lemma 3.2), we can act as if we had a finite diameter for the space of the $\lambda_t$. This results in a slight increase of regret, the constant term in Theorem 3.1 now scaling in $\mathcal{O}(\sqrt{T})$.

### 3.4 Sketch of proof

As mentioned above, when $K = 1$ (resp. $R = 0$) the problem reduces to fair online allocation as in Balseiro et al. (2021); Celli et al. (2022) (resp. to multi-armed bandits). The main technical difficulty thus lie in combining algorithms used in these problems. Ideally, we would have access to some optimal dual parameter $\lambda_k^*$ *before* we run the algorithm so that we can simply run a bandit algorithm, which is obviously not possible as the selected source $k_t$ affects the fairness, and the selected parameter $\lambda_t$ affects the arms virtual rewards. In particular it is not clear how the $\lambda_t$ evolve in the worst case while the algorithm is running. Instead we alternate between those primal and dual updates. We thus need to show that this alternation indeed achieves good performance.

We present two important lemmas used in the proof of Theorem 3.1. The first one guarantees that doing a gradient descent with $\bar{R}$ implies that the dual values $\lambda_t$ remain bounded. This is crucial for EXP3 as it implies that the virtual values $\varphi$ remain bounded along the path of the $\lambda_t$.

**Lemma 3.2.** *Let $\lambda_0 \in \mathbb{R}^d$, $\eta > 0$, and an arbitrary sequences $(\delta_1, \delta_2, \dots) \in \Delta^{\mathbb{N}}$. Assume that $\lambda_0 \in \partial R(0)$ and define recursively $\lambda_t$ by Equations (8) and (9). Then for all $t$, we have $\|\lambda_t\|_2 \leq L + 2\eta \operatorname{Diam}(\Delta)$.*

*Sketch of Proof.* The main idea is to show that the distance of $\lambda_t$ to the reunion of subgradients of $\bar{R}$ (which is bounded by Lipschitzness property) is a decreasing function in $t$. We can show this by using the $KKT$ conditions of the optimization problem. The convex set over which we optimize is a simple Euclidean ball, because of our modification, centered around the appropriate point hence allowing us to redirect the gradient $\lambda_{t+1} - \lambda_t$ towards this set. Moreover, we add some "security" around this set of the size of the gradient bound to make sure that the $\lambda_t$ remains in this set. The full proof can be found in Appendix D.3. □

The second lemma guarantees that having access to the conditional expectation $\mathbb{E}[a_t|c_{tk}]$ is enough to derive a good algorithm when considering the conditional expectation of the total utility. This way we avoid the computation of the conditional expectation of $R$, which would be more difficult.

**Lemma 3.3.** *For $(x_1, \ldots, x_T)$ and $(k_1, \ldots, k_T)$ generated according to Algorithm 1, with $\delta_t = x_t\mathbb{E}[a_t \mid c_{tk_t}]$, we have the following upper bound:*

$$\left| \mathbb{E}[R(\frac{1}{T}\sum_{t=1}^{T} a_t x_t) - R(\frac{1}{T}\sum_{t=1}^{T}\delta_t)] \right| \leq 2L\sqrt{\frac{d}{T}}.$$

*Sketch of proof.* We use the Lipschitz property of $R$ and some inequalities to directly compare the difference of the sums. The variable $x_t$ depends on the past history and needs to be carefully taken into account, through proper conditioning. Finally, we compute the variance of a sum of martingale differences. See proof in Appendix D.4. □

Using these two Lemmas, we now give the main ideas of the proof of Theorem 3.1. First, we upper-bound OPT through a dual function involving the convex conjugate $R^*$, with similar arguments as in Balseiro et al. (2021); Celli et al. (2022). Then we need to lower-bound ALG with this dual function minus the regret. The main difficulty is that the source virtual rewards distribution changes with $\lambda_t$, and so does the average $\lambda_t$ target through $\pi_t$. The performance of ALG can be decomposed at each step $t$ into the sum of the virtual reward and a term in $R^*(\lambda_t)$ encoding the fairness penalty. We deal with the virtual rewards using an adversarial bandits algorithm able to handle any reward sequence using techniques for adversarial bandits algorithm from Hazan (2022); Lattimore & Szepesvári (2020), as the $\lambda_t$ are generated by a quite complicated Markov-Chain. These rewards are bounded because of Lemma 3.2. This yields the two regret terms in $\sqrt{K\log(K)}$, the second one stems from the difference between $\bar{\Delta}$ and $\Delta$. For the fairness penalty, using the online gradient descent on the $\lambda_t$ we end up being close to the penalty of the conditional expectations up to the regret term in $\mathrm{Diam}(\Delta)$. This last term is close to the true penalty through Lemma 3.3. This provides us with a computable regret bound where all the parameters are known beforehand. The full proof can be found in Appendix E.

## 3.5 Public contexts

We now go back to the general case with $|\mathcal{Z}| > 1$ finite. We would like to derive a good algorithm, which also takes into account the public information $z_t$. Reusing the analogy with bandit problems, this seems to be akin to the contextual bandit problem, where we would simply run the algorithm for each context in parallel. However, this would be incorrect: not only for a fixed public context does the non-separability of $R$ couples the source selection with the fairness penalty, it also couples all of the public contexts $z_t$ together. Hence the optimal policy is not to take the best policy for each context, which is once again different from the classical bandit setting.

The solution is to run an anytime version of the EXP3 algorithm for each public context in $\mathcal{Z}$, but to keep a common $\lambda_t$ for the evaluation of the virtual value. While it is technically not much more difficult and uses similar ideas to what was done for different sources when $|\mathcal{Z}| = 1$, the fact that only one of the "block" of the algorithm (the bandit part) needs to be parallelized, is quite specific and surprisingly simple.

**Proposition 3.4.** *For $\mu$ the probability distribution over $\mathcal{Z}$ finite, we can derive an algorithm that has a modified regret of order $\mathcal{O}(\sqrt{TK\log(K)}\sum_{z\in\mathcal{Z}}\sqrt{\mu(z)})$, where $\mu(z)$ is the probability that the public attribute is $z$.*

The algorithm and its analysis are provided in Appendix J.1. Note that in the worst case, when $\mathcal{Z}$ is finite, one has $\sum_{z \in \mathcal{Z}} \sqrt{\mu(z)} \leq \sqrt{|\mathcal{Z}|}$ by Jensen's inequality on the square root function.

If $\mathcal{Z}$ is not finite but a bounded subset of a vector space set of dimension $r$, such as $\mathcal{Z} = [0,1]^r$, additional assumptions on the smoothness of the conditional expectations are sufficient to obtain a sub-linear regret algorithm:

**Proposition 3.5.** *If the conditional expectations of $a_t$ and $u_t$ are both Lipschitz in $z$, then discretizing the space $\mathcal{Z} = [0,1]^r$ through an $\epsilon$-cover and applying the previous algorithm considering that one public context corresponds to one of the discretized bins, we can obtain a regret bound of order $\mathcal{O}(T^{(r+1)/(r+2)})$.*

The proof of this result uses standard discretization arguments under Lipschitz assumptions from Perchet & Rigollet (2013), which can also be found in Chapter 8.2 of Slivkins (2019) or in Exercise 19.5 of Lattimore & Szepesvári (2020). Specificities for this problem, such as these assumptions being enough to guarantee that $\varphi$ is Lipschitz in $z$, can be found in Appendix J.2.

# 4 Extensions

## 4.1 Other types of fairness penalty

The fairness penalty term of Equation (1) is quantified as $TR(\sum a_t x_t / T)$. While this term is the same as the one used in Celli et al. (2022) and can encode various fairness definitions (see the discussion in the aforementioned paper), this does not encompass all possible fairness notions. For instance, one may want to express fairness as a function of $\sum a_t x_t / \sum_t x_t$, which is the conditional empirical distribution of the user's protected attributes given that they were selected. This would lead to replacing the original penalty term by $(\sum_t x_t) R(\sum a_t x_t / \sum_t x_t)$.

As pointed out in Celli et al. (2022), one possible issue is that $R(\sum a_t x_t / \sum_t x_t)$, in general, is not convex in $x$, even if $R$ is convex. However $(\sum_t x_t) R(\sum a_t x_t / \sum_t x_t)$ is the perspective function of $R(A(.))$ (with $A$ the matrix with columns the $a_t$), which is thus convex. Hence, Algorithm 1 can be adapted to handle this new fairness penalty with two modifications. First, for the bandit part, we run the algorithm with a new virtual reward function, expressed as:

$$\tilde{\varphi}(\lambda_t, c_{tk}, k) = \max(\mathbb{E}[u_t \mid c_{tk}] - \langle \lambda_t, \mathbb{E}[a_t \mid c_{tk}] \rangle + R^*(\lambda_t), 0) - p_k.$$

This leads to an allocation $x_t = 1$ if $\mathbb{E}[u_t \mid c_{tk_t}] + R^*(\lambda_t) \geq \langle \lambda_t, \mathbb{E}[a_t \mid c_{tk_t}] \rangle$ and $x_t = 0$ otherwise.

Second, we modify the set on which the dual descent is done. The set $\Delta$ now becomes $\tilde{\Delta} = \mathrm{Conv}(\mathcal{A})$ (without the union with 0), and we now use $\tilde{\delta}_t = \mathbb{E}[a_t \mid c_{tk_t}]$ instead of $\delta_t = x_t \mathbb{E}[a_t \mid c_{tk_t}]$. Line (8) remains unchanged up to replacing $\delta$ and $\Delta$ by $\tilde{\delta}$ and $\tilde{\Delta}$. Finally the dual parameter update now becomes $\lambda_{t+1} = \lambda_t - \eta x_t (\gamma_t - \tilde{\delta}_t)$, which means that whenever $x_t = 0$, $\lambda_t$ does not change.

With these modifications, the following theorem (whose proof is very similar to the one of Theorem 3.1 and detailed in Appendix G), shows that we recover similar regret bounds as previously, with some modified constants due to the presence of $R^*$ in the virtual reward, and the modified $\Delta$. This yields a new class of usable fairness penalties which was previously not known to work.

**Theorem 4.1.** *Using Algorithm 1 with the modifications detailed above, the regret with respect to the objective with the modified penalty $(\sum_t x_t) R(\sum_t a_t x_t / \sum_t x_t)$ is of order $\mathcal{O}(\sqrt{T})$.*

## 4.2 Learning conditional utilities $\mathbb{E}[u_t \mid c_{tk}]$

To compute the virtual values, the algorithm relies on the knowledge of $\mathbb{E}[u_t \mid c_{tk}]$ for all possible context values. We now show how to relax this assumption even if the decision maker receives the feedback $u_t$ only when the user is selected ($x_t = 1$). We shall make the usual structural assumption of the classical stochastic linear bandit model. The analysis relies on Chapters 19 and 20 of Lattimore & Szepesvári (2020), and only the main ideas are given here. For simplicity we will assume that there are no public contexts ($|\mathcal{Z}| = 1$).

We assume that the contexts $c_{tk}$ are now feature vectors of dimension $q_k$, and that for all $k \in [K]$, there exists some vector $\psi_k \in \mathbb{R}^{q_k}$ so that

$$u_t - \langle \psi_k, c_{tk} \rangle, \tag{10}$$

is a zero mean 1-subgaussian random variable conditioned on the previous observation (see a more precise definition in Appendix H). We make classical boundedness assumptions, that for all $k$: $\|\psi_k\|_2 \le \bar{\psi}$, that $\|c_{tk}\|_2 \le \bar{c}$, and that $\langle \psi_k, c_{tk} \rangle \le 1$ for all possible values of $c_{tk}$.

Intuitively, we aim at running a stochastic linear bandit algorithm on each of the different sources for $T_k$ steps, where $T_k$ is the number of times that source $k$ is selected, but this breaks because of dependencies among the $k$ sources. Hence, what we do is to slightly change the rewards and contextual actions, so that we do something akin to artificially running $K$ bandits in parallel for $T$ steps. We force a reward and action of $0$ for source $k$ whenever it is not selected, and consider that each source is run for $T$ steps (even if it is actually selected less than $T$ times). Thus we can directly leverage and modify the existing analysis for the stochastic linear bandit. The full algorithm is given in Appendix H.

**Proposition 4.2.** *Given the above assumptions, the* added *regret of having to learn* $\mathbb{E}[u_t \mid c_{tk}]$ *is of order* $\mathcal{O}(\sqrt{KT} \log(T))$.

If the decision maker knows the optimal combination of sources, she can simply select sources proportionally to this combination, and actually learn the conditional expectation independently for each source. The worst case is to have to pull each arm $T/K$ times, which then incurs an additional regret with the same $\sqrt{K}$ constant. This shows that this bound is actually not too wasteful, as the cost of artificially running these bandits algorithm in parallel only impacts the logarithmic term.

# 5   Conclusion

We have shown how the problem of optimal data purchase for fair allocations can be tackled, using techniques from online convex optimization and bandits algorithms. We proposed a computationally efficient algorithm yielding a $\mathcal{O}(\sqrt{T})$ regret algorithm in the most general case. Interestingly, because of the non separable penalty $R$, the benchmark is different from a bandit algorithm, as randomization can strictly improve the performance for some instances, even though the setting is stochastic. We have also presented different types of fairness penalties that we can additionally tackle compared to previous works, in particular in the full information setting, and some instances where assumptions on the decision maker's knowledge can be relaxed.

Throughout the paper (even in Section 4.2 where we relax the assumption that $\mathbb{E}[u_t \mid c_{tk}]$ is known), we assumed that the decision-maker knows $\mathbb{E}[a_t \mid c_{tk}]$. This assumption is reasonable as this is related to demographic data and not to a utility that may be specific to the decision maker. Yet, this assumption can also be relaxed in specific cases. As an example, suppose that the decision maker can pay different prices to receive more or less noisy versions of $a_t$ that correspond to the data sources (the pricing could be related to different levels of Local Differential Privacy for the users, see Dwork & Roth (2014)). Then, under some assumptions, $\mathbb{E}[a_t \mid c_{tk}]$ can also be learned online—we defer the details to Appendix I.

The works of Balseiro et al. (2021) and Celli et al. (2022) can include hard constraints on some budget consumption when making the allocation decision $x_t$, which we did not include in our work. If this budget cost is measurable with respect to $c_{tk}$, then our analysis does not preclude using similar stopping time arguments as was done in these two works. However if this budget consumption is completely random, our algorithm and analysis can not be directly applied as this budget consumption may give additional information on the $a_t$ that was just observed. Regarding the i.i.d. assumption, in Balseiro et al. (2021) adversarial inputs are also considered, and the same could be done here for the contexts $c_{t,k}$ with similar results. However some stochasticity is still needed so that $\mathbb{E}[u_t \mid c_{t,k}]$ is well defined, which leads to an ambiguous stochastic-adversarial model. An open question would be to consider an intermediate case where $\mathbb{E}[u_t \mid c_{t,k}]$ would depend on $t$, and take into account learning for non-stationary distributions.

## Acknowledgements

This work has been partially supported by MIAI @ Grenoble Alpes (ANR-19-P3IA-0003), by the French National Research Agency (ANR) through grant ANR-20-CE23-0007, ANR-19-CE23-0015, and ANR-19-CE23-0026, and from the grant "Investissements d'Avenir" (LabEx Ecodec/ANR-11-LABX-0047).

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

# A Fairness Penalty

In the main part of the paper, we have considered a penalty with a general form $R(\sum_t a_t x_t / T)$. Here we give examples of fairness notions that can be encoded through this general penalty.

**Statistical parity**    Notions likes statistical parity are defined as constraints on the independence between protected attributes and selection decisions. More precisely, if $X$ denotes a treatment variable, and $A$ the protected attributes of an individual, we say that $X$ and $A$ are fair with respect to statistical parity if $X$ and $A$ are independent.

If $X \in \mathcal{X}$ and $A \in \mathcal{A}$ can take a finite number of values, this can be equivalently rewritten using probability distribution: $X$ satisfies statistical parity if for all $(x, a) \in \mathcal{X} \times \mathcal{A}$:

$$\Pr(A = a, X = x) = \Pr(A = a) \Pr(X = x). \tag{11}$$

To encode statistical parity in our framework, we consider that the $a_t$ are one-hot encoding of the protected attributes (*i.e*, $a_t$ is a vector with $d = |\mathcal{A}|$ dimensions that is equal to $0$ everywhere except on the coordinate corresponding to the protected attribute of individual $t$, that equals $1$). For the decision variables $x_t \in \{0, 1\}$, $\sum_t x_t$ is the number of individual selected and $\sum_t (a_t)_i x_t$ is the number of selected individual whose protected attributed is $i \in [d]$. Hence, (11) rewrites as: for all $i \in [d]$:

$$\frac{\sum_t (a_t)_i x_t}{T} = \left( \frac{\sum_t (a_t)_i}{T} \right) \left( \frac{\sum_t x_t}{T} \right).$$

Recall that the variables $a_t$ are *i.i.d.* and let us denote by $\alpha \in \mathcal{P}_d$ the probability distribution vector of the $a_t$. For large $T$, we have $\sum_t a_t / T \approx \alpha$. Which means that we will say that an allocation vector $\boldsymbol{x}$ satisfies statistical parity if $\sum_t a_t x_t / T = \sum_t \alpha x_t / T$, which can be equivalently rewritten as

$$\frac{\sum_t a'_t x_t}{T} = 0,$$

with $a'_t = a_t - \alpha$.

**Penalties $R$ corresponding to statistical parity**    In practice, achieving strict statistical parity is very constraining, and most of the time some relaxation of this constraint is allowed, at the price of some penalty (Ahani et al., 2021). In our paper, we penalize the non-independence through the function $R$. There are different choices of penalty function that can be used:

- A natural choice can be to penalize as a function of the distance between joint empirical distribution $a'_t x_t / T$ and $0$. For instance, one could use $R(a'_t x_t / T) = ||a'_t x_t / T||^\beta$ for any given norm $|| \cdot ||$ and any parameter $\beta > 0$.

- It is also possible to measure the non-independence with the empirical distribution of the $x_t$ conditioned on $a_t = a$, which should be equal regardless of the protected group $a \in \mathcal{A}$. In this case, the empirical distribution conditioned on the protected group $i \in [d]$ is $\sum_t a_{ti} x_t / \sum_t a_{ti} \approx \sum_t a_{ti} x_t / (\alpha_i T)$. Similarly we the previous case, $R$ can penalize how far is this vector from $0$.

In the two cases, the fairness penalty can be written as $T R(\sum_t a'_t x_t / T)$, for some modified $a'_t$ that are not necessarily one-hot encoder. This explains why we choose the set $\mathcal{A}$ to be more general than the set of one-hot encoders.

If we want to target another specific proportion for each protected group, we can replace $\alpha$ by any other distribution $\nu$. Note that another possibility is to instead use the empirical distribution of the $a_t$ conditioned on the $x_t$. This time the form of the vector is slightly different, it is $\sum_t a_t x_t / \sum_t x_t$; this is the type of penalty tackled and discussed in Section 4.1.

**Equality of opportunity**    Another popular fairness notion is the one of equality of opportunity (Hardt et al., 2016), where the probability distribution is further conditioned on another finite random variable $y_t$ taking some discrete value in $\mathcal{Y}$ that indicates some fitness of the individual. For $Y$ a

random variable with an outcome $y \in \mathcal{Y}$ considered as a positive opportunity, we say that $X$, $A$ and $Y$ are fair with respect to equality of opportunity, if

$$X \text{ and } A \text{ are independent given } [Y = y].$$

In order to adapt this notion in the online allocation setting, is sufficient to consider the cartesian product $\mathcal{A}' = \mathcal{A} \times \mathcal{Y}$ as an extended new set. Then if we would like to penalize with a metric related to equality of opportunity, we can for instance compose $R$ with the projection over the coordinates corresponding to $y_t = 1$.

Overall, this shows that the choice of $R$, the type of fairness vectors, and the choice of $\mathcal{A}$ are all crucial when designing a fairness penalty.

## B   A More General Online Allocation Setting

In the main body of the paper, we choose to simplify the exposition and to restrict ourselves to $x_t \in \{0, 1\}$ and to formulate utilities as scalar variables $u_t$. In the following, the regret bounds of Algorithm 1 are obtained in a setting that is more general than the one previously introduced:

- We now consider that the allocations decisions $x_t$ are in some set $\mathcal{X} \subset \mathbb{R}^n_+$. This makes it possible to tackle bipartite matching problems, by letting $\mathcal{X} = \{x \in \mathbb{R}^n_+ \mid \sum_{i=1}^n x_i \leq 1\}$ be a matching polytope.

- Instead of a random scalar $u_t$ being drawn, we suppose that a random function $f_t : \mathcal{X} \to \mathbb{R}$ is drawn. The functions $f_t$ are supposed u.s.c. and are all bounded in absolute value by some $\bar{f}$. This could simply be a deterministic concave function of $u_t$ for instance. Remark that when we assume that the conditional expectation can be computed, the decision maker's required knowledge is more demanding, as she needs to be able to compute it for all $x \in \mathcal{X}$.

- Instead of a random vector, a continuous random function $a_t : \mathcal{X} \to \mathcal{A}$ can be drawn, with $\|a_t(x)\|_2$ uniformly bounded for all $a_t$ and $x$, and we define $\Delta = \text{Conv}(\mathcal{A})$. We suppose that $\|a_t(x)\|_2 \leq 1$, which can be done without loss of generality.

The setting of Section 2.1 is recovered by using $f_t(x) = u_t x$, $a_t(x) = a_t x$ and $\mathcal{X} = \{0, 1\}$. For the more general setting, we need to redefine the virtual value with $f_t$ and $a_t$ as:

$$\varphi(\lambda, c_{tk}, k) = \max_{x \in \mathcal{X}} \mathbb{E}[f_t(x) - \langle \lambda_t, a_t(x) \rangle \mid c_{tk}] - p_k \tag{12}$$

This virtual reward is well defined by Lemma D.1. We rewrite Algorithm 1 using $f_t$ and this new virtual value to obtain Algorithm 2. There are two differences with Algorithm 1: First we use the argmax function in (13) instead of the explicit form of Equation (6); Second we use the virtual value function (12) in Equation (14) instead of the virtual value function of Equation (5).

## C   Reduction from auctions

In the rest of the paper, we assume that the decision variables are the values $x_t$. In this section, we explain how our setting can be adapted the the case of targeted advertisement, that are usually allocated through auction mechanisms. In the later, the decision variables are the bids but not the $x_t$: the variables $x_t$ are a consequence of the bids of the decision maker and of the bids of others.

**Model**   The model that we consider is as follows: for user $t$, after choosing a source of information $k_t$, the decision maker observes $c_{tk_t}$ and decides to bid $b_t \in \mathcal{B}$. The allocation $x_t = 1$ is given if her bid is higher than the maximum bid of the other bidders. If the bidder wins the auction, it then pays a price that depends on the auction format (*e.g.*, $b_t$ for first price auctions and the second highest bid for second price auctions). Assuming that the bid of the others are not affected by the decision maker's strategy, the bids of the others are *i.i.d.* (see Iyer et al. (2011); Balseiro et al. (2014); Flajolet & Jaillet (2017) for a discussion about the stationarity assumption). In this case, one falls back to our model with *i.i.d.* users, except that the decision variables are the variables $b_t$ and we have $x_t(b_t)$. We can apply our algorithm to this setting, by replacing the set of strategies $\mathcal{X}$ by the set of possible bids $\mathcal{B}$, which is also compact. For that, we need to consider a new utility function $b \mapsto u_t(b) x_t(b)$ which is a bounded u.s.c. function, and the new protected attributes function $b \mapsto a_t x_t(b)$. If $\mathbb{E}[a_t x_t(b) \mid c_{tk}]$ is continuous in $b$, this falls under the setting that we described in Appendix B.

---
**Algorithm 2** Online Fair Allocation with Source Selection — General algorithm
---
**Input:** Initial dual parameter $\lambda_0$, initial source-selection-distribution $\pi_0 = (1/K, \ldots, 1/K)$, dual gradient descent step size $\eta$, EXP3 step size $\rho$, cumulative estimated rewards $S_0 = 0 \in \mathbb{R}^K$.
**for** $t \in [T]$ **do**

    Draw a source $k_t \sim \pi_t$, where $(\pi_t)_k \propto \exp(\rho S_{(t-1)k})$ and observe $c_{tk_t}$.
    Compute the allocation for user $t$:

$$x_t = \arg\max_{x \in \mathcal{X}} \mathbb{E}[f_t(x) - \langle \lambda_t, a_t(x) \rangle \mid c_{tk_t}]. \tag{13}$$

    Update the estimated rewards sum and sources distributions for all $k \in [K]$:

$$S_{tk} = S_{t-1,k} + \hat{\varphi}(\lambda_t, c_{tk}, k_t), \tag{14}$$

    Compute the expected protected group allocation $\delta_t = \mathbb{E}[a_t(x_t) \mid c_{tk_t}, x_t]$ and compute the dual protected groups allocation target and update the dual parameter:

$$\gamma_t = \arg\max_{\gamma \in \Delta_{\delta_t}} \{ \langle \lambda_t, \gamma \rangle - \bar{R}(\gamma) \},$$
$$\lambda_{t+1} = \lambda_t - \eta(\gamma_t - \delta_t).$$

    **end for**
---

**Examples of utilities: first-price and second price auctions**    Let us denote by $v_t$ the value of the user $t$ for the decision maker, and by $d_t$ the highest bid of the others for this user. The decision maker will win the auction if $b_t \geq d_t$. Hence, the allocations will be $x_t(b_t) = \mathbb{1}_{b_t > d_t}$. Moreover, the utility function will be:

- $f_t(b_t) = (v_t - b_t)x_t(b_t)$ for first price auctions.
- $f_t(b_t) = (v_t - d_t)x_t(b_t)$ for second price auctions.

These functions are u.s.c with respect to $b_t$. Moreover, under mild assumption that the distribution of the highest bid of the others $d_t$ has a density, the function $\mathbb{E}[a_t x_t(b)|c_{tk_t}]$ is continuous. Hence, our algorithm can be applied.

**Difficulty: Knowledge is power**    To apply the algorithm, the main difficulty is to be able to compute the value $b_t$ that maximizes Equation (12). For that it is sufficient for the decision maker to have access to $\mathbb{E}[f_t(b_t)|c_{tk_t}]$ and $\mathbb{E}[a_t x_t(b_t)|c_{t,k_t}]$. This presupposes a lot of distributional knowledge from her part. In the case of full information with $a_t$ and $v_t$ directly given, Balseiro et al. (2021); Celli et al. (2022) were able to obtain an efficient reduction from online allocations to second price auctions, using the incentive compatible nature of this kind of auction. Similarly, Flajolet & Jaillet (2017) were able to derive efficient bidding strategies using limited knowledge for repeated auctions with budget constraints under some independence assumptions. Translating what is done in these two lines of work to our general model is difficult because we would need to rely on the assumption that $a_t$ and $d_t$ are independent conditioned on $c_{tk_t}$. While we think that this assumption is not reasonable because different agents might have access to different information sources, and because as the $a_t$ is common to all bidders it is correlated with all the bids hence also with $d_t$, we explain in the remark below how assuming independence can lead to a simple algorithm for second price auctions.

Some remarks:

- For our model, the first price and second price auctions are equally difficult: The amount of knowledge needed to compute $b_t$ and $\pi_t$ is the same for both cases and first price auction is not more difficult than the second-price auction setting (contrary to what usually happens).

- During this whole reduction from auctions, we have not assumed that the $d_t$ (or $x_t$) are observed as a feedback (whether the bidder wins the auction or not), and thus they are not assumed to be observed in the benchmark either. This is because observing $d_t$ or $x_t$ could give us access to information on $a_t$ that are not contained in $c_{tk_t}$.

- If we do make the assumption that $(u_t, a_t)$ is independent of $d_t$ conditionally on all $c_{tk}$, then we do recover an easy algorithm for second price auctions, and we can allow for a

benchmark taking into account the feedback $d_t$ (simply because now this feedback does not give any additional information for the estimation of $a_t$). Indeed, in that case the optimal bid is $b_t = \mathbb{E}[v_t - \langle \lambda_t, a_t \rangle \mid c_{tk}]$. This follows from similar argument made in Flajolet & Jaillet (2017), as for $b_t$ some $\mathcal{H}_t$ measurable bidding strategy we have:

$$
\begin{aligned}
\mathbb{E}[v_t x_t - \langle \lambda_t, a_t \rangle x_t - d_t x_t \mid c_{tk_t}] &= \mathbb{E}[\mathbb{E}[v_t x_t - \langle \lambda_t, a_t \rangle x_t - d_t x_t \mid \mathcal{H}_t, d_t] \mid c_{tk_t}] \\
&= \mathbb{E}[\mathbb{E}[v_t \mid c_{tk_t}] x_t - \langle \lambda_t, \mathbb{E}[a_t \mid c_{tk_t}] \rangle x_t - d_t x_t \mid c_{tk}] \\
&= \mathbb{E}[x_t (\mathbb{E}[v_t \mid c_{tk_t}] - \langle \lambda_t, \mathbb{E}[a_t \mid c_{tk_t}] \rangle - d_t) \mid c_{tk}] \\
&\leq \mathbb{E}[\max(\mathbb{E}[v_t \mid c_{tk_t}] - \langle \lambda_t, \mathbb{E}[a_t \mid c_{tk_t}] \rangle - d_t, 0) \mid c_{tk}],
\end{aligned}
$$

and this last inequality is reached for $b_t = \mathbb{E}[v_t - \langle \lambda_t, a_t \rangle \mid c_{tk}]$.

# D   Technical & Useful Lemmas

In this section, we prove several lemmas that are useful for the proof of the main Theorem 3.1.

## D.1   Preliminary lemmas

We first provide some simple lemmas. For completeness, we provide proofs of all results, even if they are almost direct consequences of the definitions.

**Lemma D.1.** *The virtual value function $\varphi$ is well defined by Equation* (12)*, as the $\arg\max$ is non-empty and the maximum is reached.*

*Proof.* We simply need to have that $\mathbb{E}[f_t(x) \mid c_{tk}]$ and $-\langle \lambda, \mathbb{E}[a_t(x) \mid c_{tk}] \rangle$ are u.s.c. in $x$. The continuity of $a_t$ is used to guarantee that $-\langle \lambda, a_t(x) \rangle$ is u.s.c. regardless of the values of $\lambda$. This is immediate from the definition of upper semi-continuity and the Reverse Fatou Lemma. We detail it here for $f_t$ for completeness. We say that $f_t$ is u.s.c. if and only if

$$
\forall x \in \mathcal{X} \text{ if there is a sequence } (x_n)_{n \in \mathbb{N}} \text{ so that } x_n \xrightarrow[n \to \infty]{} x, \text{ then } \limsup_{n \to \infty} f_t(x_n) \leq f_t(x).
$$

For $x \in \mathcal{X}$, let $(x_n)_{n \in \mathcal{N}}$ be a sequence so that $x_n \to x$. Let $P$ be a probability distribution over $f_t$ conditionally on $c_{tk}$. The function $|f_t(x_n)| \leq \bar{f}$, $f_t(x_n)$ is integrable, we can apply the Reverse Fatou Lemma for conditional expectation. If $f_t(x_n)$ is not positive, we can apply it to $f_t(x_n) + \bar{f}$ which is positive. Using the Reverse Fatou Lemma and the upper semi-continuity of $f_t$ we obtain:

$$
\limsup_{n \to \infty} \int f_t(x_n) dP \leq \int \limsup_{n \to \infty} f_t(x_n) dP \leq \int f_t(x) dP.
$$

This means that $x \mapsto \mathbb{E}[f_t(x) \mid c_{tk}]$ is u.s.c. over $\mathcal{X}$.

Because $\mathcal{X}$ is compact, and because the maximum of an u.s.c. function is reached over a compact set, the maximum is well defined.   □

**Lemma D.2.** *Let $\bar{R} : \delta \mapsto \inf_{\delta' \in \Delta} \{R(\delta') + L\|\delta - \delta'\|_2\}$. Then $\bar{R} = R$ over $\Delta$, $Lip(R) = Lip(\bar{R})$, and $\bar{R}$ is convex.*

*Proof.* The function $\bar{R}$ is obtained through the explicit formula of the Kirszbraun theorem, which states that we can extend functions over Hilbert spaces with the same Lipschitz constant. Hence the equality over $\Delta$ and the same Lipschitz constant.

It remains to prove the convexity. Note that because $R$ is proper, it is lower semi-continuous, $\|\delta - \cdot\|_2$ as well, and $\Delta$ is compact, hence the inf is reached and it is actually a minimum. Let $\delta_1, \delta_2$ in $\mathbb{R}^d$ and $\alpha \in [0, 1]$, and let $u_1$ and $u_2$ be the minimums of the optimization problem for respectively $\delta_1$ and $\delta_2$. Let $u = \alpha u_1 + (1 - \alpha)u_2$. We have $u \in \Delta$ because $\Delta$ is convex, and using that $R$ is convex and we have

$$
\begin{aligned}
R(u) + L\|(\alpha\delta_1 + (1 - \alpha)\delta_2) - u\|_2 &\leq \alpha(R(u_1) + L\|\delta_1 - u_1\|_2) + (1 - \alpha)(R(u_2) + L\|\delta_2 - u_2\|_2) \\
&= \alpha\bar{R}(\delta_1) + (1 - \alpha)\bar{R}(\delta_2),
\end{aligned}
$$

and taking the inf over $\Delta$ on the left hand-side we obtain the convexity of $\bar{R}$.   □

## D.2 Conditional expectation: notation and abuse of notations

In the analysis of the algorithm, we will consider conditional expectation by fixing some of the random variables to specific values. To simplify notations, we will use the following notation: if $X$ and $Y$ are random variables and $g$ is a measurable function, then we denote $E_X[g(X,Y)]$ the expectation with respect to $X$, that is:

$$E_X[g(X,Y)] = E[g(X',Y)|Y],$$

where $X'$ is a random variable independent of $Y$ that has the same law as $X$. This is equivalent to the random variable obtained by fixing $Y$ and taking the expectation with respect to the law of $X$ only.

In Algorithm 2, we have assumed that the decision maker is able to compute $\mathbb{E}[f_t(x_t) \mid c_{tk_t}]$ and $\mathbb{E}[a_t(x_t) \mid c_{tk_t}]$. While the law of $c_{tk_t}$ is complicated (because it involves all past decision up to time $t$), what the decision maker needs to compute is the value $\zeta(c_{tk_t}, x, k_t)$, where $\zeta(c, x, k) = \mathbb{E}[f_t(x) \mid c_{tk} = c]$. Those two quantities are equal because users are *i.i.d.* and $k_t$ and $f_t$ are independent (conditioned on $z_t$ and the history).

This manipulation is properly stated by Lemma D.3. Indeed by denoting $\mathcal{H}_{t-1}$ the whole history up to time $t-1$, and applying this Lemma: for $x_t$ the decision generated by the algorithm which is $(\mathcal{H}_{t-1}, k_t, c_{tk_t})$ measurable, we have that

$$\mathbb{E}[f_t(x_t) \mid c_{tk_t}, \mathcal{H}_{t-1}, k_t] = \zeta(c_{tk_t}, x_t, k_t) = \mathbb{E}_{f_t}[f_t(x_t) \mid c_{tk_t}]. \tag{15}$$

It is identical for $a_t(x_t)$: for $\zeta'(c, x, k) = \mathbb{E}[a_t(x) \mid c_{tk} = c]$ we have the following equality

$$\mathbb{E}[a_t(x_t) \mid c_{tk_t}, \mathcal{H}_{t-1}, k_t] = \zeta'(c_{tk_t}, x_t, k_t) = \mathbb{E}_{a_t}[a_t(x_t) \mid c_{tk_t}]. \tag{16}$$

The following Lemma is a modification of the so-called "Freezing Lemma" (see Lemma 8 of Cesari & Colomboni (2020)) which helps simplifying conditional expectations by allowing you to "fix" the random variable if it is measurable with respect to the conditioning of the expectation:

**Lemma D.3.** *Let $X$,$Y$ and $Z$ be random variables with $(X, Z)$ independent of $Y$, and $g$ some bounded measurable function. Let us denote by $\sigma(X)$ the sigma-algebra generated by the random variable $X$. Then*
$$\mathbb{E}[g(X,Y,Z) \mid \sigma(Y,Z)] = \mathbb{E}_X[g(X,Y,Z) \mid \sigma(Z)].$$

*Proof.* Let $V(Y,Z) = \mathbb{E}_X[g(X,Y,Z) \mid \sigma(Z)]$. To show that $W$ is the conditional expectation of $g(X,Y,Z)$ given $\sigma(Y,Z)$, we need to show that $V$ is $\sigma(Y,Z)$ measurable and that for any bounded $W$ that is $\sigma(Y,Z)$ measurable we have

$$\mathbb{E}[VW] = \mathbb{E}[g(X,Y,Z)W].$$

We denote by $\mu$ the distribution of $(X,Z)$ and $\nu$ the distribution of $Y$. By the Doob-Dynkin Lemma, because $W$ is $\sigma(Y,Z)$ measurable, we have that $W = h(Y,Z)$ where $h$ is measurable. First, by independence of $Y$ and $(X,Z)$, we have that the distribution of $(X,Y,Z)$ is $d\nu \otimes d\mu$. Thus

$$\mathbb{E}[g(X,Y,Z)W] = \int \int g(x,y,z)h(y,z)d\nu(x,z)d\nu(y).$$

Now let us compute the other expectation using Fubini theorem as both $g$ and $h$ are bounded:

$$\begin{aligned}
\mathbb{E}[VW] &= \int \int V(y,z)h(y,z)d\mu(z)d\nu(y) \\
&= \int d\nu(y) \left( \int V(y,z)h(y,z)d\mu(z) \right) \\
&= \int d\nu(y) \left( \mathbb{E}_Z[V(y,Z)h(y,Z)] \right).
\end{aligned}$$

The random variable $h(y, Z)$ is $\sigma(Z)$ measurable, thus because $V(y, Z)$ is a conditional expectation given $\sigma(Z)$, we have

$$\mathbb{E}_Z[V(y,Z)h(y,Z)] = \mathbb{E}_{X,Z}[g(X,y,Z)h(y,Z)] = \int g(x,y,z)h(y,z)d\mu(x,z).$$

Using Fubini again we obtain

$$\mathbb{E}[VW] = \int d\nu(y) \left( \int g(x, y, z)h(y, z)d\mu(x, z) \right)$$
$$= \int \int g(x, y, z)h(y, z)d\mu(x, z)d\nu(y)$$
$$= \mathbb{E}[g(X, Y, Z)W],$$

which proves that this is indeed the conditional expectation, as the lemma stated. $\square$

### D.3 Boundedness of $\lambda_t$ — proof of Lemma 3.2

We next show the proof of the boundedness of the $\lambda_t$ for the sequence generated by our Algorithm 1.

**Lemma.** *Let $\lambda_0 \in \mathbb{R}^d$, $\eta > 0$, and an arbitrary sequences $(\delta_1, \delta_2, \dots) \in \Delta^{\mathbb{N}}$. We define recursively $\lambda_t$ as follows:*

$$\gamma_t = \arg\max_{\gamma \in \Delta_{\delta_t}} \{\langle \lambda_t, \gamma \rangle - \bar{R}(\gamma)\},$$
$$\lambda_{t+1} = \lambda_t + \eta(\delta_t - \gamma_t).$$

*Then for $\Lambda = \cup_{\delta \in \bar{\Delta}} \partial\bar{R}(\delta)$ we have the following upper bound:*

$$\|\lambda_t\|_2 \leq L + 2\eta \operatorname{Diam}(\Delta) + \operatorname{Dist}(\lambda_0, \Lambda + B(0, 2\eta \operatorname{Diam}(\Delta))),$$

*where $B(0, 2\eta \operatorname{Diam}(\Delta))$ is the ball centered in $0$ and of diameter $2\eta \operatorname{Diam}(\Delta)$, with the diameter and the distance taken with respect to the euclidean $L_2$ norm.*

*Proof.* Recall that $\bar{\Delta} = \cup_{\delta \in \Delta} \Delta_\delta$, with $\Delta_\delta = \{\gamma : \|\gamma - \delta\|_2 \leq \operatorname{Diam}(\Delta)\}$. Hence, for any $\delta \in \Delta$ and $\gamma \in \bar{\Delta}$, we have $\|\gamma - \delta\|_2 \leq 2\operatorname{Diam}(\Delta)$. Let $\Lambda = \cup_{\delta \in \bar{\Delta}} \partial\bar{R}(\delta)$, and $\mathcal{C} = \operatorname{Conv}(\Lambda) + B(0, \eta 2\operatorname{Diam}(\Delta))$. Using that $\bar{R}$ is L-Lipschitz, we have that $\Lambda$ is bounded by $L$ for the norm $\|\cdot\|_2$, and so does $\operatorname{Conv}(\Lambda)$. Thus $\mathcal{C}$ is bounded by $L + 2\eta \operatorname{Diam}(\Delta)$ for $\|\cdot\|_2$. This implies that, to prove the theorem, it suffices to show that $\operatorname{Dist}(\lambda_{t+1}, \mathcal{C}) \leq \operatorname{Dist}(\lambda_t, \mathcal{C})$ is true for all $t$. Indeed, this would imply that

$$\|\lambda_t\|_2 \leq \sup_{c \in \mathcal{C}} \|c\|_2 + \operatorname{Dist}(\lambda_t, \mathcal{C}) \leq L + 2\eta \operatorname{Diam} \Delta + \operatorname{Dist}(\lambda_0, \mathcal{C}).$$

In the remainder of the proof, we show that $\operatorname{Dist}(\lambda_{t+1}, \mathcal{C}) \leq \operatorname{Dist}(\lambda_t, \mathcal{C})$. First, let us remark that $\gamma_t$ is the solution of a convex optimization problem, wit a constraint set $\Delta_{\delta_t}$ that can be rewritten as a single inequality : $h_t(\gamma) \leq 0$, with $h_t(\gamma) = \|\gamma - \delta_t\|_2^2 - \operatorname{Diam}(\Delta)^2$. The derivative of this function $h_t()$ is $\nabla h_t(\gamma) = 2(\gamma - \delta_t)$. As the point $\delta_t$ is in the interior (assuming that $\operatorname{Diam}(\Delta) > 0$) of $\Delta_{\delta_t}$, Slater's conditions hold, and we can apply the KKT conditions at the optimal point $(\gamma_t)$. This implies that there exists $\mu \geq 0$ such that $0 \in -\lambda_t + \partial\bar{R}(\gamma_t) + \mu 2(\gamma_t - \delta_t)$. This implies that there exists $\lambda_{\delta_t} \in \partial\bar{R}(\gamma_t)$ such that

$$\lambda_{\delta_t} - \lambda_t = 2\mu(\delta_t - \gamma_t).$$

The norm of the left-hand side of the above equation must be equal to the norm of its right-hand side, which means that $\|\lambda_{\delta_t} - \lambda_t\|_2 = 2\mu\|\delta_t - \gamma_t\|_2$. This determines the value of $\mu$. Let $\alpha_t = \eta\|\delta_t - \gamma_t\|_2/\|\lambda_{\delta_t} - \lambda_t\|_2$, the value $\lambda_{t+1}$ is therefore equal to:

$$\lambda_{t+1} = \lambda_t + \eta(\delta_t - \gamma_t) = \lambda_t + \alpha_t(\lambda_{\delta_t} - \lambda_t) = (1 - \alpha_t)\lambda_t + \alpha_t\lambda_{\delta_t}.$$

We now consider multiple cases:

*Case 1.* If $\alpha_t > 1$, then we rewrite $\lambda_{t+1}$ as:

$$\lambda_{t+1} = \lambda_{\delta_t} + (1 - \alpha_t)(\lambda_t - \lambda_{\delta_t})$$
$$= \lambda_{\delta_t} + \eta(\|\delta_t - \gamma_t\|_2 - \|\lambda_{\delta_t} - \lambda_t\|_2)\frac{\lambda_t - \lambda_{\delta_t}}{\|\lambda_t - \lambda_{\delta_t}\|_2}.$$

By assumption, the $\lambda_{\delta_t} \in \partial\bar{R}(\gamma_t) \subset \Lambda$. Moreover, the norm of the second term is bounded by $2\eta \operatorname{Diam}(\Delta)$ because

$$|\|\delta_t - \gamma_t\|_2 - \|\lambda_{\delta_t} - \lambda_t\|_2| = \|\delta_t - \gamma_t\|_2 - \|\lambda_{\delta_t} - \lambda_t\|_2 \leq 2\operatorname{Diam}(\Delta).$$

This implies that $\lambda_{t+1} \in \mathrm{Conv}(\Lambda) + B(0, 2\eta\,\mathrm{Diam}(\Delta)) = \mathcal{C}$, and therefore that $\mathrm{Dist}(\lambda_{t+1}, \mathcal{C}) = 0 \le \mathrm{Dist}(\lambda_t, \mathcal{C})$.

*Case 2.* If $\alpha_t \le 1$ and $\lambda_t \in \mathcal{C}$. In this case, $\lambda_{t+1}$ is a convex combination of $\lambda_{\delta_t}$ and $\lambda_t$. As, $\lambda_{\delta_t} \in \Lambda \subset \mathcal{C}$ and $\lambda_t \in \mathcal{C}$, and because $\mathcal{C}$ is convex, we have that $\lambda_{t+1} \in \mathcal{C}$.

*Case 3.* If $\alpha_t \le 1$ and $\lambda_t \notin \mathcal{C}$, we define $\pi_t = \lambda_t$ the projection of $\lambda_t$ on $\mathcal{C}$. Let $y_t = (1 - \alpha_t)\pi_t + \alpha_t\lambda_{\delta_t}$. Because $\mathcal{C}$ is convex, $\alpha_t \in [0, 1]$, $\lambda_{\delta_t} \in \mathcal{C}$, and $\pi_t \in \mathcal{C}$, we have $y_t \in \mathcal{C}$. This implies that

$$\begin{aligned}
\mathrm{Dist}(\lambda_{t+1}, \mathcal{C}) &\le \|\lambda_{t+1} - y_t\|_2 \\
&= (1 - \alpha_t)\|\lambda_t - \pi_t\|_2 \\
&= (1 - \alpha_t)\,\mathrm{Dist}(\lambda_t, \mathcal{C}) \\
&\le \mathrm{Dist}(\lambda_t, \mathcal{C}).
\end{aligned}$$

This shows that for all cases, $\mathrm{Dist}(\lambda_{t+1}, \mathcal{C}) \le \mathrm{Dist}(\lambda_t, \mathcal{C})$, which concludes the proof. $\qquad\square$

Using this previous Lemma, we can simply bound the virtual reward functions.

**Lemma D.4.** *For $\lambda_0 \in \partial R(0)$, and for $\lambda_t$ generated according to Algorithm 1, the virtual reward function*

$$\varphi(\lambda_t, c_{tk}, k) = \max_{x \in \mathcal{X}} \mathbb{E}[f_t(x) - \langle \lambda_t, a_t(x) \rangle \mid c_{tk}] - p_k,$$

*is bounded in absolute value by*

$$m := \bar{f} + L + 2\eta\,\mathrm{Diam}(\Delta) + \max_k |p_k|.$$

*Proof.* Because $a_t(x)$ used for the update of $\lambda_t$ is in $\Delta$ for all $t$, we can apply Lemma 3.2 to bound the trajectory of the $(\lambda_t)_{t \in [T]}$ generated by the algorithm. Using a triangle inequality first, then Cauchy-Schwartz inequality, and finally the boundedness assumptions on $f_t$ and $a_t$ with the Lemma, we have

$$\begin{aligned}
|\max_{x \in \mathcal{X}} \mathbb{E}[f_t(x) - \langle \lambda_t, a_t(x) \rangle \mid c_{tk}] - p_k| &\le |\max_{x \in \mathcal{X}} \mathbb{E}[f_t(x) \mid c_{tk}]| + |\max_{x \in \mathcal{X}} \mathbb{E}[\langle \lambda_t, a_t(x) \rangle \mid c_{tk}]| + |p_k| \\
&\le |\max_{x \in \mathcal{X}} \mathbb{E}[f_t(x) \mid c_{tk}]| + |\max_{x \in \mathcal{X}} \mathbb{E}[\|\lambda_t\|_2 \|a_t(x)\|_2 \mid c_{tk}]| + |p_k| \\
&\le \bar{f} + L + 2\eta\,\mathrm{Diam}(\Delta) + \max_k |p_k|.
\end{aligned}$$

$\square$

**Lemma D.5.** *For $\lambda_0 \in \partial R(a)$ (for some $a \in \mathcal{A}$), and for $\lambda_t$ generated according to Algorithm 1, if the quantity $a_t(x)/x$ is always well defined and in $\Delta$ for all $a_t$ and all $x$, then the virtual reward function*

$$\tilde{\varphi}(\lambda_t, c_{tk}, k) = \max_{x \in \mathcal{X}} \left( \mathbb{E}[f_t(x) - \langle \lambda_t, a_t(x) \rangle \mid c_{tk}] + R^*(\lambda_t) \right) - p_k,$$

*is bounded in absolute value by*

$$\tilde{m} := \bar{f} + L + 2\eta\,\mathrm{Diam}(\Delta) + \max\{R^*(\lambda) \mid \|\lambda\|_2 \le L + 2\eta\,\mathrm{Diam}(\Delta)\} + \max_k |p_k|.$$

*Proof.* The proof of this Lemma is identical to the previous one, except that we also need to bound $R^*$. As a convex conjugate, $R^*$ is convex. By Corollary 10.1.1 of Rockafellar (1970), every convex function from $\mathbb{R}^d$ to $\mathbb{R}$ is continuous. By Lemma 1 of Celli et al. (2022), we know that the domain of $R^*$ is indeed $\mathbb{R}^d$, hence it is continuous.

Now let us look at the modified dual update $\lambda_{t+1} = \lambda_t + \eta(\gamma_t x_t - a_t(x_t))$. We can factorize by $x_t$ to recover $\lambda_{t+1} = \lambda_t + x_t\eta(\gamma_t - a_t(x_t)/x_t)$. Because of our hypothesis, $a_t(x_t)/x_t \in \Delta$, therefore we can apply Lemma 3.2 with $\eta_t = \eta x_t$, which is upper bounded by $\eta$. Hence the $\lambda_t$ are all in $\{\lambda \in \mathbb{R}^d \mid \|\lambda\|_2 \le L + 2\eta\,\mathrm{Diam}(\Delta)\}$ which is compact. By continuity of $R^*$, it is continuous over this compact set hence bounded, and the virtual reward is bounded by $\tilde{m}$ overall. $\qquad\square$

*Remark.* We made the technical assumption that $a_t(x)/x \in \Delta$. It is clear that when $a_t$ is linear, such as $a_t x$, then $a_t x/x = a_t \in \mathcal{A} \subset \Delta$ and the condition is satisfied. Regardless, if $a_t(x)/x$ is always well defined and bounded, we can simply increase the size of $\Delta$, so that this quantity always remain in $\Delta$. Of course, this will degrade the performance of the algorithm as it depends on $\mathrm{Diam}(\Delta)$.

## D.4 Expected difference between penalty and penalty of expectation — Proof of Lemma 3.3

We next prove the following lemma, which tells us that observing the conditional expectation of the $a_t(x_t)$ is enough to observe the expectation of the penalty $R$:

**Lemma.** *For $(x_1, \ldots, x_T)$ and $(k_1, \ldots, k_T)$ generated according to Algorithm 1, we have the following upper bound:*

$$|\mathbb{E}[R(\frac{1}{T}\sum_{t=1}^{T}a_t x_t) - R(\frac{1}{T}\sum_{t=1}^{T}\mathbb{E}_{a_t}[a_t(x_t) \mid c_{tk_t}])]| \leq 2L\sqrt{\frac{d}{T}}.$$

*Proof.* Let us denote by $\delta_t = \mathbb{E}_{a_t}[a_t(x_t) \mid c_{tk_t}]$ the expectation of $a_t(x_t)$ with $x_t$ fixed. Using first the triangle inequality for the expectation, and then the Lipschitzness of $R$ we have that

$$|\mathbb{E}[R(\frac{1}{T}\sum_{t=1}^{T}a_t(x_t)) - R(\frac{1}{T}\sum_{t=1}^{T}\mathbb{E}_{a_t}[a_t(x_t) \mid c_{tk_t}])]| \leq \mathbb{E}[|R(\frac{1}{T}\sum_{t=1}^{T}a_t(x_t)) - R(\frac{1}{T}\sum_{t=1}^{T}\delta_t)|]$$

$$\leq \frac{L}{T}\mathbb{E}[\|\sum_{t=1}^{T}(a_t(x_t) - \delta_t)\|_2]$$

$$\leq \frac{L}{T}\sqrt{\mathbb{E}[\|\sum_{t=1}^{T}(a_t(x_t) - \delta_t)\|_2^2]}, \qquad (17)$$

where we used Jensen's inequality with $x \mapsto \sqrt{x}$ for the last inequality.

We denote by $\delta_{t,l}$ the $l$-th coordinate of $\delta_t$ for $l \in [d]$, and similarly for $a_{t,l}$. We have:

$$\mathbb{E}[\|\sum_{t=1}^{T}(a_t(x_t) - \delta_t)\|_2^2] = \sum_{l=1}^{d}\mathbb{E}[\left(\sum_{t=1}^{T}(a_{t,l}(x_t) - \delta_{t,l})\right)^2]$$

By Equation (16), the sequence $a_{t,l}(x_t) - \delta_{t,l}$ is a Martingale difference sequence for the filtration $\sigma((a_\tau, c_{\tau k_\tau}, k_\tau)_{\tau \in [t-1]}, k_t, c_{tk_t})$. Hence, using that the expectation is 0 by the law of total expectation, we obtain that

$$\mathbb{E}\left[\left(\sum_{t=1}^{T}(a_{t,l}(x_t) - \delta_{t,l})\right)^2\right] = \text{Var}[\sum_{t=1}^{T}(a_{t,l}(x_t) - \delta_{t,l})] = \sum_{t=1}^{T}\text{Var}[a_{t,l}(x_t) - \delta_{t,l}]. \qquad (18)$$

The terms $a_{t,l}(x_t) - \delta_{t,l}$ are bounded in $[-2, 2]$ by the assumption that $\|a_t(x)\|_2 \leq 1$, therefore we can bound each of these variances by 4, for a total bound of $4T$. Plugging this into Equation (17) concludes the proof. $\qquad \square$

# E    Proof of the main Theorem 3.1

The proof builds upon the proof used in the full information case ($a_t$ and $f_t$ are given) from Balseiro et al. (2021); Celli et al. (2022).

In order to bound the regret, we upper bound the performance of OPT, and lower bound the performance of ALG. We proceed by first upper-bounding the optimum.

Recall that the convex conjugate $R^*$ of $R$ (which we consider as a function from $\mathbb{R}^d \to \mathbb{R}$ that is $+\infty$ outside of $\Delta$) is a function that associates to a dual parameter $\lambda$ the quantity $R^*(\lambda)$:

$$R^*(\lambda) = \max_{\gamma \in \mathbb{R}^d}\{\langle \gamma, \lambda \rangle - R(\gamma)\} = \max_{\gamma \in \Delta}\{\langle \gamma, \lambda \rangle - R(\gamma)\}.$$

The Fenchel-Moreau theorem, tells us that for a proper convex function, the biconjugate is equal to the original function: $R^{**} = R$.

Recall that $\varphi$ is defined in Equation (5). We define the dual vector function $\mathcal{D}(\lambda) = (\mathcal{D}(\lambda, 1), \ldots, \mathcal{D}(\lambda, K))$ which is a vector representing the value of the dual conjugate problem. For coordinate $k \in [K]$, it is defined as

$$\mathcal{D}(\lambda, k) = \mathbb{E}[\varphi(\lambda, c_{tk}, k)] + R^*(\lambda).$$

We denote the unit simplex of dimension $K$ by $\mathcal{P}_K = \{\pi \in \mathbb{R}^K \mid \pi_k \geq 0, \sum_{k=1}^K \pi_k = 1\}$.

**Lemma E.1.** *We have the following upper-bound for the offline optimum:*

$$\text{OPT} \leq T \sup_{\pi \in \mathcal{P}_K} \inf_{\lambda \in \mathbb{R}^d} \langle \pi, \mathcal{D}(\lambda) \rangle,$$

*Proof.* We upper bound the performance of OPT through a Lagrangian relaxation using the convex conjugate of $R$, as was done similarly in Jenatton et al. (2016).

Let $(k_1, \ldots, k_T) \in [K]^T$ be the (deterministic) sequence of sources selected, and $(x_1, \ldots, x_T) \in \mathcal{X}^T$ the offline allocation. These $x_t$ are all $\sigma(c_{1k_1}, \ldots, c_{Tk_T})$ measurable. We define $\tilde{f}_t(x) = \mathbb{E}_{f_t}[f_t(x) \mid c_{tk_t}]$ and $\delta_t = \mathbb{E}_{a_t}[a_t(x_t) \mid c_{tk_t}]$.

Using Jensen's inequality for $R$ convex we have

$$\mathbb{E}[\mathcal{U}(\boldsymbol{k}, \boldsymbol{x}) \mid c_{1k_1}, \ldots, c_{Tk_T}] = \sum_{t=1}^T \mathbb{E}[f_t(x_t) - p_{k_t} \mid c_{1k_1}, \ldots, c_{Tk_T}] - T\mathbb{E}[R(\frac{1}{T}\sum_{t=1}^T a_t(x_t)) \mid c_{1k_1}, \ldots, c_{Tk_T}]$$

$$\leq \sum_{t=1}^T \mathbb{E}[f_t(x_t) - p_{k_t} \mid c_{1k_1}, \ldots, c_{Tk_T}] - TR(\mathbb{E}[\frac{1}{T}\sum_{t=1}^T a_t(x_t) \mid c_{1k_1}, \ldots, c_{Tk_T}]).$$

Because the $x_t$ are $\sigma(c_{1k1}, \ldots, c_{Tk_T})$ measurable, and by independence of the $(f_t, a_t, c_{t1}, \ldots, c_{tK})$ we have that $\mathbb{E}[a_t(x_t) \mid \sigma(c_{1k_1}, \ldots, c_{tk_T})] = \mathbb{E}_{a_t}[a_t(x_t) \mid c_{tk_t}] = \delta_t$. With the same argument we obtain that $\mathbb{E}[f_t(x_t) \mid \sigma(c_{1k_1}, \ldots, c_{tk_t})] = \tilde{f}_t(x_t)$. Therefore

$$\mathbb{E}[\mathcal{U}(\boldsymbol{k}, \boldsymbol{x}) \mid c_{1k_1}, \ldots, c_{Tk_T}] \leq \sum_{t=1}^T \tilde{f}_t(x_t) - p_{k_t} - TR(\frac{1}{T}\sum_{t=1}^T \delta_t). \qquad (19)$$

We define the function $\mathcal{L} : (\boldsymbol{k}, \mathbf{c}, \mathbf{x}, \lambda) \mapsto \sum_{t=1}^T \tilde{f}_t(x_t) - p_{k_t} - \langle \lambda, \delta_t \rangle + TR^*(\lambda)$.

We now show that this function is always greater than the conditional expectation of $\mathcal{U}$:

$$\mathcal{L}(\boldsymbol{k}, \mathbf{c}, \mathbf{x}, \lambda) \geq \inf_{\lambda \in \mathbb{R}^d} \mathcal{L}(\boldsymbol{k}, \mathbf{c}, \mathbf{x}, \lambda)$$

$$= \sum_{t=1}^T \tilde{f}_t(x_t) - p_{k_t} - T \sup_{\lambda \in \mathbb{R}^d} \{\langle \lambda, \frac{1}{T}\sum_{t=1}^T \delta_t \rangle - R^*(\lambda)\}$$

$$= \sum_{t=1}^T \tilde{f}_t(x_t) - p_{k_t} - TR^{**}(\frac{1}{T}\sum_{t=1}^T \delta_t)$$

$$= \sum_{t=1}^T \tilde{f}_t(x_t) - p_{k_t} - TR(\frac{1}{T}\sum_{t=1}^T \delta_t)$$

$$\geq \mathbb{E}[\mathcal{U}(\boldsymbol{k}, \boldsymbol{x}) \mid \mathbf{c}_{\boldsymbol{k}}], \qquad (20)$$

the last equality results from applying Fenchel-Moreau theorem, and the last inequality is obtained by using Equation (19).

Let us now compute the max in $(x_1, \ldots, x_T)$ of $\mathcal{L}$. Looking at the terms inside the sum, the only possible dependency of $\tilde{f}_t(x_t) - p_{k_t} - \langle \lambda, \delta_t \rangle$ to other contexts for $t' \neq t$, is possible through the allocation $x_t$, which can depend on the other contexts. Therefore the maximum of the sum is the sum of the maximums. This yields that

$$\max_{(x_1, \ldots, x_T)} \mathcal{L}(\boldsymbol{k}, \mathbf{c}, \mathbf{x}, \lambda) = \sum_{t=1}^T \max_{x_t \in \mathcal{X}} \mathbb{E}_{f_t, a_t}[f_t(x_t) - p_{k_t} - \langle \lambda, a_t(x_t) \rangle \mid c_{tk_t}] + TR^*(\lambda)$$

$$= \sum_{t=1}^T (\varphi(\lambda, c_{tk_t}, k_t) + R^*(\lambda)).$$

Thus by taking the $\max$ in $(x_1, \ldots, x_T)$ on both sides of Equation (20) and then taking the expectation (which is over the contexts as everything is measurable with respect to them), we obtain

$$\mathbb{E}[\max_{(x_1,\ldots,x_T)} \mathbb{E}[\mathcal{U}(\boldsymbol{k}, \boldsymbol{x}) \mid c_{1k_1}, \ldots, c_{Tk_T}]] \leq \sum_{t=1}^{T} \mathbb{E}[\varphi(\lambda, c_{tk_t}, k_t) + R^*(\lambda)] = \sum_{t=1}^{T} \mathcal{D}(\lambda, k_t). \quad (21)$$

What is nice here, is that by looking at the dual problem which was originally to address the non-separability in $x$, we also have the separability in $k_t$.

We denote by $T_k = \sum_{t=1}^{T} \mathbb{1}[k_t = k]$ the number of times source $k$ is selected, and by $\hat{\pi} = (T_1/T, \ldots, T_K/T) \in \mathcal{P}_K$ the selection proportions of each source $k$ over the $T$ rounds. This vector of proportions can be seen as the empirical distribution of selected sources. We can rewrite in the following way the dual objective

$$\sum_{t=1}^{T} \mathcal{D}(\lambda, k_t) = \sum_{k=1}^{K} T_k \mathcal{D}(\lambda, k) = T \sum_{k=1}^{K} \hat{\pi}_k \mathcal{D}(\lambda, k) = T\langle \hat{\pi}, \mathcal{D}(\lambda) \rangle.$$

Finally, using the previous equation with Equation (21), taking first the $\inf$ in $\lambda$ and then the $\max$ in $k_t$ we can conclude:

$$\mathrm{OPT} \leq T \max_{k_1,\ldots,k_T} \inf_{\lambda \in \mathbb{R}^d} \langle \hat{\pi}, \mathcal{D}(\lambda) \rangle \leq T \sup_{\pi \in \mathcal{P}_K} \inf_{\lambda \in \mathbb{R}^d} \langle \pi, \mathcal{D}(\lambda) \rangle.$$

$\square$

*Remark.* Note that the function $\mathcal{D}(\lambda)$ is convex and lower semi-continuous (l.s.c.) (as the supremum of a family of l.s.c. functions). In addition, $\langle \pi, \mathcal{D}(\lambda) \rangle$ is concave and u.s.c. in $\pi$ by linearity, and $\mathcal{P}_K$ is compact convex. Hence Sion's minimax theorem can be applied, and the $\sup$ in $\pi$ is actually a $\max$.

We can now proceed to the rest of the proof of the main Theorem 3.1:

**Theorem E.2.** *For* $\eta = L/(2\,\mathrm{Diam}(\Delta)\sqrt{T})$, $m = \bar{f} + L + \max_k |p_k| + 2\eta\,\mathrm{Diam}(\Delta)$, $\rho = \sqrt{\log(K)/(TKm^2)}$, *and* $\lambda_0 \in \partial R(0)$, *algorithm Algorithm 1 has the following regret upper bound:*

$$\mathrm{Reg} \leq 2((L + \bar{f} + \max_k |p_k|)\sqrt{K \log(K)} + L\sqrt{d} + \mathrm{Diam}(\Delta))\sqrt{T} + 2L\sqrt{K \log(K)}$$

*Proof.* We first lower bound the performance of the algorithm. Then we simply need to apply Lemma E.1 to get the regret bound. Let $\mathcal{H}_t$ be the natural filtration generated by the $(c_{\tau k_\tau}, k_\tau)_{\tau \in [t]}$. Let $x_t, k_t, \lambda_t$ and $\gamma_t$ defined as in 1. We denote by $\delta_t = \mathbb{E}_{a_t}[a_t(x_t) \mid c_{tk_t}]$. The quantity $\delta_t$ is in $\Delta$ because of the expectation and by convexity of $\Delta$. Note that here these are all random variables, with $\gamma_t$ and $\lambda_t$ being $\mathcal{H}_{t-1}$ measurable and $x_t$ being $\mathcal{H}_t$ measurable.

To first get a broad picture on how we are going to lower bound the utility of the algorithm, let us decompose it into different parts. It can be rewritten as

$$\mathbb{E}[\sum_{t=1}^{T} f_t(x_t) - p_{k_t} - TR(\frac{1}{T}\sum_{t=1}^{T} a_t(x_t))] = \mathbb{E}\left[\left(\sum_{t=1}^{T} f_t(x_t) - p_{k_t} - \bar{R}(\gamma_t)\right)\right. \quad (22)$$

$$+ \left(\sum_{t=1}^{T} \bar{R}(\gamma_t) - TR(\frac{1}{T}\sum_{t=1}^{T} \delta_t)\right) \quad (23)$$

$$+ \left.\left(TR(\frac{1}{T}\sum_{t=1}^{T} \delta_t) - TR(\frac{1}{T}\sum_{t=1}^{T} a_t(x_t))\right)\right]. \quad (24)$$

The first line (Equation (22)) corresponds to some adjusted separable rewards, and we bound it by first reducing it to the virtual rewards $\varphi$, and then using bandits algorithms techniques. The second line (Equation (23)) corresponds to the difference between the unfairness of the adjusted rewards, and the true unfairness, which is not big because of the gradient descent on $\lambda_t$. Finally the last line (Equation (24)) corresponds to the difference between the penalty of the expected allocation and the expected penalty of the allocation, which we have already shown in Lemma 3.3 is bounded by $\mathcal{O}(\sqrt{T})$.

**Reduction of Equation (22) to a bandit problem with** $\varphi$    Using first the tower property of the conditional expectation, and then $Equation$ (15), we have that

$$\mathbb{E}[f_t(x_t)] = \mathbb{E}\left[\mathbb{E}[f_t(x_t) \mid \mathcal{H}_t]\right] = \mathbb{E}\left[\mathbb{E}_{f_t}[f_t(x_t) \mid c_{tk_t}]\right].$$

We add and remove $\langle \lambda_t, \delta_t \rangle$, and use that $x_t$ is the allocation that yields the virtual reward function:

$$\mathbb{E}[f_t(x_t) - p_{k_t}] = \mathbb{E}\left[\mathbb{E}_{f_t}[f_t(x_t) \mid c_{tk_t}] - p_{k_t} - \langle \lambda_t, \delta_t \rangle + \langle \lambda_t, \delta_t \rangle\right]$$
$$= \mathbb{E}[\varphi(\lambda_t, c_{tk_t}, k_t) + \langle \lambda_t, \delta_t \rangle].$$

Using the definition of $\gamma_t$ as a maximizer, we have

$$-\bar{R}(\gamma_t) = -\bar{R}(\gamma_t) + \langle \lambda_t, \gamma_t \rangle - \langle \lambda_t, \gamma_t \rangle$$
$$= \max_{\gamma \in \Delta_{\delta_t}} \{\langle \lambda_t, \gamma \rangle - \bar{R}(\gamma)\} - \langle \lambda_t, \gamma_t \rangle.$$

Because $\Delta \subset \Delta_{\delta_t}$ and, because $\bar{R}$ and $R$ are equal over $\Delta$ we recover a lower bound with $R^*$:

$$\max_{\gamma \in \Delta_{\delta_t}} \{\langle \lambda_t, \gamma \rangle - \bar{R}(\gamma)\} - \langle \lambda_t, \gamma_t \rangle \geq \max_{\gamma \in \Delta}\{\langle \lambda_t, \gamma \rangle - \bar{R}(\gamma)\} - \langle \lambda_t, \gamma_t \rangle$$
$$= \max_{\gamma \in \Delta}\{\langle \lambda_t, \gamma \rangle - R(\gamma)\} - \langle \lambda_t, \gamma_t \rangle$$
$$= R^*(\lambda_t) - \langle \lambda_t, \gamma_t \rangle.$$

Thus

$$\mathbb{E}[f_t(x_t) - p_{k_t} - \bar{R}(\gamma_t)] \geq \mathbb{E}[\varphi(\lambda_t, c_{tk_t}, k_t) + R^*(\lambda_t) + \langle \lambda_t, \delta_t - \gamma_t \rangle]. \tag{25}$$

**Application of Bandits Algorithm**    Let $\pi_t$ be the distribution generated according to the algorithm. We actually have that $\mathbb{E}[\varphi(\lambda_t, c_{tk_t}) + R^*(\lambda_t)]$ is just (through independence, measurability, and total expectation arguments) $\mathbb{E}[\langle \pi_t, \mathcal{D}(\lambda_t) \rangle]$. Hence the main idea is to apply Online Convex Optimization theorems to the linear rewards $\mathcal{D}(\lambda_t)$ with the decision variable $\pi_t$ (see Hazan (2022), Orabona (2019) or Lattimore & Szepesvári (2020) for introductions to these types of problems). Here we don't have access either to $\mathcal{D}(\lambda_t)$, nor even to $\mathcal{D}(\lambda_t, k_t)$: we are in an adversarial bandit setting regarding the $\lambda_t$, and also in a stochastic setting regarding the expectation in $\mathcal{D}$ taken with respect to $c_t$. Hence we use an unbiased estimator of the gradient $\mathcal{D}(\lambda_t)$ for the linear gain, which we will describe further down.

Notice that it is crucial to be able to deal with any sequence $\mathcal{D}(\lambda_t)$, this is because these are neither independent nor related to martingales, as the $\lambda_t$ are generated from a quite complicated Markov Chain with states $(\lambda_t, c_t, \pi_t, k_t)$. Hence the outside expectation does not help us much, and we will face as losses (or here rewards) the random arbitrary sequence of the $\mathcal{D}(\lambda_t)$. We don't care about $R^*(\lambda_t)$ as it is $\mathcal{H}_{t-1}$ measurable.

Here the different arms of a bandit problem correspond to different sources. Using the tower rule, we have

$$\mathbb{E}\left[\sum_{t=1}^{T} \varphi(\lambda_t, c_{tk_t}, k_t)\right] = \mathbb{E}\left[\sum_{t=1}^{T} \mathbb{E}[\varphi(\lambda_t, c_{tk_t}, k_t) \mid \mathcal{H}_{t-1}]\right].$$

To use results for adversarial bandits, the bandits rewards need to be bounded. By Lemma D.4, the virtual reward $\varphi$, and consequently its conditional expectation, is bounded by $m := \bar{f} + L + 2\eta \operatorname{Diam}(\Delta) + \max_k |p_k|$. Therefore $\mathbb{E}[\varphi(\lambda_t, c_{tk}, k) \mid \mathcal{H}_{t-1}]/m \in [-1, 1]$.

We now construct an unbiased estimator of $\mathbb{E}[\varphi(\lambda_t, c_{tk}, k) \mid \mathcal{H}_{t-1}]$, which is the following importance weighted estimator:

$$\hat{\varphi}(\lambda_t, c_{tk}, k) = m - \mathbb{1}[k_t = k]\frac{(m - \varphi(\lambda_t, c_{tk}, k))}{\pi_{tk}}.$$

This estimator is unbiased for the conditional expectation given $\mathcal{H}_{t-1}$. Indeed because $\pi_{tk}$ is $\mathcal{H}_{t-1}$ measurable, and because $\mathbb{1}[k_t = k]$ and $\varphi(\lambda_t, c_{tk}, k)$ are independent conditionally on $\mathcal{H}_{t-1}$ we

have:

$$\mathbb{E}[\hat{\varphi}(\lambda_t, c_{tk}, k) \mid \mathcal{H}_{t-1}] = m - \frac{1}{\pi_{tk}}\mathbb{E}[(m - \varphi(\lambda_t, c_{tk}, k))\mathbb{1}[k_t = k] \mid \mathcal{H}_{t-1}]$$

$$= m - \frac{1}{\pi_{tk}}\mathbb{E}[m - \varphi(\lambda_t, c_{tk}, k) \mid \mathcal{H}_{t-1}]\mathbb{E}[\mathbb{1}[k_t = k] \mid \mathcal{H}_{t-1}]$$

$$= m - \mathbb{E}[m - \varphi(\lambda_t, c_{tk}, k) \mid \mathcal{H}_{t-1}]$$

$$= \mathbb{E}[\varphi(\lambda_t, c_{tk}, k) \mid \mathcal{H}_{t-1}],$$

which is exactly the reward needed.

We can now use the following Theorem 11.2 from Lattimore & Szepesvári (2020):

**Theorem** (EXP3 Regret Bound). *Let $X \in [0, 1]^{T \times K}$ be the rewards of an adversarial bandit, then running EXP3 with with learning rate $\rho = \sqrt{2\log(K)/(TK)}$ and denoting $k_t$ the arm chosen at time $t$, achieves the following regret bound:*

$$\max_{k \in K}\sum_{t=1}^{T} X_{t,k} - \mathbb{E}[\sum_{t=1}^{T} X_{t,k_t}] \leq \sqrt{2TK\log(K)}$$

The fact that we can possibly have negative rewards instead of in $[0, 1]$ like requested in the theorem, just changes that the loss estimator used as an intermediary is bounded by 2 instead of 1. Hence by selecting $\rho = \sqrt{\log(K)/TK}/m$, and using the importance weighted estimator $\hat{\varphi}(\lambda_t, c_t)$ we can use the EXP3 Theorem an apply the expectation to obtain the following regret bound:

$$\mathbb{E}[\max_{k \in [K]}\sum_{t=1}^{T}\mathbb{E}[\varphi(\lambda_t, c_{tk}, k) \mid \mathcal{H}_{t-1}]] - \mathbb{E}[\sum_{t=1}^{T}\varphi(\lambda_t, c_{tk_t}, k_t)] \leq 2m\sqrt{TK\log(K)}.$$

We know that any convex combination of the sum of the rewards of multiple sources is smaller than the sum of the rewards for the best source. Let us denote by $(\pi^n)_{n \in \mathbb{N}}$ the sequence of distributions that maximizes the dual objective of Lemma E.1. We will use $\pi^n$ as a convex combination, and taking the expectation over the regret bound we obtain:

$$\mathbb{E}[\sum_{t=1}^{T}\varphi(\lambda_t, c_{tk_t}, k_t)] \geq \mathbb{E}\left[\sum_{k=1}^{K}\pi_k^n\sum_{t=1}^{T}\mathbb{E}[\varphi(\lambda_t, c_{tk}, k) \mid \mathcal{H}_{t-1}]\right] - 2m\sqrt{TK\log(K)}.$$

Moreover using that $c_{tk}$ is independent of $\mathcal{H}_{t-1}$ and that $\lambda_t$ is $\mathcal{H}_{t-1}$ measurable we can deduce using the freezing Lemma that

$$\mathbb{E}[\varphi(\lambda_t, c_{tk}, k) \mid \mathcal{H}_{t-1}] + R^*(\lambda_t) = \mathbb{E}_{c_{tk}}[\varphi(\lambda_t, c_{tk}, k)] + R^*(\lambda_t) = \mathcal{D}(\lambda_t, k).$$

Hence

$$\mathbb{E}[\sum_{t=1}^{T}\varphi(\lambda_t, c_{tk_t}) + R^*(\lambda_t)] \geq \mathbb{E}[\sum_{k=1}^{K}\pi_k^n\sum_{t=1}^{T}\mathcal{D}(\lambda_t, k)] - 2m\sqrt{TK\log(K)}$$

$$= \mathbb{E}[\sum_{t=1}^{T}\langle\pi^n, \mathcal{D}(\lambda_t)\rangle] - 2m\sqrt{TK\log(K)}$$

$$\geq \sum_{t=1}^{T}\inf_{\lambda \in \mathbb{R}^d}\langle\pi^n, \mathcal{D}(\lambda)\rangle - 2m\sqrt{TK\log(K)}.$$

Taking the limit in $n \to \infty$ and by definition of $\pi^n$ we conclude that

$$\mathbb{E}[\sum_{t=1}^{T}\varphi(\lambda_t, c_{tk_t}) + R^*(\lambda_t)] \geq T\sup_{\pi \in \mathcal{P}_K}\inf_{\lambda \in \mathbb{R}^d}\langle\pi, \mathcal{D}(\lambda)\rangle - 2m\sqrt{TK\log(K)}.$$

Using the above regret bound and Equation (25) we obtain

$$\mathbb{E}\left[\sum_{t=1}^{T} f_t(x_t) - p_{k_t} - \bar{R}(\gamma_t)\right] \geq T\sup_{\pi \in \mathcal{P}_K}\inf_{\lambda \in \mathbb{R}^d}\langle\pi, \mathcal{D}(\lambda)\rangle - 2m\sqrt{TK\log(K)}$$

$$+ \mathbb{E}\left[\sum_{t=1}^{T}\langle\lambda_t, \delta_t - \gamma_t\rangle\right]. \tag{26}$$

**Analysis of Equation (23) and dual gradient descent**  We will now compare $TR(\sum_{t=1}^{T} \delta_t/T)$ and $\sum_{t=1}^{T} \bar{R}(\gamma_t)$. Here we consider a modified version of $\bar{R}$, which is equal to $\bar{R}$ over $\bar{\Delta}$ and is equal to $+\infty$ outside of $\bar{\Delta}$, and we will use its convex conjugate $\bar{R}^*$ (note that the convex conjugate of $\bar{R}$ without these modification would have been different !). Let

$$\hat{\lambda} \in \arg\max_{\lambda \in \mathbb{R}^d} \{\langle \lambda, \frac{1}{T} \sum_{t=1}^{T} \delta_t \rangle - \bar{R}^*(\lambda)\}.$$

We have by the Fenchel-Moreau theorem that

$$\bar{R}(\frac{1}{T} \sum_{t=1}^{T} \delta_t) = \bar{R}^{**}(\frac{1}{T} \sum_{t=1}^{T} \delta_t) = \langle \hat{\lambda}, \frac{1}{T} \sum_{t=1}^{T} \delta_t \rangle - \bar{R}^*(\hat{\lambda}).$$

By definition of the convex conjugate of $\bar{R}$, for any $\gamma \in \mathbb{R}^d$, thus in particular for all $\gamma_t$, we have

$$\bar{R}^*(\hat{\lambda}) \geq \langle \hat{\lambda}, \gamma_t \rangle - \bar{R}(\gamma_t).$$

Summing for every $t \in [T]$ we obtain

$$T\bar{R}^*(\hat{\lambda}) \geq \sum_{t=1}^{T} \langle \hat{\lambda}, \gamma_t \rangle - \bar{R}(\gamma_t)$$

Hence

$$\langle \hat{\lambda}, \sum_{t=1}^{T} \delta_t \rangle - T\bar{R}(\frac{1}{T} \sum_{t=1}^{T} \delta_t) \geq \sum_{t=1}^{T} \langle \hat{\lambda}, \gamma_t \rangle - \bar{R}(\gamma_t)$$

$$\Leftrightarrow \quad \langle \hat{\lambda}, \sum_{t=1}^{T} \delta_t - \gamma_t \rangle \geq T\bar{R}(\frac{1}{T} \sum_{t=1}^{T} \delta_t) - \sum_{t=1}^{T} \bar{R}(\gamma_t).$$

Finally because $\bar{R}$ and $R$ are equal over $\Delta$, and because $\sum_{t=1}^{T} \delta_t/T \in \Delta$, we derive the following inequality:

$$\sum_{t=1}^{T} \bar{R}(\gamma_t) - TR(\frac{1}{T} \sum_{t=1}^{T} \delta_t) \geq \sum_{t=1}^{T} \langle \hat{\lambda}, \gamma_t - \delta_t \rangle. \tag{27}$$

We now would like to track the left-hand sum evaluated at the dual parameter $\hat{\lambda}$. To do so we will use straightforward Online Gradient Descent on the linear function $\lambda \mapsto \langle \lambda, \gamma_t - \delta_t \rangle$, with sub-gradients $\gamma_t - \delta_t$.

We will use Lemma 6.17 from Orabona (2019):

**Theorem** (Online Gradient Descent). *For $g_t = \gamma_t - \delta_t$ a sub-gradient of a convex loss function $l_t$ with $\|g_t\|_2 \leq G$, for $w_t$ generated according to a gradient descent update with parameter $\eta$, we have that for any $u \in \mathbb{R}^d$:*

$$\sum_{t=1}^{T} l_t(w_t) - l_t(u) \leq \frac{\|u\|_2^2}{2\eta} + \frac{\eta}{2} TG^2.$$

This is exactly our setting, and the way we update $\lambda_t$. Therefore

$$\sum_{t=1}^{T} \langle \lambda_t, \gamma_t - \delta_t \rangle - \langle \hat{\lambda}, \gamma_t - \delta_t \rangle \leq \frac{\|\hat{\lambda}\|_2^2}{2\eta} + \frac{\eta}{2} TG^2.$$

Moreover, because $\hat{\lambda}$ is the $\arg\max$ which yields the convex conjugate of $\bar{R}^*$, we have $\hat{\lambda} \in \partial\bar{R}^{**}(\sum \delta_t/T) = \partial R^{**}(\sum \delta_t/T)$. Because $R$ is $L$-Lipschitz for $\|\cdot\|_2$, we have $\|\hat{\lambda}\|_2 \leq L$. Hence

$$\sum_{t=1}^{T} \langle \lambda_t, \gamma_t - \delta_t \rangle - \langle \hat{\lambda}, \gamma_t - \delta_t \rangle \leq \frac{L^2}{2\eta} + \frac{\eta}{2} TG^2.$$

What do we lose by using $\bar{R}$ instead of $R$ to generate $\gamma_t$ in our gradient descent ? We have larger gradients bounds by a factor 2 in our case. Indeed instead of having $G \leq \text{Diam}(\Delta)$, because $\gamma_t$ can be outside of $\Delta$, we have $\|\gamma_t - \delta_t\|_2 \leq 2\,\text{Diam}(\Delta)$. In this case, choosing $\eta = L/(2\,\text{Diam}(\Delta)\sqrt{T})$ is optimal, and yields the following regret bound:

$$\sum_{t=1}^{T}\langle \lambda_t, \gamma_t - \delta_t\rangle - \langle \hat{\lambda}, \gamma_t - \delta_t\rangle \leq 2L\,\text{Diam}(\Delta)\sqrt{T} \tag{28}$$

Using both Equation (27) and Equation (28) yield:

$$\sum_{t=1}^{T}\bar{R}(\gamma_t) - TR(\frac{1}{T}\sum_{t=1}^{T}\delta_t) \geq \sum_{t=1}^{T}\langle \lambda_t, \gamma_t - \delta_t\rangle - 2L\,\text{Diam}(\Delta)\sqrt{T} \tag{29}$$

We can finally put everything together. Through the decomposition with Equations (22) to (24), using respectively Equations (26) and (29) and Lemma 3.3 for each of those differences and summing the inequalities, the terms in $\langle \lambda_t, \gamma_t - \delta_t\rangle$ cancel out, and we obtain that

$$\text{ALG} \geq T\sup_{\pi \in \mathcal{P}_K}\inf_{\lambda \in \mathbb{R}^d}\langle \pi, \mathcal{D}(\lambda)\rangle - 2m\sqrt{TK\log(K)} - 2L\,\text{Diam}(\Delta)\sqrt{T} - 2L\sqrt{dT}.$$

By applying Lemma E.1 we conclude that

$$\text{Reg} = \text{OPT} - \text{ALG} \leq 2\big((L + \bar{f} + \max_{k}|p_k|)\sqrt{K\log(K)} + L\sqrt{d} + L\,\text{Diam}(\Delta)\big)\sqrt{T} + 2L\sqrt{K\log(K)}.$$

$\square$

# F    The Importance of Randomizing Between Sources: OPT vs static-OPT

## F.1    Proof of Proposition 2.1

We can directly derive from the proof of Lemma E.1 and Algorithm 1 the following proposition, which is a rewriting of Proposition 2.1.

**Proposition.** *There are instances of the problem such that*

$$\lim_{T\to\infty}\frac{\text{static}-\text{OPT}}{T} < \lim_{T\to\infty}\frac{\text{OPT}}{T}.$$

*Proof.* What we have shown indirectly in the proof of Theorem 3.1, is that

$$|T\max_{\pi \in \mathcal{P}_K}\inf_{\lambda \in \mathbb{R}^d}\langle \pi, \mathcal{D}(\lambda)\rangle - \text{OPT}| \leq \mathcal{O}(\sqrt{T}),$$

therefore $\lim_{T\to\infty}\text{OPT}/T = \lim_{T\to\infty}\max_{\pi \in \mathcal{P}_K}\inf_{\lambda \in \mathbb{R}^d}\langle \pi, \mathcal{D}(\lambda)\rangle/T$ which also gives us a simple way to compute a close upper bound for OPT as a saddle point, instead of having to solve a combinatorial optimization problem.

Following the same arguments as in Lemma E.1 besides the last maximization step, we can also derive that

$$\text{static}-\text{OPT} \leq T\max_{k\in[K]}\inf_{\lambda \in \mathbb{R}^d}\mathcal{D}(\lambda, k).$$

It remains to find an example such that

$$\max_{k\in[K]}\inf_{\lambda \in \mathbb{R}^d}\mathcal{D}(\lambda, k) < \max_{\pi \in \mathcal{P}_K}\inf_{\lambda \in \mathbb{R}^d}\langle \pi, \mathcal{D}(\lambda)\rangle.$$

We used such an example in Appendix F.2, where $\max_{k\in[K]}\inf_{\lambda \in \mathbb{R}^d}\mathcal{D}(\lambda, k) = 0$ and $\max_{\pi \in \mathcal{P}_K}\inf_{\lambda \in \mathbb{R}^d}\langle \pi, \mathcal{D}(\lambda)\rangle = 0.25$. $\square$

## F.2 Simple example and numerical illustration

We illustrate Proposition 2.1 and give an example that shows that randomizing over the different sources can outperform the best performance achievable by a single source.

We consider a case where the utility $u_t \in \{-1, 1\}$ and the protected attribute of a user $a_t \in \{-1, 1\}$. All combination of $u_t$ and $a_t$ are equiprobable: $\mathbb{P}(u_t = u, a_t = a) = 1/4$ for all $u, a \in \{-1, 1\}$. The penalty function is $R(\delta) = 5|\delta|$. There are $K = 2$ sources of information. The first source is capable of identifying good indivduals of group 1: it gives a context $c_{t1} = 1$ when $(a_t, u_t) = (1, 1)$ and $c_{t2} = 0$ otherwise. The second source does the same for good individuals of group $-1$: it gives $c_{t2} = 1$ when $(a_t, u_t) = (-1, 1)$ and $c_{t2} = 0$ otherwise. In terms of conditional expectations, this translates into $\mathbb{E}[u_t|c_{t1}{=}1] = \mathbb{E}[a_t|c_{t1}{=}1] = 1$, $\mathbb{E}[u_t|c_{t1}{=}0] = \mathbb{E}[a_t|c_{t1}{=}0] = -1/3$ for the first source; and $\mathbb{E}[u_t|c_{t2}{=}1] = 1$, $\mathbb{E}[a_t|c_{t2}{=}1] = -1$, $\mathbb{E}[u_t|c_{t2}{=}0] = -1/3$, $\mathbb{E}[a_t|c_{t2}{=}0] = 1/3$ for the second source.

We apply Algorithm 1 to this example, and we compare its performance to $(i)$ the one of an algorithm that has only access to the first source (which is the best source since both sources are symmetric), $(ii)$ a greedy algorithm that only use the first source and selects $x_t = 1$ whenever $\mathbb{E}[u_t \mid c_{t1}] > 0$, and $(iii)$ the offline optimal bounds OPT and static-OPT. The results are presented in Figure 1. We observe that our algorithm is close to OPT as shown in Theorem 3.1. It also vastly outperforms the algorithm that simply picks the best source, whose performance is close to static-OPT. The greedy has a largely negative utility because it is unfair. Our randomizing algorithm picks an allocation $x_t = 1$ only when the context implies that $u_t = 1$. It obtain a fair allocation by switching sources. An algorithm that uses a single source cannot be fair unless it chooses $x_t = 0$, which is why static-OPT=0 for this example. The $1^{st}$ and $3^{rd}$ quartiles are reported for multiple random seeds, the solid lines are the means. The computations were done on a laptop with an i7-10510U and 16 Gb of ram.

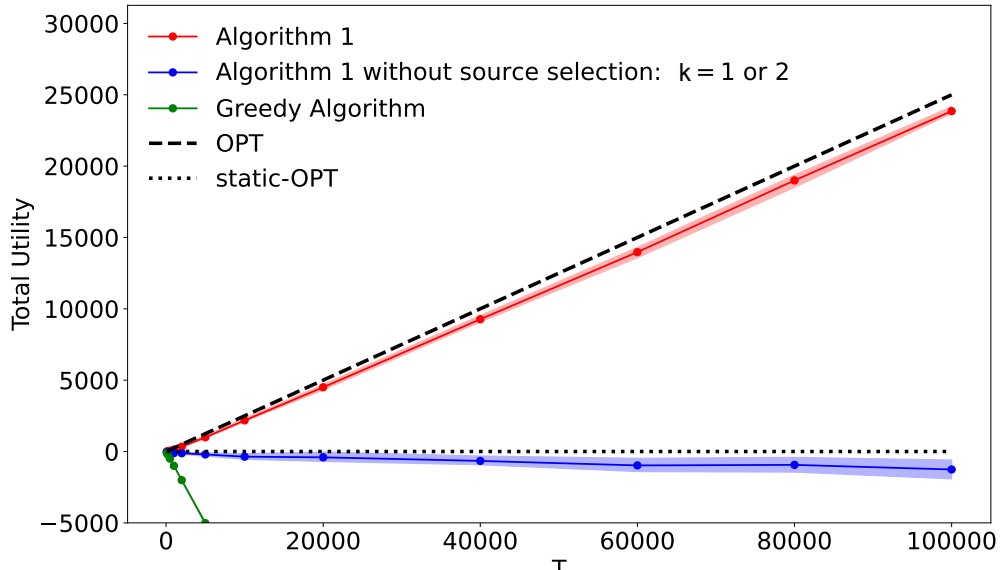

Figure 1: Static source vs randomization: The red curve is the total utility of Algorithm 1, The blue the utility of Algorithm 1 ran on the best-fixed source, and the green the utility of an unfair greedy algorithm. Black curves correspond to the upper bounds $T \lim_{T \to \infty}(\text{OPT}/T)$ and similarly for static-OPT.

## F.3 Additional experiments on the effect on OPT of sources and fairness prices

In the main body of the paper, we study how one is able to use the online algorithm Algorithm 1 so as to achieve a good total utility asymptotically close to that of the offline optimal OPT. However we only consider $R$ and the prices $p_k$ of the different sources as inputs for the algorithm, and we did not discuss how their variation actually affect the performance of $OPT$. While the previous illustrating example shows how a mixture of sources may achieve strictly better performance, we now only focus on the static optimization problem and how various metrics may change as we vary $R$ and $p_k$.

**Data model:** We consider a fairness penalty $R(x) = rx^2$ with $r \geq 0$, and $\mathcal{A} = \{-1, 1\}$ so that $a_t$ can only be either $-1$ or $1$, encoding men and women for instance. We suppose that the protected attributes and the utility $(u_t, a_t)$ are distributed according to $\Pr(a_t = 1) = \Pr(a_t = -1) = 1/2$ and $u_t \sim \mathcal{N}(a_t, 1)$, hence the group for which $a_t = 1$ is generally correlated to having better utilities than those for which $a_t = -1$. We suppose that there are no public context, and that there are $K = 2$ information sources, with $c_{t,1} = (a_t, u_t)$ for source 1 associated with a price of $p_1 = p$, and for source $k = 2$ we can observe $c_{t,2} = u_t$ with a price of 0. Here the sources are 'monotone' in that one contains strictly more information than the other, but is more expensive. Basically we are going to observe the utility $u_t$ in all cases, but before observing $u_t$ we can pay $p$ to additionally observe $a_t$. Note that when $r = 0$ then $R = 0$ and there is no fairness penalty meaning that we don't actually care about $a_t$ and will always select source $k = 2$, while when $p = 0$ the information about $a_t$ is free and there is no uncertainty.

We study the asymptotic regime with $T \to \infty$, and compute the optimal solution of the offline optimization problem through its dual problem. Figure 1 represent the quantities associated to the optimal solution for varying $p$ and $r$: (top-left) $\pi^*$ the percentage of times information is bought (i.e., $k = 1$ is chosen), (top-right) the probability of group 1 being selected $\Pr(a_t = 1 \mid x_t = 1)$, (bottom-left) the fairness cost relative to utility $R(\mathbb{E}[a_t x_t])/\mathbb{E}[u_t x_t]$, and (bottom-right) the information cost relative to utility $p\pi^*/\mathbb{E}[u_t x_t]$.

**Analysis and interpretation**:

- *(top-left)* Looking at the optimal action in terms of information (to buy the observation of $a_t$ or not), we see that there are three regimes: $\pi^* = 1$, $\pi^* = 0$ and a transitive regime $\pi^* \in (0, 1)$. When $r = 0$ there is no reason to care about $a_t$, thus we simply do not buy information ($\pi^* = 0$), this remains true as long as the cost of $p$ is large over $r$. When $p = 0$ there are no costs to observe $a_t$ and thus $\pi^* = 1$ and this remains true as long as the fairness penalty $r$ is large enough over $p$. In between, we may obtain optimal actions which are strictly between 0 and 1: in this case the best action is a strict convex combination between the two sources and there is some trade-off between buying costly information and selecting an unfair allocation.

- *(top-right)* Regarding the proportion of individuals of group 1 selected, we see that it is always higher than 0.5 as group $a_t = 1$ correlates to higher utility compared to group $a_t = -1$. Therefore it is decreasing in $r$ and goes towards the perfectly fair allocation 0.5 which would incur no penalty even for high $r$. It is also increasing in $p$ as higher information costs make it less desirable to pay to observe $a_t$, and thus more difficult to accurately achieve a fair allocation: the support of the utility $u_t$ conditioned on $a_t = 1$ and conditioned on $a_t = -1$ are not disjoint, which makes it difficult only by observing $u_t$ to determine whether $a_t = 1$ or $-1$.

- *(bottom-left)* For the relative fairness cost compared to utility, when the information cost $p$ increases the allocation becomes more unfair as it becomes more difficult without access to $a_t$ to choose a fair allocation with good utility as describe above. Moreover the average utility $u_t$ of the selected groups tend to decrease as the individuals which are more likely to be $a_t = -1$ based on $u_t$ have low utility. Therefore the ratio fairness penalty over expected utility tend to increase. When $r$ increases there are three conflicting effects: high $r$ makes it more appealing to achieve a fair allocation at the cost of expected utility $u_t$ which makes the latter decrease, the expected group allocation becomes more fair as seen previously, and the scaling of $R$ increases. When $r$ becomes high there is some balance between a fairer allocation and higher penalty, and when $r = 0$ the fairness penalty is clearly 0. Overall for each fixed $p$ there is some maximum in terms of fairness penalty relative to utility.

- *(bottom-right)* Finally for the information cost relative to utility, it is increasing in $r$. Indeed due to large fairness penalties it is better to buy information thus $\pi^*$ increases while $p$ remains fixed, hence the product $p\pi^*$ increases. When $p$ increases, we buy information less often ($\pi^*$ decreases), and because $p\pi^* = 0$ for $p = 0$ or $p$ large, there is some maximum for each $r$ fixed.

Overall, we see that higher fairness penalties tend to make it more likely to buy information and select a fair allocation even at the cost of utility, while high information cost makes it unattractive to buy this information and can lead to unfairness due to the difficulty of identifying the protected attribute without this additional data.

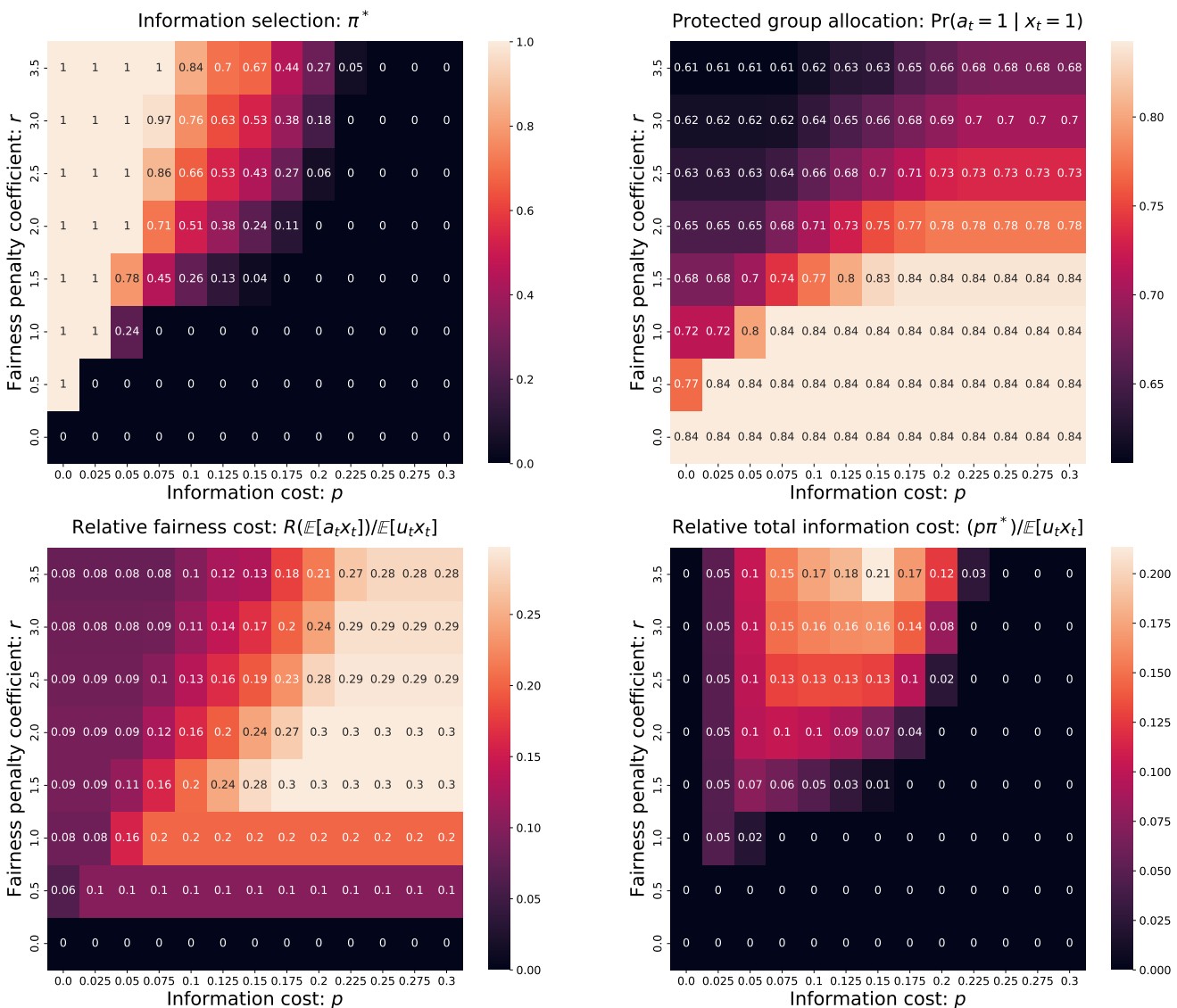

Figure 2: For varying fairness scaling $r$ and information cost $p$: (top-left) optimal information purchase frequency $\pi^*$, (top-right) optimal probability of group 1 selection $\Pr(a_t = 1 \mid x_t = 1)$, (bottom-left) optimal fairness cost relative to utility $R(\mathbb{E}[a_t x_t])/\mathbb{E}[u_t x_t]$, and (bottom-right) optimal information cost relative to utility $p\pi^*/\mathbb{E}[u_t x_t]$

Here we looked at a simple example to try to isolate how sensible is the optimal solution to some variations on $r$ and $p$. Other parameters can lead to unfairness, such as an unbalance in the protected group population where one would still need a balanced allocation, or penalty functions with varying convexity strength.

## G   Other Fairness Penalty — proof of Theorem 4.1

In this section, we consider an other type of fairness penalty as described in Section 4.1.

We redefine $\mathcal{U}(.)$ in the following way:

$$\mathcal{U}(\boldsymbol{k}, \boldsymbol{x}) = \sum_{t=1}^{T} (f_t(x_t) - p_{k_t}) - \Big(\sum_{t=1}^{T} x_t\Big) R\left(\frac{\sum_{t=1}^{T} a_t(x_t)}{\sum_{t=1}^{T} x_t}\right). \tag{30}$$

When $\mathcal{X} \subset \mathbb{R}_+^n$ with $n > 1$, there are two ways to intuitively generalize this penalty for higher dimension allocations. We can replace $\sum_{t=1}^{T} x_t$ by $\sum_{t=1}^{T} \sum_{i=1}^{n} x_{ti}$, which is still a scalar and the proof follows directly. For instance, this corresponds for online bipartite matching with node arrival to a fairness condition on all online arriving nodes (it does not matter to which offline nodes they are matched). Otherwise we can instantiate $n$ penalties $R_1, \ldots, R_n$, using $\sum_t x_{ti}$ for each penalty $R_i$ with $i \in [n]$. This corresponds to a fairness condition on the selected online nodes for each of the $n$ offline node. This can be dealt with using one parameter $\lambda_i$ for each penalty $R_i$. For this proof we will assume that $n = 1$ without loss of generality.

We require the following technical assumption: $\sum_{\tau=1}^{t} a_\tau(x_\tau) / \sum_{\tau=1}^{t} x_\tau \in \Delta$ for all $x_\tau$, $a_\tau$ and $t$. Of course, the size of $\Delta$ can be increased to guarantee — if possible — that this quantity stays inside $\Delta$, but with the trade-off that $\mathrm{Diam}(\Delta)$ increases. Clearly, this assumption is verified for the special case of $a_t(x_t) = a x_t$ for some $a \in \mathcal{A}$, as the $x_\tau / \sum_t x_t$ form a convex combination.

Let us redefine the virtual reward associated to this other fairness penalty:

$$\tilde{\varphi}(\lambda, c_{tk}, k) = \max_{x \in \mathcal{X}} \mathbb{E}\left([f_t(x) - \langle \lambda_t, a_t(x) \rangle \mid c_{tk}] + x R^*(\lambda_t)\right) - p_k.$$

We also redefine the unbiased estimator $\hat{\varphi}$ with this new $\tilde{\varphi}$.

We now use Algorithm 3. There are 4 differences with Algorithm 2: the new function $\varphi$ contains in addition $x R^*(\lambda)$, $x_t$ is the $\arg\max$ with respect to this new virtual reward, $\gamma_t$ is the $\arg\max$ over $\Delta_{\delta_t/x_t}$ instead of $\Delta_{\delta_t}$, and finally the update for $\lambda_{t+1}$ uses $x_t \gamma_t$ instead of $\gamma_t$.

---

**Algorithm 3** Online Fair Allocation with Source Selection — Other fairness penalty

---

**Input:** Initial dual parameter $\lambda_0$, initial source-selection-distribution $\pi_0 = (1/K, \ldots, 1/K)$, dual gradient descent step size $\eta$, EXP3 step size $\rho$, cumulative estimated rewards $S_0 = 0 \in \mathbb{R}^K$.
**for** $t \in [T]$ **do**

Draw a source $k_t \sim \pi_t$, where $(\pi_t)_k \propto \exp(\rho S_{(t-1)k})$ and observe $c_{tk_t}$.
Compute the allocation for user $t$:

$$x_t = \arg\max_{x \in \mathcal{X}} \left(\mathbb{E}[f_t(x) - \langle \lambda_t, a_t(x) \rangle \mid c_{tk_t}] + x R^*(\lambda_t)\right). \tag{31}$$

Update the estimated rewards sum and sources distributions for all $k \in [K]$:

$$S_{tk} = S_{t-1,k} + \hat{\varphi}(\lambda_t, c_{tk}, k_t), \tag{32}$$

Compute the expected protected group allocation $\delta_t = \mathbb{E}[a_t(x_t) \mid c_{tk_t}, x_t]$ and compute the dual protected groups allocation target and update the dual parameter:

$$\gamma_t = \arg\max_{\gamma \in \Delta_{\delta_t/x_t}} \{\langle \lambda_t, \gamma \rangle - \bar{R}(\gamma)\},$$
$$\lambda_{t+1} = \lambda_t - x_t \eta (\gamma_t - \delta_t/x_t).$$

**end for**

---

Let us first prove an upper bound on OPT, which actually guides the design of the modified virtual reward $\tilde{\varphi}$:

**Lemma G.1.** *We have the following upper-bound for the offline optimum:*

$$\tilde{\mathrm{OPT}} \leq T \sup_{\pi \in \mathcal{P}_K} \inf_{\lambda \in \mathbb{R}^d} \langle \pi, \tilde{\mathcal{D}}(\lambda) \rangle,$$

*where $\tilde{\mathcal{D}}(\lambda) = (\tilde{\mathcal{D}}(\lambda, 1), \ldots, \tilde{\mathcal{D}}(\lambda, K))$ is a vector representing the value of the dual conjugate problem, with for $k \in [K]$ the coordinates*

$$\tilde{\mathcal{D}}(\lambda, k) = \mathbb{E}_{c_{tk}}[\tilde{\varphi}(\lambda, c_{tk}, k)].$$

*Proof.* The idea will be the same as in Lemma E.1, we only use a different lagrangian function.

We recall that the $x_t$ are $\sigma(c_{1k_1}, \ldots, c_{tk_t})$ measurable (because the $k_t$ are deterministic). Let $\delta_t = \mathbb{E}_{a_t}[a_t(x_t) \mid c_{tk}]$, and $\tilde{f}_t(x) = \mathbb{E}_{f_t}[a_t(x_t) \mid c_{tk}]$. Using Jensen's inequality for $R$ convex we have

$$
\begin{aligned}
\mathbb{E}[\mathcal{U}(\boldsymbol{k}, \boldsymbol{x}) \mid c_{1k_1}, \ldots, c_{Tk_T}] &= \sum_{t=1}^T \mathbb{E}[f_t(x_t) - p_{k_t} \mid c_{1k_1}, \ldots, c_{Tk_T}] - \mathbb{E}[(\sum_{t=1}^T x_t) R\left(\frac{\sum_{t=1}^T a_t(x_t)}{\sum_{t=1}^T x_t}\right) \mid c_{1k_1}, \ldots, c_{Tk_T}] \\
&= \sum_{t=1}^T \mathbb{E}[f_t(x_t) - p_{k_t} \mid c_{1k_1}, \ldots, c_{Tk_T}] - (\sum_{t=1}^T x_t) \mathbb{E}[R\left(\frac{\sum_{t=1}^T a_t(x_t)}{\sum_{t=1}^T x_t}\right) \mid c_{1k_1}, \ldots, c_{Tk_T}] \\
&\leq \sum_{t=1}^T \mathbb{E}[f_t(x_t) - p_{k_t} \mid c_{1k_1}, \ldots, c_{Tk_T}] - (\sum_{t=1}^T x_t) R\left(\mathbb{E}[\frac{\sum_{t=1}^T a_t(x_t)}{\sum_{t=1}^T x_t} \mid c_{1k_1}, \ldots, c_{Tk_T}]\right) \\
&= \sum_{t=1}^T \mathbb{E}[f_t(x_t) - p_{k_t} \mid c_{1k_1}, \ldots, c_{Tk_T}] - (\sum_{t=1}^T x_t) R\left(\frac{\mathbb{E}[\sum_{t=1}^T a_t(x_t) \mid c_{1k_1}, \ldots, c_{Tk_T}]}{\sum_{t=1}^T x_t}\right).
\end{aligned}
$$

Moreover by independence of the $(f_t, a_t, c_{t1}, \ldots, c_{tK})$ we have that $\mathbb{E}[a_t(x_t) \mid \sigma(c_{1k_1}, \ldots, c_{tk_t})] = \mathbb{E}_{a_t}[a_t(x_t) \mid c_{tk_t}] = \delta_t$. This is basically an application of Lemma D.3 where we state it more carefully. With the same argument we obtain that $\mathbb{E}[f_t(x_t) \mid \sigma(c_{1k_1}, \ldots, c_{Tk_T})] = \tilde{f}_t(x_t)$. Therefore

$$
\mathbb{E}[\mathcal{U}(\boldsymbol{k}, \boldsymbol{x}) \mid c_{1k_1}, \ldots, c_{Tk_T}] \leq \sum_{t=1}^T \tilde{f}_t(x_t) - p_{k_t} - (\sum_{t=1}^T x_t) R\left(\frac{\sum_{t=1}^T \delta_t}{\sum_{t=1}^T x_t}\right). \tag{33}
$$

We define the function $\mathcal{L} : (\boldsymbol{k}, \mathbf{c}, \mathbf{x}, \lambda) \mapsto \sum_{t=1}^T \tilde{f}_t(x_t) - p_{k_t} - x_t \langle \lambda, \tilde{a}_t \rangle + x_t R^*(\lambda)$.

Using the Fenchel-Moreau theorem, and the previous inequality we have

$$
\begin{aligned}
\mathcal{L}(\boldsymbol{k}, \mathbf{c}, \mathbf{x}, \lambda) &\geq \inf_{\lambda \in \mathbb{R}^d} \mathcal{L}(\mathbf{c}, \mathbf{x}, \lambda) \\
&= \sum_{t=1}^T \tilde{f}_t(x_t) - p_{k_t} - \sup_{\lambda \in \mathbb{R}^d} \{\langle \lambda, \sum_{t=1}^T \tilde{a}_t x_t \rangle - (\sum_{t=1}^T x_t) R^*(\lambda)\} \\
&= \sum_{t=1}^T \tilde{f}_t(x_t) - p_{k_t} - (\sum_{t=1}^T x_t) \sup_{\lambda \in \mathbb{R}^d} \{\langle \lambda, \frac{\sum_{t=1}^T \delta_t}{\sum_{t=1}^T x_t} \rangle - R^*(\lambda)\} \\
&= \sum_{t=1}^T \tilde{f}_t(x_t) - p_{k_t} - (\sum_{t=1}^T x_t) R^{**}\left(\frac{\sum_{t=1}^T \delta_t}{\sum_{t=1}^T x_t}\right) \\
&= \sum_{t=1}^T \tilde{f}_t(x_t) - p_{k_t} - (\sum_{t=1}^T x_t) R\left(\frac{\sum_{t=1}^T \delta_t}{\sum_{t=1}^T x_t}\right) \\
&\geq \mathbb{E}[\mathcal{U}(\boldsymbol{k}, \boldsymbol{x}) \mid \mathbf{c_k}]. \tag{34}
\end{aligned}
$$

We can then finish as we did for Lemma E.1. $\qquad\square$

We now proceed to the rest of the proof of Theorem 4.1:

**Theorem.** *For $\eta = L/(2 \operatorname{Diam}(\Delta)\sqrt{T})$, $m = \bar{f} + L + \max_k |p_k| + 2\eta \operatorname{Diam}(\Delta) + m^*$ ($m^*$ is a constant defined in the proof and depends on $\Delta$ and $R$), $\rho = \sqrt{\log(K)/(TKm^2)}$, and $\lambda_0 \in \partial R(a)$ for some $a \in \mathcal{A}$, algorithm Algorithm 1 modified as suggested in Section 4.1 has the following regret upper bound:*

$$
\operatorname{Reg} \leq 2((L + \bar{f} + \max_k |p_k| + m^*)\sqrt{K \log(K)} + L\sqrt{d} + \operatorname{Diam}(\Delta))\sqrt{T} + 2L\sqrt{K \log(K)}
$$

*Proof.* We present here only the parts that are different from the proof of Theorem 3.1, and we refer to the corresponding proof for more details.

**Comparison of performance with $\tilde{\varphi}$** By adding and removing $-x_t\langle\lambda_t, \tilde{a}_t\rangle + x_t R^*(\lambda_t)$, and using that $x_t$ is the maximizer that yields $\varphi$ we derive that

$$\mathbb{E}[f_t(x_t)] = \mathbb{E}[\mathbb{E}_{f_t}[f_t(x_t) \mid c_{tk_t}] - \langle\lambda_t, \delta_t\rangle + \langle\lambda_t, \delta_t\rangle + x_t R^*(\lambda_t) - x_t R^*(\lambda_t)]$$
$$= \mathbb{E}[\varphi(\lambda_t, c_{tk_t}) + \langle\lambda_t, \delta_t\rangle - x_t R^*(\lambda_t)].$$

Because $\Delta \subset \Delta_{\delta_t}$ and, because $\bar{R}$ and $R$ are equal over $\Delta$ we have:

$$R^*(\lambda_t) = \max_{\gamma\in\Delta}\{\langle\lambda_t, \gamma\rangle - R(\gamma)\}$$
$$= \max_{\gamma\in\Delta}\{\langle\lambda_t, \gamma\rangle - \bar{R}(\gamma)\}$$
$$\leq \max_{\gamma\in\Delta_{\tilde{a}_t}}\{\langle\lambda_t, \gamma\rangle - \bar{R}(\gamma)\}$$
$$= \langle\lambda_t, \gamma_t\rangle - \bar{R}(\gamma_t).$$

Thus

$$\mathbb{E}[f_t(x_t) - p_{k_t} - x_t\bar{R}(\gamma_t)] \geq \mathbb{E}[\tilde{\varphi}(\lambda_t, c_{tk_t}) + \langle\lambda_t, \delta_t - x_t\gamma_t\rangle]. \tag{35}$$

**Application of bandit algorithms** Compared to the previous setting, the only change is that the function $R^*$ is included in $\varphi$, so we need to bound it as well. See Lemma D.5. This is where $m^*$ comes from.

**Dual gradient descent** Let us now focus in the tracking of the fairness parameter. In the evaluation of our lower bound of the algorithm performance, we used $-x_t\bar{R}(\gamma_t)$ instead of the true penalty $(\sum_{t=1}^T x_t)R(\sum_{t=1}^T a_t(x_t)/\sum_{t=1}^T x_t)$. Because of a small modification of Lemma 3.3 we only need to compare it with the penalty over the conditional expectation of $a_t x_t$. Indeed, by the Lipschitz property of $R$, whether we re-scale by $T$ or $\sum x_t$ does not change anything.

Hence will now compare those two penalties. Here we will consider a modified version of $\bar{R}$, which is equal to $\bar{R}$ over $\bar{\Delta}$ and is equal to $+\infty$ outside of $\bar{\Delta}$, and we will use its convex conjugate $\bar{R}^*$ (note that the convex conjugate of $\bar{R}$ without the modification would have been different !). Let

$$\hat{\lambda} \in \arg\max_{\lambda\in\mathbb{R}^d}\{\langle\lambda, \frac{\sum_{t=1}^T \delta_t}{\sum_{t=1}^T x_t}\rangle - \bar{R}^*(\lambda)\}.$$

We have by Fenchel-Moreau theorem that

$$\langle\hat{\lambda}, \frac{\sum_{t=1}^T \delta_t}{\sum_{t=1}^T x_t}\rangle - \bar{R}^*(\hat{\lambda}) = \bar{R}^{**}\left(\frac{\sum_{t=1}^T \delta_t}{\sum_{t=1}^T x_t}\right)$$
$$= \bar{R}\left(\frac{\sum_{t=1}^T \delta_t}{\sum_{t=1}^T x_t}\right).$$

Hence

$$\langle\hat{\lambda}, \sum_{t=1}^T \delta_t\rangle - (\sum_{t=1}^T x_t)\bar{R}\left(\frac{\sum_{t=1}^T \delta_t}{\sum_{t=1}^T x_t}\right) = (\sum_{t=1}^T x_t)\bar{R}^*(\hat{\lambda}).$$

By definition of the convex conjugate of $\bar{R}$, for any $\gamma\in\mathbb{R}^d$, thus in particular for all $\gamma_t$, we have

$$\bar{R}^*(\hat{\lambda}) \geq \langle\hat{\lambda}, \gamma_t\rangle - \bar{R}(\gamma_t).$$

Multiplying by $x_t$ and summing for every $t\in[T]$ we obtain

$$(\sum_{t=1}^T x_t)\bar{R}^*(\hat{\lambda}) \geq \sum_{t=1}^T \langle\hat{\lambda}, x_t\gamma_t\rangle - x_t\bar{R}(\gamma_t)$$

Hence

$$\langle\hat{\lambda}, \sum_{t=1}^T \delta_t\rangle - (\sum_{t=1}^T x_t)\bar{R}\left(\frac{\sum_{t=1}^T \delta_t}{\sum_{t=1}^T x_t}\right) \geq \sum_{t=1}^T \langle\hat{\lambda}, \gamma_t x_t\rangle - \bar{R}(\gamma_t)$$
$$\Leftrightarrow \quad \langle\hat{\lambda}, \sum_{t=1}^T \delta_t - x_t\gamma_t\rangle \geq (\sum_{t=1}^T x_t)\bar{R}\left(\frac{\sum_{t=1}^T \delta_t}{\sum_{t=1}^T x_t}\right) - \sum_{t=1}^T x_t\bar{R}(\gamma_t).$$

Finally because $\bar{R}$ and $R$ are equal over $\Delta$, and because $\sum_{t=1}^{T} \delta_t / \sum_{t=1}^{T} x_t \in \Delta$ by the assumption, we have that

$$\langle \hat{\lambda}, \sum_{t=1}^{T} \delta_t - x_t \gamma_t \rangle \geq (\sum_{t=1}^{T} x_t) R \left( \frac{\sum_{t=1}^{T} \delta_t}{\sum_{t=1}^{T} x_t} \right) - \sum_{t=1}^{T} x_t \bar{R}(\gamma_t) \tag{36}$$

As we did earlier, we apply Online Gradient Descent to track the term evaluated at $\hat{\lambda}$ and obtain Equation (28). We just need to make sure that as before $\hat{\lambda}$ is bounded. By the definition of $\hat{\lambda}$:

$$\hat{\lambda} \in \partial \bar{R}^{**}(\frac{\sum_{t=1}^{T} \delta_t}{\sum_{t=1}^{T} x_t}) = \partial \bar{R}(\frac{\sum_{t=1}^{T} \delta_t}{\sum_{t=1}^{T} x_t}) = \partial R(\frac{\sum_{t=1}^{T} \delta_t}{\sum_{t=1}^{T} x_t}).$$

Because $R$ is $L$-Lipschitz for $\| \cdot \|_2$, we have $\|\hat{\lambda}\|_2 \leq L$.

The rest of the proof is identical.

$\square$

# H    Learning $u_t$ — proof of Proposition 4.2

We prove in this section the result regarding the added regret of learning $u_t$. In terms of utility functions $f_t$, protected attributes $a_t$, and allocations $x_t$, we go back to the setting of Section 2.1 with $f_t(x) = u_t x$ and $\mathcal{X} = \{0, 1\}$.

In addition to the assumptions made in Section 4.2, we give more precision regarding the structure of $u_t$. We define $\theta_{tk} = \mathbb{1}[k_t = k]$ the indicator variable of selecting source $k$ at time $t$. We define the following filtration for all $k \in [K]$ and all $t \in [T]$:

$$\mathcal{F}_{tk} = \sigma(x_1, c_{1k}, \theta_{1k}, u_1, \ldots, x_t, c_{tk}, \theta_{tk}, u_t, x_{t+1}, c_{t+1,k}, \theta_{t+1,k}).$$

We assume that the contexts $c_{tk}$ are now feature vectors of dimension $q_k$, and that for all $k \in [K]$, there exists some vector $\psi_k \in \mathbb{R}^{q_k}$ so that

$$\eta_{tk} = u_t - \langle \psi_k, c_{tk} \rangle, \tag{37}$$

is a zero mean 1-subgaussian random variable conditionally on $\mathcal{F}_{t-1,k}$.

Consider that each source $k$ corresponds to one bandit problem, with an action set at time $t$ composed of two contextual actions $\{c_{tk}\theta_{tk}x, x \in \mathcal{X}\} = \{c_{tk}\theta_{tk}, 0\}$, and a reward $u_t x_t \theta_{tk}$. If $\theta_{tk} = 0$ then the action selected is 0, and the new reward is 0, which means both are "observed". If $\theta_{tk} = 1$, then the feedback is $u_t$ if $x_t = 1$, and 0 otherwise. Regardless of the source selected, the feedback received and the action producing this feedback are observed for all arms in parallel. This is akin to artificially playing all the $[K]$ bandits in simultaneously. This new reward still follows the conditional linearity assumption, and $x_t \theta_{tk} \eta_t$ is still $\mathcal{F}_{t-1,k}$ conditionally 1-subgaussian.

With these rewards and actions, we can define for each source $\hat{\psi}_{tk}$ and $V_{tk}$ the least squares estimator and the design matrix as done in Equations (19.5) and (19.6) of Lattimore & Szepesvári (2020). Let $\delta \in (0, 1)$. We define the confidence set parameter $\beta_t$ as

$$\sqrt{\beta_t} = \bar{\psi} + \sqrt{2 \log \left( \frac{1}{\delta} \right) + \max_{k \in [k]} q_k \log \left( 1 + \frac{t \bar{c}^2}{\max_{k \in [k]} q_k} \right)}. \tag{38}$$

This parameter is independent of $k$, this will make the computations easier later on. We now define the ellipsoid confidence set of $\psi_k$ as

$$\mathcal{I}_{tk} = \{\psi \in \mathbb{R}^{q_k}, \|\psi - \hat{\psi}_{tk}\|_{V_{t-1,k}}^2 \leq \beta_t\}. \tag{39}$$

Algorithm 1 is modified in the following way. For each time $t$, let $\tilde{\psi}_{tk} = \arg\max_{\psi \in \mathcal{I}_{tk}} \langle \psi, c_{tk} \rangle$. We now select $x_t$ using this optimistic estimate of $\psi_k$:

$$x_t = \arg\max_{x \in \{0,1\}} \{x(\langle \tilde{\psi}_{tk}, c_{tk} \rangle - \langle \lambda_t, \tilde{a}_t \rangle)\}. \tag{40}$$

**Proposition.** *Given the above assumptions, the* added *regret of having to learn* $\mathbb{E}[u_t \mid c_{tk}]$ *is of order* $\mathcal{O}(\sqrt{KT}\log(T))$.

The proof will proceed as follows: we first highlight where the learning will impact the algorithm, leverage the concentration results from Abbasi-yadkori et al. (2011) to decompose into a good event and bad event, and finally bound the loss over the good event.

Let us now analyze the main part of the proof in Appendix E that is modified, which is "Comparison of performance with $\varphi$", because the only change is the quantity that $x_t$ maximizes. Let $x_t^*$ be the maximizer obtained by replacing the optimistic estimate with the true parameter $\psi_k$. Let $\tilde{a}_t = \mathbb{E}[a_t \mid c_{tk_t}]$. Because $x_t$ maximizes the quantity with the optimistic estimate we have

$$
\begin{aligned}
x_t \mathbb{E}[u_t \mid c_{tk_t}] - x_t \langle \lambda_t, \tilde{a}_t \rangle &= x_t \langle \psi_{k_t}, c_{tk_t} \rangle - x_t \langle \lambda_t, \tilde{a}_t \rangle \\
&= x_t (\langle \psi_{k_t}, c_{tk_t} \rangle - \langle \tilde{\psi}_{tk_t}, c_{tk_t} \rangle) + x_t \langle \tilde{\psi}_{tk_t}, c_{tk_t} \rangle - x_t \langle \lambda_t, \tilde{a}_t \rangle \\
&\geq x_t \langle \psi_{k_t} - \tilde{\psi}_{tk_t}, c_{tk_t} \rangle + x_t^* \langle \tilde{\psi}_{tk_t}, c_{tk_t} \rangle - x_t^* \langle \lambda_t, \tilde{a}_t \rangle \\
&= x_t \langle \psi_{k_t} - \tilde{\psi}_{tk_t}, c_{tk_t} \rangle + x_t^* \langle \tilde{\psi}_{tk_t} - \psi_{k_t}, c_{tk_t} \rangle + x_t^* \langle \psi_{k_t}, c_{tk_t} \rangle - x_t^* \langle \lambda_t, \tilde{a}_t \rangle \\
&= x_t \langle \psi_{k_t} - \tilde{\psi}_{tk_t}, c_{tk_t} \rangle + x_t^* \langle \tilde{\psi}_{tk_t} - \psi_{k_t}, c_{tk_t} \rangle + \varphi(\lambda_t, c_{tk_t}, k_t) + p_{k_t}.
\end{aligned}
$$

The last term $\varphi$ is the one we want, we now need to lower bound the first two terms.

By Theorem 20.5 of Lattimore & Szepesvári (2020), we know with probability at least $1 - \delta$ that $\psi_k \in \mathcal{I}_{tk}$ for all $t$. By union bound, the probability that the $\psi_k$ are simultaneously all in their confidence set is at least $1 - K\delta$. We denote this event by $I$.

Suppose that the event $I$ is satisfied. Then by definition of $\tilde{\psi}_{tk_t}$, we have

$$
x_t^* \langle \tilde{\psi}_{tk_t} - \psi_{k_t}, c_{tk_t} \rangle \geq 0.
$$

Now let us look at the last term. We can first decompose it over the $K$ sources:

$$
x_t \langle \psi_{k_t} - \tilde{\psi}_{tk_t}, c_{tk_t} \rangle = \sum_{k=1}^{K} \theta_{tk} x_t \langle \psi_k - \tilde{\psi}_{tk}, c_{tk} \rangle = \sum_{k=1}^{K} \langle \psi_k - \tilde{\psi}_{tk}, \theta_{tk} x_t c_{tk} \rangle.
$$

Now, consider one source $k$, and sum the difference term for every $t$. By remarking that $\theta_{tk} = \theta_{tk}^2$, we have that

$$
\begin{aligned}
\sum_{t=1}^{T} |\langle \psi_k - \tilde{\psi}_{tk} \mid \theta_{tk} x_t c_{tk} \rangle| &= \sum_{t=1}^{T} \theta_{tk} |\langle \psi_k - \tilde{\psi}_{tk} \mid \theta_{tk} x_t c_{tk} \rangle| \\
&\leq \sum_{t=1}^{T} \theta_{tk} \|\psi_k - \tilde{\psi}_{tk}\|_{V_{t-1,k}} \|\theta_{tk} x_t c_{tk}\|_{V_{t-1,k}^{-1}} \quad \text{(dual-norm inequality)} \\
&\leq 2 \sum_{t=1}^{T} \theta_{tk} \sqrt{\beta_t} \|\theta_{tk} x_t c_{tk}\|_{V_{t-1,k}^{-1}} \quad \text{(by the event } I\text{)} \\
&\leq 2 \sum_{t=1}^{T} \theta_{tk} \sqrt{\beta_T} \max\{1, \|\theta_{tk} x_t c_{tk}\|_{V_{t-1,k}^{-1}}\} \quad (\beta_t \text{ are increasing}) \\
&\leq 2 \sqrt{\beta_T} \sqrt{\sum_{t=1}^{T} \theta_{tk}^2} \sqrt{\sum_{t=1}^{T} \max\{1, \|\theta_{tk} x_t c_{tk}\|_{V_{t-1,k}^{-1}}^2\}} \quad \text{(C.S. inequality)} \\
&= 2 \sqrt{\beta_T} \sqrt{\sum_{t=1}^{T} \theta_{tk}} \sqrt{\sum_{t=1}^{T} \max\{1, \|\theta_{tk} x_t c_{tk}\|_{V_{t-1,k}^{-1}}^2\}} \\
&= 2 \sqrt{\beta_T} \sqrt{T_k} \sqrt{\sum_{t=1}^{T} \max\{1, \|\theta_{tk} x_t c_{tk}\|_{V_{t-1,k}^{-1}}^2\}},
\end{aligned}
$$

where $T_k$ is the number of times source $k$ is selected.

By Lemma 19.4 of Lattimore & Szepesvári (2020), we have that

$$\sum_{t=1}^{T} \max\{1, \|\theta_{tk} x_t c_{tk}\|_{V_{t-1,k}^{-1}}^2\} \leq 2q_k \log\left(1 + \frac{T\bar{c}^2}{q_k}\right).$$

Therefore, using Jensen inequality for the square root function which is concave over the $T_k$, and because $\sum_k T_k = T$, we obtain

$$\sum_{t=1}^{T} \sum_{k=1}^{K} |\langle \psi_k - \tilde{\psi}_{tk} \mid \theta_{tk} x_t c_{tk}\rangle| \leq \sum_{k=1}^{K} 2\sqrt{\beta_T}\sqrt{T_k}\sqrt{2q_k \log\left(1 + \frac{T\bar{c}^2}{q_k}\right)}$$

$$\leq 2\sqrt{\beta_T}\sum_{k=1}^{K} \sqrt{T_k}\sqrt{2q_k \log\left(1 + \frac{T\bar{c}^2}{q_k}\right)}$$

$$\leq \sqrt{8T\beta_T}\sqrt{\sum_{k=1}^{K} q_k \log\left(1 + \frac{T\bar{c}^2}{q_k}\right)}.$$

This yields a $\mathcal{O}(\sqrt{KT}\log(T))$ error.

And with probability at most $K\delta$ the concentration event $I$ does not hold, but the error is bounded by 4 for each $t$. Thus overall in expectation, it is bounded by $4TK\delta$. Picking $\delta = 1/T$, yields the desired sub-linear regret bound.

# I  Learning $a_t$

In this section, we provide further intuitions on the existence of $[K]$ different sources, and how it is possible to adapt the algorithm to learn $\mathbb{E}[a_t \mid c_{tk}]$ on the fly. Let us consider the special case of our model, where the additional information given by each source is a noisy observation $\hat{a}_{tk}$ of $a_t$, with different noise levels; in the main setting this can be described as $c_{tk} = (z_t, \hat{a}_{tk})$. Sources with lower noise levels will have higher prices (one directly pays for the level of accuracy). This setting can be used to give users the level of privacy they are willing to give up (as in Local Differential Privacy (Dwork & Roth, 2014)), by revealing some noisy information about their private data; of course, the more data is revealed (or the smaller the noise), the higher the compensation (the price).

Furthermore, if the source $k$ corresponds to adding additive independent noise $L_{tk}$ and to returning $\hat{a}_{tk} = a_t + L_{tk}$, then knowing the law of both $L_{tk}$ and $a_t$, the decision maker can directly compute the conditional expectation of $a_t$ having observed $\hat{a}_{tk}$. Nevertheless, if she only knows the noise distribution, but not the law of $a_t$, conditional expectation can still be learned in an online fashion in some cases.

Indeed, suppose for simplicity that $a_t \in \{-1, 1\}$, $\Pr(a_t = 1) = \alpha \in (0, 1)$, $(z_t, u_t)$ is independent of $a_t$ (therefore we can focus only in dealing with $a_t$), and that the $L_{tk}$ are 0 mean $\sigma_k$ sub-exponential random variables (see 2.7.5 Vershynin (2018)). We define the following running empirical estimate of $\alpha$:

$$\hat{\alpha}_t = \frac{1}{t}\sum_{\tau=1}^{t} (\hat{a}_{\tau,k_\tau} + 1)/2.$$

It is an unbiased estimator of $\alpha$ as

$$\mathbb{E}[\hat{a}_{tk}] = \mathbb{E}[a_t] + \mathbb{E}[L_{tk}] = 2\alpha - 1 + 0 = 2\alpha - 1.$$

Note that the $\hat{a}_{t,k_t}$ are *not* independent, and the conditional expectation is *not* Lispchitz-continuous with respect to the parameter $\alpha$. Still, we can derive an algorithm with a sub-linear regret bound.

First let us state a Lemma on the concentration of $\hat{\alpha}_t$

**Lemma I.1.** *We have the following concentration bound for $t \gtrsim \log(T)$:*

$$\Pr\left(|\hat{\alpha}_t - \alpha| \geq \sqrt{\frac{\kappa^2 \log(T)}{ct}}\right) \leq \frac{2}{T} \tag{41}$$

*Proof.* We refer to Chapter 2 of Vershynin (2018) for concentration results of sum of independent random variables. Let us recall some definitions, properties, and a concentration inequality.

Let us consider the following Orlicz space norm for random variables. For a random variable $X$, we define

$$\|X\|_{\psi^1} = \inf\{t > 0, \mathbb{E}[\exp(|X|/t)] \leq 2\}.$$

If $\|X\|_{\psi^1}$ is finite, then $X$ is called sub-exponential. It is a norm, and for a bounded variable $X$ in $[-1, 1]$, we have $\|X\|_{\psi^1} \leq 1/\log(2)$ (see exercise 2.5.7, example 2.5.8, and Lemma 2.7.6 of Vershynin (2018)). Hence because $(a_t + 1)/2 - \alpha \in [-\alpha, 1 - \alpha] \subset [-1, 1]$, for $\hat{\alpha}_{tk}$ the estimate obtained by always choosing source $k$, we have by triangle inequality that

$$\|\hat{\alpha}_{tk} - \alpha\|_{\psi^1} \leq \frac{1}{\log(2)} + \frac{\sigma_k}{2}.$$

We now recall Bernstein's inequality:

**Theorem** (Theorem 2.8.1 of Vershynin (2018)). *Let $X_1, \ldots, X_N$ be $N$ independent, mean $0$, sub-exponential random variables, with $\kappa = \max_i \|X_i\|_{\psi^1}$. Then for $c > 0$ a constant, for every $t > 0$ we have*

$$\Pr\left\{\left|\sum_{i=1}^N X_i\right| \geq t\right\} \leq 2\exp\left[-c\min\left(\frac{t^2}{N\kappa^2}, \frac{t}{\kappa}\right)\right].$$

We want to apply this Theorem to $(\hat{a}_{t,k_t} + 1)/2 - \alpha$ the centered $\hat{\alpha}_t$ where we now change the source depending on $k_t$. Remark that now however the $\hat{\alpha}_t$ is not a sum of independent variables anymore because of $k_t$ which does depend on the previous realizations of the $\hat{a}_{t,k_t}$. Nevertheless it is a martingale difference for the filtration $\mathcal{H}_t$, and similar results will apply. We denote by $\kappa = 1/\log(2) + \max_k \sigma_k/2$ an upper bound on the sub-exponential norm of the centered $\hat{\alpha}_t$ conditional on $\mathcal{H}_{t-1}$. Then the martingale $\sum_{\tau=1}^t \hat{\alpha}_\tau - \alpha$ is also a sub-exponential random variable.

This is a known result, which is stated here for completeness, but that can otherwise be skipped. Indeed from proposition 2.7.1 part $e$) of Vershynin (2018), we know that finite sub-exponential norm for a centered random variable $X$ means that the moment generating function of $X$ at $y$ is bounded by $\exp(c_1\|X\|_{\psi^1}^2 y^2)$ for $y \leq c_2/\|X\|_{\psi^1}$, with $c_1$ and $c_2$ some positive constants.

Therefore, for $y \leq c_2/\kappa$, using the property of the conditional sub-exponentiality of $\hat{\alpha}_t$, we have

$$\mathbb{E}[\exp(y\sum_{\tau=1}^t(\hat{\alpha}_\tau - \alpha))] = \mathbb{E}[\mathbb{E}[\exp(y\sum_{\tau=1}^t(\hat{\alpha}_\tau - \alpha)) \mid \mathcal{F}_{t-1}]]$$

$$= \mathbb{E}[\exp(y\sum_{\tau=1}^{t-1}(\hat{\alpha}_\tau - \alpha))\mathbb{E}[\exp(y(\hat{\alpha}_t - \alpha)) \mid \mathcal{F}_{t-1}]]$$

$$\leq \mathbb{E}[\exp(y\sum_{\tau=1}^{t-1}(\hat{\alpha}_\tau - \alpha))\mathbb{E}[\exp(c_1 y^2\kappa^2) \mid \mathcal{F}_{t-1}]].$$

Iterating this inequality we obtain for $y \leq c_2/\kappa$ that

$$\mathbb{E}[\exp(y\sum_{\tau=1}^t(\hat{\alpha}_\tau - \alpha))] \leq \mathbb{E}[\exp(c_1 t\kappa^2 y^2)].$$

This yields inequality (2.24) in Vershynin (2018) the main element needed for the proof of Bernstein's Inequality. Thus Bernstein's inequality still holds for centered martingale sequence which are conditionally sub-exponential. We stress that this is not a new concentration inequality result.

Let us now apply Bernstein's Inequality with a deviation of order $\sqrt{\kappa^2 \log(T)/(ct)}$, for $t \geq \log(T)/c$:

$$\Pr\left(|\hat{\alpha}_t - \alpha| \geq \sqrt{\frac{\kappa^2 \log(T)}{ct}}\right) = \Pr\left(|\sum_{\tau=1}^{t}(\hat{\alpha}_\tau - \alpha)| \geq \sqrt{\frac{\kappa^2 t \log(T)}{c}}\right)$$

$$\leq 2\exp\left[-c\min\left(\frac{\log(T)}{c}, \sqrt{\frac{\log(T)t}{c}}\right)\right]$$

$$= \frac{2}{T}$$

$\square$

**Proposition I.2.** *Given these assumptions, using Algorithm 1 with $\hat{\alpha}_t$ instead of $\alpha$ to compute the conditional expectations yields an added regret of order $\mathcal{O}((\max_k \sigma_k + 2/\log(2))\sqrt{T\log(T)})$.*

*Proof.* There are two parts of the proof of Theorem 3.1 that we need to modify, when the $x_t$ is involved, and when we compare the penalty. We will deal with the first part (the more complicated), and the second one follows immediately by the same arguments. This will be done in 3 steps: we first highlight where the learning error occurs, then use the concentration result to decompose the error under a good and bad event, and finally show that under the good event the error in expectation is small enough.

**Impact of learning error** We suppose for simplicity that $L_{tk}$ has a density $g_k$ over the support $\mathbb{R}$, but this also holds for discrete random variables. We denote by $\nu_k$ the density of $\hat{a}_{tk}$. First let us express $\tilde{a}_t = \mathbb{E}[a_t \mid c_{tk_t}]$ as a function of $\hat{a}_{t,k_t} = \hat{a}$ and $\alpha$:

$$\mathbb{E}[a_t \mid \hat{a}_{t,k_t} = \hat{a}] = \Pr(a_t = 1 \mid \hat{a}_{t,k_t} = \hat{a}) - \Pr(a_t = -1 \mid \hat{a}_{t,k_t} = \hat{a})$$

$$= \frac{\Pr(a_t = 1)g_{k_t}(\hat{a} - 1)}{\nu_{k_t}(\hat{a})} - \frac{\Pr(a_t = -1)g_{k_t}(\hat{a} + 1)}{\nu_{k_t}(\hat{a})}$$

$$= \frac{\alpha g_{k_t}(\hat{a} - 1) - (1 - \alpha)g_{k_t}(\hat{a} + 1)}{\alpha g_{k_t}(\hat{a} - 1) + (1 - \alpha)g_{k_t}(\hat{a} + 1)}.$$

Let us define the plug-in estimate of $\tilde{a}_t$ using $\bar{\alpha}_{t-1}$ whenever $\bar{\alpha}_{t-1}$ is in $[0, 1]$ by:

$$s_t(\hat{a}) = \frac{\bar{\alpha}_{t-1}g_{k_t}(\hat{a} - 1) - (1 - \bar{\alpha}_{t-1})g_{k_t}(\hat{a} + 1)}{\bar{\alpha}_{t-1}g_{k_t}(\hat{a} - 1) + (1 - \bar{\alpha}_{t-1})g_{k_t}(\hat{a} + 1)}. \tag{42}$$

When the estimated parameter is not in $[0, 1]$, we use $s_t = 0$.

We define by $x_t$ the maximisation of the virtual value function $\varphi$ when replacing $\tilde{a}_t$ by $s_t$, and by $x_t^*$ the allocation obtained if we were to actually use $\tilde{a}_t$. Because $x_t$ is a maximizer of $u_t x_t - \langle \lambda_t, s_t \rangle x_t$, we have

$$u_t x_t = u_t x_t - \langle \lambda_t, s_t \rangle x_t + \langle \lambda_t, s_t \rangle x_t$$
$$\geq u_t x_t^* - \langle \lambda_t, s_t \rangle x_t^* + \langle \lambda_t, s_t \rangle x_t$$
$$= \varphi(\lambda_t, \hat{a}_{t,k_t}, k_t) + \langle \lambda_t, \tilde{a}_t - s_t \rangle x_t^* + \langle \lambda_t, s_t \rangle x_t.$$

The term $\langle \lambda_t, s_t \rangle x_t$ will be tracked with the help of the OGD, it remains to take care of $\langle \lambda_t, \tilde{a}_t - s_t \rangle x_t^*$. By Cauchy Schwartz:

$$|\langle \lambda_t, \tilde{a}_t - s_t \rangle x_t^*| \leq \|\lambda_t\|_2 \|\tilde{a}_t - s_t\|_2.$$

We know that $\lambda_t$ is already bounded, so we only need to bound the right-hand term. Decomposing depending on the values of $k_t$, we can rewrite

$$s_t - \tilde{a}_t = \sum_{k=1}^{K} \mathbb{1}[k_t = k](s_t - \mathbb{E}[a_t \mid \hat{a}_{tk}])$$

**Decomposition under good and bad event** Denote by $\mathcal{C}_t = [|\hat{\alpha}_t - \alpha| \leq \sqrt{(\kappa^2 \log(T)/(ct)}]$ the event of $\hat{\alpha}_t$ concentrating around its mean. For $t$ large enough, that is to say $t \geq (\kappa^2 \log(T))/(c\min(\alpha, 1-\alpha))$, the event $\mathcal{C}_t$ is included in $[\bar{\alpha}_{t-1} \in (0,1)]$. We can then decompose the analysis over these events using the triangle inequality: $|s_t - \tilde{a}_t| \leq \mathbb{1}_{\mathcal{C}_{t-1}}|s_t - \tilde{a}_t| + \mathbb{1}_{\bar{\mathcal{C}}_{t-1}}|s_t - \tilde{a}_t|$, and we know that for the term multiplied by $\mathbb{1}_{\mathcal{C}_{t-1}}$ we have that either it is 0 or $\bar{\alpha}_{t-1} \in [0,1]$. Let us consider such a $t-1$ big enough, and first analyze the left-hand term with the "good" event.

**Small error under good event in expectation** We consider the conditional expectation of this sum given $k_t$, and $\bar{\alpha}_{t-1}$. By independence between $(k_t, \bar{\alpha}_{t-1}, E)$ and $\hat{a}_{tk}$ for all $k$, we have:

$$\mathbb{E}[\mathbb{1}_{\mathcal{C}_{t-1}}|s_t - \tilde{a}_t| \mid k_t, \bar{\alpha}_{t-1}]$$
$$\leq \mathbb{1}_{\mathcal{C}_{t-1}} \sum_{k=1}^{K} \mathbb{1}[k_t = k] \int_{\mathbb{R}} \left| \frac{\bar{\alpha}_{t-1}g_k(\hat{a}-1) - (1-\bar{\alpha}_{t-1})g_k(\hat{a}+1)}{\bar{\alpha}_{t-1}g_k(\hat{a}-1) + (1-\bar{\alpha}_{t-1})g_k(\hat{a}+1)} - \frac{\alpha g_k(\hat{a}-1) - (1-\alpha)g_k(\hat{a}+1)}{\alpha g_k(\hat{a}-1) + (1-\alpha)g_k(\hat{a}+1)} \right| \nu_k(\hat{a})d\hat{a}.$$

Because $\nu_{k_t}(\hat{a}) = (\alpha g_k(\hat{a}-1) + (1-\alpha)g_k(\hat{a}+1))$ we can simplify the integral:

$$\mathbb{E}[\mathbb{1}_{\mathcal{C}_{t-1}}|s_t - \tilde{a}_t| \mid k_t, \bar{\alpha}_{t-1}] \leq \mathbb{1}_{\mathcal{C}_{t-1}} \sum_{k=1}^{K} \mathbb{1}[k_t = k] \int_{\mathbb{R}} 2|\bar{\alpha}_{t-1} - \alpha| \frac{g_k(\hat{a}-1)g_k(\hat{a}+1)}{\bar{\alpha}_{t-1}g_k(\hat{a}-1) + (1-\bar{\alpha}_{t-1})g_k(\hat{a}+1)} d\hat{a}.$$

For $\alpha \in [0,1]$, we have the following inequality:

$$\frac{xy}{\alpha x + (1-\alpha)y} \leq \max(x, y) \leq x + y.$$

Thus using this inequality and the fact that $g_k$ is a density (which sums to 1), we obtain that

$$\mathbb{E}[\mathbb{1}_{\mathcal{C}_{t-1}}|s_t - \tilde{a}_t| \mid k_t, \bar{\alpha}_{t-1}] \leq 2\mathbb{1}_{\mathcal{C}_{t-1}}|\bar{\alpha}_{t-1} - \alpha| \sum_{k=1}^{K} \mathbb{1}[k_t = k]\left(\int_{\mathbb{R}} g_k(\hat{a}+1)d\hat{a} + \int_{\mathbb{R}} g_k(\hat{a}-1)d\hat{a}\right)$$

$$\leq 4\mathbb{1}_{\mathcal{C}_{t-1}}|\bar{\alpha}_{t-1} - \alpha| \sum_{k=1}^{K} \mathbb{1}[k_t = k]$$

$$\leq 4\sqrt{\frac{\kappa^2 \log(T)}{c(t-1)}} \sum_{k=1}^{K} \mathbb{1}[k_t = k].$$

Finally taking the full expectation yields

$$\mathbb{E}[\mathbb{1}_{\mathcal{C}_{t-1}}|s_t - \tilde{a}_t|] \leq 4\sqrt{\frac{\kappa^2 \log(T)}{c(t-1)}}.$$

For the complementary event $\bar{\mathcal{C}}_{t-1}$:

$$\mathbb{E}[\mathbb{1}_{\bar{\mathcal{C}}_{t-1}}|s_t - \tilde{a}_t|] \leq 2\mathbb{E}[\mathbb{1}_{\bar{\mathcal{C}}_{t-1}}] = \frac{4}{T}.$$

**Putting everything back together**   Now denoting by $m_\lambda$ the upper bound over the $\lambda_t$ obtained by Lemma 3.2 and summing over all $t$:

$$|\mathbb{E}[\sum_{t=1}^{T}\langle \lambda_t, \tilde{a}_t - s_t\rangle x_t^*]| \leq m_\lambda \sum_{t=1}^{T}\mathbb{E}[|s_t - \tilde{a}_t|]$$

$$\leq 2m_\lambda \frac{\kappa^2 \log(T)}{c\min(\alpha, 1-\alpha)} + m_\lambda \sum_{t=\frac{\kappa^2 \log(T)}{c\min(\alpha, 1-\alpha)}+1}^{T}\mathbb{E}[|s_t - \tilde{a}_t|]$$

$$\leq 2m_\lambda \frac{\kappa^2 \log(T)}{c\min(\alpha, 1-\alpha)} + m_\lambda \sum_{t=\frac{\kappa^2 \log(T)}{c\min(\alpha, 1-\alpha)}+1}^{T}(4\sqrt{\frac{\kappa^2 \log(T)}{c(t-1)}} + \frac{4}{T})$$

$$\leq 2m_\lambda \frac{\kappa^2 \log(T)}{c\min(\alpha, 1-\alpha)} + m_\lambda \sum_{t=1}^{T}(4\sqrt{\frac{\kappa^2 \log(T)}{ct}} + \frac{4}{T})$$

$$\leq m_\lambda (\frac{8}{\sqrt{c}}\sqrt{\kappa^2 T \log(T)} + 2\frac{\kappa^2 \log(T)}{c\min(\alpha, 1-\alpha)} + 4) = \mathcal{O}(\sqrt{T\log(T)}),$$

where we used a series-integral comparison of $x \mapsto 1/\sqrt{x}$ for the second to last inequality.

There is a second place where replacing $\tilde{a}_t$ by $s_t$ impacts the performance of the algorithm. Indeed, through the gradient descent on $\lambda_t$, we will have a penalty converging towards $R(\sum_{t=1}^{T} s_t x_t/T)$ and not $R(\sum_{t=1}^{T} \tilde{a}_t x_t/T)$. This is because the Online Gradient Descent for the $\lambda_t$ uses $s_t$ and not $\tilde{a}_t$. Nevertheless, because of the Lipschitz property of $R$:

$$\mathbb{E}[T|R(\frac{\sum_{t=1}^{T}\tilde{a}_t x_t}{T}) - R(\frac{\sum_{t=1}^{T}s_t x_t}{T})|] \leq L\sum_{t=1}^{T}\mathbb{E}[|\tilde{a}_t - s_t|] \leq \mathcal{O}(\sqrt{T\log(T)}),$$

where the last inequality is obtained by similar manipulations on $\mathcal{C}_t$ to what we just did.

Overall, the added regret when using $s_t$ instead of $\tilde{a}_t$ is of order $\sqrt{T\log(T)}$ which still guarantees a sub-linear regret. $\qquad\square$

Another algorithm is possible: first sample the source with the lowest sub-exponential constant $T^{2/3}\log(T)$ times, and then use the estimate obtained to compute $s_t$ (without updating it with the newest noise received). The advantage is that even though we incur a $T^{2/3}$ regret (this can be shown using basically the same methods), we can obtain a lower $\kappa$. Hence if there is a big difference between the sub-exponential constant of the different sources, or even one source with infinite sub-exponential constant, this may be a viable algorithm.

# J   Public Contexts

For this section and in order to make the dependence in the public context $z_t$ appear more clearly, we suppose that $c_{tk}$ represent only the additional information of the source $k$, and thus $(c_{tk_t}, z_t)$ represent the whole information available at time $t$ after selecting a source.

## J.1   Finite number of Contexts — proof of Proposition 3.4

We propose the following Algorithm 4.

**Proposition.** *For $\mu$ the probability distribution over $\mathcal{Z}$ finite, we can derive an algorithm that has a modified regret of order $\mathcal{O}(\sqrt{TK\log(K)}\sum_{z\in\mathcal{Z}}\sqrt{\mu(z)})$, where $\mu(z)$ is the probability that the public attribute is $z$.*

*Proof.* Most of the proof consists is identical to the proof of Theorem 3.1, and only the different parts will be highlighted, which is the upper bound of OPT, and applying bandits algorithms.

**Algorithm 4** Online Fair Allocation with Source Selection — With public contexts

---

**Input:** Initial dual parameter $\lambda_0$, initial source-selection-distribution $\pi_0(z) = (1/K, \ldots, 1/K)$, dual gradient descent step size $\eta$, EXP3 step size $\rho_t(z)$ that uses a doubling trick, cumulative estimated rewards $S_0(z) = 0 \in \mathbb{R}^K$ for all $z \in \mathcal{Z}$.

**for** $t \in [T]$ **do**

    Observe $z_t \in \mathcal{Z}$

    Draw a source $k_t \sim \pi_t(z_t)$, where $\pi_{tk}(z_t) \propto \exp(\rho_t(z_t)S_{(t-1)k}(z_t))$ and observe $c_{tk_t}$.

    Compute the allocation for user $t$:

$$x_t = \arg\max_{x \in \mathcal{X}} \mathbb{E}[f_t(x) - \langle \lambda_t, a_t(x) \rangle \mid c_{tk_t}, z_t].$$

    Update the estimated rewards sum and sources distributions for all $k \in [K]$:

$$\begin{aligned} S_{tk}(z_t) &= S_{t-1,k}(z_t) + \hat{\varphi}(\lambda_t, c_{tk}, k_t, z_t), \\ S_{tk}(z) &= S_{t-1,k}(z), \quad \forall z \in \mathcal{Z} \setminus \{z_t\}. \end{aligned}$$

    Compute the expected protected group allocation $\delta_t = \mathbb{E}[a_t(x_t) \mid c_{tk_t}, z_t, x_t]$ and compute the dual protected groups allocation target and update the dual parameter:

$$\begin{aligned} \gamma_t &= \arg\max_{\gamma \in \Delta_{\delta_t}} \{ \langle \lambda_t, \gamma \rangle - \bar{R}(\gamma) \}, \\ \lambda_{t+1} &= \lambda_t - \eta(\gamma_t - \delta_t). \end{aligned}$$

**end for**

---

We first upper bound the performance of OPT. For all $t$ let $h_t \in [K]^{\mathcal{Z}}$ be the deterministic source selection policies, $k_t = h_t(z_t)$ the selected sources, $x_t \in \mathcal{X}$ be $(z_t, c_{t,k_t})$ measurable, and $\lambda \in \mathbb{R}^d$ be a dual variable. In an identical manner to Equation (20) we can obtain the following inequality:

$$\max_{\mathbf{x} \in \mathcal{X}^T} \mathbb{E}[\mathcal{U}(\boldsymbol{x}, \boldsymbol{k}) \mid z_1, c_{1,k_1}, \ldots, z_T, c_{T,k_T}] \leq \sum_{t=1}^{T} \max_{\mathbf{x} \in \mathcal{X}} \mathbb{E}[f_t(x) - \langle \lambda, a_t(x) \rangle \mid z_t, c_{tk_t}] - p_{k_t} + TR^*(\lambda).$$

Let us define $\varphi(\lambda, c_{tk}, k, z) = \max_{\mathbf{x} \in \mathcal{X}} \mathbb{E}[f_t(x) - \langle \lambda, a_t(x) \rangle \mid z, c_{tk}] - p_k$, and $\mu(z) = \Pr(z_t = z)$. We also define $\pi_k(z) = \sum_{t=1}^{T} \mathbb{1}[h_t(z) = k]/T$ the total number of times we would have selected source $k$ for the public context $z$.

We now take the expectation of the sum. We obtain by tower property of the conditional expectation, and using the i.i.d. assumption:

$$\begin{aligned} \sum_{t=1}^{T} \mathbb{E}[\varphi(\lambda, c_{t,h_t(z_t)}, h_t(z_t), z_t)] &= \sum_{t=1}^{T} \mathbb{E}[\mathbb{E}[\varphi(\lambda, c, h_t(z_t), z_t) \mid z_t]] \\ &= \sum_{t=1}^{T} \sum_{z \in \mathcal{Z}} \mathbb{E}[\varphi(\lambda, c, h_t(z), z) \mid z]\mu(z) \\ &= \sum_{z \in \mathcal{Z}} \sum_{t=1}^{T} \mathbb{E}[\varphi(\lambda, c, h_t(z), z) \mid z]\mu(z) \\ &= \sum_{z \in \mathcal{Z}} \sum_{k \in [K]} \sum_{\substack{t=1 \\ h_t(z)=k}}^{T} \mathbb{E}[\varphi(\lambda, c, k, z) \mid z]\mu(z) \\ &= T \sum_{z \in \mathcal{Z}} \sum_{k \in [K]} \pi_k(z)\mathbb{E}_c[\varphi(\lambda, c, k, z) \mid z]\mu(z). \end{aligned}$$

Clearly $\pi_k(z) \in \mathcal{P}_K$, thus if we maximize over this space for the $\pi_k(z)$ instead of over $([K]^{\mathcal{Z}})^T$, it will be higher. Therefore

$$\text{OPT} \leq T \sup_{\pi \in \mathcal{P}_K^{\mathcal{Z}}} \inf_{\lambda \in \mathbb{R}^d} \left( R^*(\lambda) + \sum_{z \in \mathcal{Z}} \sum_{k \in [K]} \pi_k(z)\mathbb{E}_c[\varphi(\lambda, c, k, z) \mid z]\mu(z) \right). \tag{43}$$

Notice that $\mu(z)$ plays an important role in determining the optimal policy. If we had simply run the previous algorithm for each context, this would be akin to considering that the $z_t$ are distributed uniformly according to $\mu(z) = 1/Z$, which would have yielded a sub-optimal policy $\pi$.

Now let us take care of the lower bound. As done in Equation (25), we have that

$$\mathbb{E}[f_t(x_t) - p_{k_t} - \bar{R}(\gamma_t)] \geq \mathbb{E}[\varphi(\lambda_t, k_t, z_t, c_{tk_t}) + R^*(\lambda_t) + \langle \lambda_t, \delta_t - \gamma_t \rangle],$$

and as we did before we will use bandits algorithm on these rewards.

We would now like to bound the following adversarial contextual bandit problem regret:

$$\text{Reg} = \sum_{z \in \mathcal{Z}} \max_{k \in [K]} \sum_{\substack{t=1 \\ z_t = z}}^{T} \mathbb{E}[\varphi(\lambda_t, c_t, k, z) \mid \mathcal{H}_{t-1}, z_t] - \sum_{t=1}^{T} \mathbb{E}[\varphi(\lambda_t, c_t, k_t, z_t) \mid \mathcal{H}_{t-1}, z_t].$$

The weighted estimator $\hat{\varphi}$ is unbiased as $\varphi(\lambda_t, c_t, k_t, z_t)\mathbb{1}[k_t = k]/\pi_{tk}(z_t)$ is still unbiased. Indeed, $c_t$ and $k_t$ are conditionally independent given $(z_t, \mathcal{H}_{t-1})$, and $\pi_{tk}(z_t)$ is $(z_t, \mathcal{H}_{t-1})$ measurable. Thus

$$\mathbb{E}\Big[\frac{\varphi(\lambda_t, c_t, k_t, z_t)\mathbb{1}[k_t = k]}{\pi_{tk}(z_t)} \mid \mathcal{H}_{t-1}, z_t\Big] = \frac{\mathbb{E}[\varphi(\lambda_t, c_t, k_t, z_t)\mathbb{1}[k_t = k] \mid \mathcal{H}_{t-1}, z_t]}{\pi_{tk}(z_t)}$$

$$= \frac{\mathbb{E}[\varphi(\lambda_t, c_t, k_t, z_t) \mid \mathcal{H}_{t-1}, z_t]\mathbb{E}[\mathbb{1}[k_t = k] \mid \mathcal{H}_{t-1}, z_t]}{\pi_{tk}(z_t)}$$

$$= \mathbb{E}[\varphi(\lambda_t, c_t, k_t, z_t) \mid \mathcal{H}_{t-1}, z_t].$$

As suggested section $(18.1)$ of Lattimore & Szepesvári (2020) running one anytime version of $EXP3$ (using a doubling trick for instance) on each of these public contexts $z \in \mathcal{Z}$, achieves a regret bound of order $\mathcal{O}\big(\sqrt{K \log(K)} \sum_{z \in \mathcal{Z}} \sqrt{\sum_{t=1}^{T} \mathbb{1}[z_t = z]}\big)$.

Taking the expectation (for $z_t$) of this regret bound, and using Jensen's inequality for the square root function we have a regret term of order

$$\mathbb{E}\big[\mathcal{O}\big(\sqrt{K \log(K)} \sum_{z \in \mathcal{Z}} \sqrt{\sum_{t \in [T]} \mathbb{1}[z_t = z]}\big)\big] \leq \mathcal{O}\big(\sqrt{K \log(K)} \sum_{z \in \mathcal{Z}} \sqrt{\mathbb{E}[\sum_{t \in [T]} \mathbb{1}[z_t = z]]}\big)$$

$$= \mathcal{O}\big(\sqrt{TK \log(K)} \sum_{z \in \mathcal{Z}} \sqrt{\mu(z)}\big). \tag{44}$$

Let $(\pi^n) \in (\mathcal{P}_K^{\mathcal{Z}})^{\mathbb{N}}$ be the sequence of contextual source mixing that maximizes the upper bound for OPT in Equation (43). Then because the maximum of $[K]$ points is greater than any convex combination, $\pi^n(z)$ in particular, we have

$$\sum_{z \in \mathcal{Z}} \max_{k \in [K]} \sum_{\substack{t=1 \\ z_t = z}}^{T} \mathbb{E}[\varphi(\lambda_t, c_t, k, z) \mid \mathcal{H}_{t-1}, z] \geq \sum_{z \in \mathcal{Z}} \sum_{k \in [K]} \pi_k^n(z) \sum_{\substack{t=1 \\ z_t = z}}^{T} \mathbb{E}[\varphi(\lambda_t, c, k, z) \mid \mathcal{H}_{t-1}, z]$$

$$= \sum_{k \in [K]} \sum_{t=1}^{T} \pi_k^n(z_t)\mathbb{E}[\varphi(\lambda_t, c, k, z_t) \mid \mathcal{H}_{t-1}, z_t].$$

In expectation this yields:

$$\mathbb{E}\big[\sum_{z \in \mathcal{Z}} \max_{k \in [K]} \sum_{\substack{t=1 \\ z_t = z}}^{T} \mathbb{E}[\varphi(\lambda_t, c_t, k, z) \mid \mathcal{H}_{t-1}, z]\big] \geq \mathbb{E}\big[\sum_{k \in [K]} \sum_{t=1}^{T} \pi_k^n(z_t)\mathbb{E}[\varphi(\lambda_t, c, k, z_t) \mid \mathcal{H}_{t-1}, z_t]\big]$$

$$= \sum_{k \in [K]} \sum_{t=1}^{T} \mathbb{E}[\mathbb{E}[\pi_k^n(z_t)\mathbb{E}[\varphi(\lambda_t, c, k, z_t) \mid \mathcal{H}_{t-1}, z_t] \mid \mathcal{H}_{t-1}]],$$

where we used the linearity of the expectation, and the tower property of the expectation for the last equality.

Because $z_t$ and $\mathcal{H}_{t-1}$ are independent, we can first use Lemma D.3 and then the freezing lemma to express the conditional expectation given $\mathcal{H}_{t-1}$ as

$$\mathbb{E}[\mathbb{E}[\pi_k^n(z_t)\mathbb{E}[\varphi(\lambda_t, c, k, z_t) \mid \mathcal{H}_{t-1}, z_t] \mid \mathcal{H}_{t-1}]] = \mathbb{E}[\mathbb{E}[\pi_k^n(z_t)\mathbb{E}_c[\varphi(\lambda_t, c, k, z) \mid z] \mid \mathcal{H}_{t-1}]]$$
$$= \mathbb{E}[\sum_{z \in \mathcal{Z}} \mu(z)\pi_k^n(z_t)\mathbb{E}_c[\varphi(\lambda_t, c, k, z) \mid z]]$$

Hence,

$$\mathbb{E}[\sum_{z \in \mathcal{Z}} \max_{k \in [K]} \sum_{\substack{t=1 \\ z_t = z}}^{T} \mathbb{E}[\varphi(\lambda_t, c_t, k, z) \mid \mathcal{H}_{t-1}, z]] \geq \sum_{t=1}^{T} \mathbb{E}[\sum_{k \in [K]} \sum_{z \in \mathcal{Z}} \mu(z)\pi_k^n(z)\mathbb{E}_c[\varphi(\lambda_t, c, k, z) \mid z]].$$

Grouping it together with the $R^*(\lambda_t)$ terms, and taking the inf over $\lambda$ we derive that

$$\mathbb{E}\left[\sum_{t=1}^{T} R^*(\lambda_t) + \sum_{z \in \mathcal{Z}} \max_{k \in [K]} \sum_{\substack{t=1 \\ z_t = z}}^{T} \mathbb{E}[\varphi(\lambda_t, c_t, k, z) \mid \mathcal{H}_{t-1}, z]\right]$$

$$\geq \sum_{t=1}^{T} \mathbb{E}\left[\inf_{\lambda \in \mathbb{R}^d}\left(R^*(\lambda) + \sum_{k \in [K]} \sum_{z \in \mathcal{Z}} \mu(z)\pi_k^n(z)\mathbb{E}_c[\varphi(\lambda, c, k, z) \mid z]\right)\right]$$

$$\geq T \inf_{\lambda \in \mathbb{R}^d}\left(R^*(\lambda) + \sum_{k \in [K]} \sum_{z \in \mathcal{Z}} \mu(z)\pi_k^n(z)\mathbb{E}_c[\varphi(\lambda, c, k, z) \mid z]\right).$$

As $n \to \infty$ and by definition of $\pi^n$, we can conclude that

$$\mathbb{E}\left[\sum_{t=1}^{T} R^*(\lambda_t) + \sum_{z \in \mathcal{Z}} \max_{k \in [K]} \sum_{\substack{t=1 \\ z_t = z}}^{T} \mathbb{E}[\varphi(\lambda_t, c_t, k, z) \mid \mathcal{H}_{t-1}, z]\right]$$

$$= T \sup_{\pi \in \mathcal{P}_K^{\mathcal{Z}}} \inf_{\lambda \in \mathbb{R}^d}\left(R^*(\lambda) + \sum_{z \in \mathcal{Z}} \sum_{k \in [K]} \pi_k(z)\mathbb{E}_c[\varphi(\lambda, c, k, z) \mid z]\mu(z)\right)$$

$$\geq \text{OPT}.$$

Finally, using Equation (44) and the previous inequality we obtain

$$\mathbb{E}[\sum_{t=1}^{T} \varphi(\lambda_t, k_t, z_t, c_t) + R^*(\lambda_t)] \geq \text{OPT} - \mathcal{O}\big(\sqrt{TK \log(K)} \sum_{z \in \mathcal{Z}} \sqrt{\mu(z)}\big). \tag{45}$$

The rest of the proof follows Appendix E. $\qquad\square$

Clearly, if $\mu$ has a non-zero probability only for one $z$, we recover the previous regret bound without public information.

## J.2  Infinite number of contexts — proof of Proposition 3.5

**Lemma.** *If $\mathbb{E}[a_t(x) \mid c_{tk}, z_t]$ and $\mathbb{E}[f_t(x) \mid c_{tk}, z_t]$ are Lipschitz continuous in $z$ with respective constants $L_a$ and $L_f$ for all $x \in \mathcal{X}$ with respect to $\|\cdot\|_2$, then so is $\varphi$ in $z$ with Lipschitz constant $L_a + L_f$.*

*Proof.* Let $\lambda \in \mathbb{R}^d$, $k \in [K]$, $c_{tk}$ the additional information, $z_1, z_2 \in \mathcal{Z}$, and $x_1, x_2 \in \mathcal{X}$ be the maximizers that respectively yields $\varphi(\lambda, c_{tk}, k, z_1)$ and $\varphi(\lambda, c_{tk}, k, z_2)$. Because $x_2$ is a maximizer, we have that

$$\varphi(\lambda, c_{tk}, k, z_1) - \varphi(\lambda, c_{tk}, k, z_2)$$
$$\leq \mathbb{E}[f_t(x_1) \mid c_{tk}, z_1] - \mathbb{E}[f_t(x_1) \mid c_{tk}, z_1] + \langle \lambda \mid \mathbb{E}[a_t(x_1) \mid c_{tk}, z_2] - \mathbb{E}[a_t(x_1) \mid c_{tk}, z_2]\rangle$$
$$\leq L_f\|z_1 - z_2\|_2 + L_a\|a_t(x_1)\|_2\|z_1 - z_2\|_2$$
$$\leq L_f + L_a.$$

Using the same arguments because $x_1$ is a maximizer, we obtain the symmetric inequality. $\qquad\square$

The main idea for the discretization (that averages the rewards over the discretized bins), is that the performance of the algorithm is close to the discretized optimal, which itself is close to the continuous optimal for smoothness reasons. However here the discretization is applied not to the original total utility $\mathcal{U}$, but instead to the bandits reward part. The number of discretized bins is then tuned depending on the lipschitz constants and the space dimension $r$.

The proof of the proposition Proposition 3.5 then directly follows from exercise 19.5 of Lattimore & Szepesvári (2020) applied to the contextual bandit problem with rewards $\varphi$.

*Remark.* In the case when $\mathcal{Z}$ is continuous, there are no guarantees that there exists an optimal policy $\pi^*$. Indeed, if for the product topology the space $\mathcal{P}_K^{\mathcal{Z}}$ is compact (by Tychonoff's theorem), the dual function used to derive OPT is not continuous. Vice versa, for a topology which makes this dual function continuous, it is unlikely for this space of functions to be compact. Hence why we need to take a maximizing sequence of $\pi_n$ so that the dual function of these $\pi_n$ converges to the upper bound of OPT.

