# OpenReview forum: "Trading-off price for data quality to achieve fair online allocation"
_NeurIPS.cc/2023/Conference — NeurIPS 2023 poster_

### Official Review · Reviewer_aPcw · 2023-07-05

**Soundness:** 4 excellent
**Presentation:** 4 excellent
**Contribution:** 4 excellent
**Rating:** 7
**Confidence:** 3

**Summary:**

Existing works in the fair online allocation space assume that the decision-maker has access to protected attributes. In contrast, this paper studies the choice of a data source as a multi-armed bandit problem in conjunction with the primary allocation problem. This paper introduces a computationally efficient algorithm with sublinear regret in the general case. The contributions of this paper are exclusively theory-based, including regret bounds for various types of regret penalties.

**Strengths:**

This is the first paper to consider optimizing over data sources in the problem space of fair online allocation. This paper contributes an algorithm with proven regret bounds, which is significant. I believe that other researchers are likely to build off of this work, and that it advances research in a demonstrable way.

This paper is excellently written. It is clear, well organized, and adequately informs the reader. In particular, I am happy to see good explanations of the reasoning, limitations, and assumptions made in this work.


**Weaknesses:**

This paper does not contribute code or empirical results; both of which would be of interest to the research community.

**Questions:**

N/A

**Limitations:**

Yes

---

> ### Author Rebuttal · Authors · 2023-08-09
>
> Thank you for this review!
>
> Regarding code and experimental results, we have presented a simple numerical toy example in the appendix F.2. Additionally, we made additional numerical simulations in a simple case showing how the optimal solution and value vary as we vary some parameters. We refer the reviewer to the global response and the PDF attached to it for details about this simulation, and will provide the code in the supplementary material.
>
> We are happy to discuss any additional questions you might have during the discussion period.

---

### Official Review · Reviewer_uY3q · 2023-07-07

**Soundness:** 4 excellent
**Presentation:** 2 fair
**Contribution:** 2 fair
**Rating:** 5
**Confidence:** 2

**Summary:**

The authors formulate the online fair allocation problem, where a decision maker may choose an information source $k_t$ based on public context $z_t$ and generate an allocation decision $x_t$ with a reward $1$ if $x_t=1$. The fairness penalty is some convex function that is minimized when the decision $x_t$ is uncoupled with the private type $a_t$. The authors developed a virtual-value-based algorithm that achieves $O(\sqrt{T})$ online regret.

**Strengths:**

The fairness problem is of interest to the community, especially when the private attributes are unseen.

**Weaknesses:**

It seems that the source selection part is simply the exponential weight algorithm with some virtual values which can be updated by the EXP3 algorithm. The allocation is based on the dual value $\lambda_t$ which can then be updated by the dual gradient descent. In my opinion, the authors did not make enough effort to explain the intuitions and justified the use of these algorithms, let alone the novelty of the work. Also, I'm not sure about the technical involvement, since it seems that the only challenge here is having multiple sources of information.

**Questions:**

What is the theoretical lower bound for this problem?

**Limitations:**

No concerns here.

---

> ### Author Rebuttal · Authors · 2023-08-09
>
> Thank you for this review!
>
> **Concerning the novelty and the technical difficulties**:
> As discussed in the abstract (line 7-8) and in the introduction (line 57-58), the main novelty of this work is to consider jointly the problem of source selection and of long term fairness penalty. Indeed, those two difficulties in isolation are already studied in the literature: When $R=0$, the model reduces to a stochastic bandit problem which can be solved using UCB or EXP3 as mentioned by the reviewer, and when $K=1$ we recover the existing works of Balseiro (2021) and Celli (2022) for long term fairness penalty. While it may seem that the joint problem can be simply treated by naively handling them separately, this is *not* the case as the fairness penalty introduces non linearity, and introduces interaction between those two aspects. Thus to a fixed source $k$ corresponds an optimal fair parameter $\lambda_k^*$, and to a fixed fairness parameter $\lambda$ corresponds an optimal source $k_{\lambda}^*$. Hence the selected source $k_t$ affects the fairness, and vice versa $\lambda_t$ affects the virtual fair-penalized mean of source $k$: both parameters change simultaneously and we are not necessarily selecting the best source for the best fairness penalty. How can we combine these two algorithms? Moreover, we have also shown that the interaction between those two problems is strong (line 180-181, line 63-65), as while optimal solutions in bandits problems are usually a fixed arm $k^*$, here due to the nonlinearity of $R$ the optimal source selection can be a strict convex combination of the sources (Proposition 2.1).
>
> Thus an important question is whether it is possible to combine the dual and primal step, and if yes, how. Perhaps surprisingly, alternating between the primal and dual step is sufficient and efficient, and there is no need to instantiate and keep track of a dual fairness parameter $\lambda_{k,t}$ for each source. This is, however, in no way obvious and is rather an outcome of our analysis. The main technical difficulty is then showing that this alternation does lead to a tight upper bound. On a more technical level we show how one can model the randomness and estimates, how we can bound the fair dual parameters to guarantee that adversarial bandits techniques can be used, and how to combine the analysis of primal and dual steps (see Sketch of Proof 3.4).
>
> Two other novelties of this work concern Section $4.2$ where we show how to obtain online estimates, and Section $4.1$ where we show that a similar algorithm works for another formulation of fairness which was desired in Celli (2022), even when there is only one source of information.
>
> **Lower bound**: Regarding the lower bound, the specific case $R=0$ reduces the problem to a pure stochastic bandit problem, which has a problem-independent $\Omega(\sqrt{T})$ lower bound. Therefore the proposed algorithm and its analysis is tight in $T$. We will add this comment after the upper bound result.
>
> We hope to have clarified the technical difficulties and novelty of our work, and would be happy to give more details during the discussion period.

---

> > ### Comment · Reviewer_uY3q · 2023-08-16
> >
> > I appreciate the authors' efforts in addressing my concerns. In my opinion, it would be beneficial for the authors to incorporate these clarifications and discuss in more detail on relations/differences to previous work in their paper.  Furthermore, I suggest considering a formal discussion of the challenges inherent in the problem. This is particularly important as the distinctions in the setup might be nuanced, and even a minor alteration could give rise to significant technical complexities.

---

> > > ### Author Response · Authors · 2023-08-18
> > >
> > > We thank the reviewer for their reply and the constructive feedback. We will make the distinction with previous works clearer in the introduction near the contribution section, as well as better highlight the technical difficulties and challenges inherent to this problem in Section 3.

---

### Official Review · Reviewer_Gv3t · 2023-07-07

**Soundness:** 3 good
**Presentation:** 2 fair
**Contribution:** 3 good
**Rating:** 6
**Confidence:** 3

**Summary:**

This paper studies an online allocation problem with long-term fairness constraints where, unlike prior works, the protected attributes of online users (e.g., gender, ethnicity) are not observed, but they can be purchased from $K$ different sources for some additional cost. The goal is to maximize the overall utility of the users minus the cost of additional information that was bought minus a fairness penalty that is characterized by a convex function $R(\cdot)$. The authors study this problem under an i.i.d. assumption on the users and propose Algorithm 1 with sublinear regret for the problem.

**Strengths:**

- Addressing fairness constraints in online allocation is a very well-motivated problem and the authors' assumption that protected attributes are not observed is a more realistic model of the applications. Therefore, the contributions of this work are of potential interest to a wide audience.
- Simultaneous source selection and dual gradient descent in Algorithm 1 is an important contribution that shows the naive approach of performing these two tasks separately is not good enough for this problem.
- The proposed algorithm is compatible with various notions of group fairness.

**Weaknesses:**

- The problem setup is a bit confusing. I'm not sure how the public context $z_t$, protected attributes $a_t$, and the purchased piece of information $c_{tk_t}$ are related to each other. On line 152, the authors make the assumption that $c_{tk_t}$ contains $z_t$ and I'm not quite sure if I understand what that means.
- The authors have distinguished their paper from prior works based on the fact that protected attributes are not observed and could only be purchased. However, a comparison of the paper with prior works in terms of techniques and proof ideas is missing. It appears that both primal and dual updates in Algorithm 1 are quite standard and the main challenge is to combine these two steps. This point needs to be highlighted and clarified.
- Given the vagueness of the problem framework, explaining the setting in the context of a few motivating applications is very helpful. In particular, the process of purchasing data from data sources needs clarification.

**Questions:**

- Can you provide a few motivating applications that could be cast into your framework and explain what data sources are and how they relate to each other?
- Can you explain the connection between $a_t$, $z_t$, and $c_{tk_t}$ in the context of a motivating application? How do these quantities relate to each other? What is the dimension of each quantity? How is $\mathbb{E}[a_t|c_{tk_t}]$ estimated?
- Assuming that $\sum_{t}x_t>0$, the function $(\sum_t x_t)R(\frac{\sum_t a_tx_t}{\sum_t x_t})$ is simply the perspective of the convex function $R(\cdot)$ and therefore, it should be convex in $x=(x_1,\dots,x_T)$. However, on lines 329-330, it is claimed that the opposite is true. Can you explain why convexity doesn't hold?
- Is it possible to use Algorithm 1 to obtain results for the adversarial setting? What would be the main challenges?
- On line 131, why is $c_{tk}$ enough to estimate $u_t$ and $a_t$? How about the public context information $z_t$, is it not helpful for estimation?


--------------------------------
I've read the authors' rebuttal, thanks for addressing my questions and concerns.

**Limitations:**

Discussion of potential negative societal impact is not necessary.

---

> ### Author Rebuttal · Authors · 2023-08-09
>
> Thank you for this review!
>
> **Relationship between various quantities and usefulness of $z_t$**: The quantities $(z_t,a_t,u_t,c_{t1},...,c_{tK})$ are random variables non necessarily independent, which are simultaneously drawn from some distribution, but which are only partially observed by the advertiser. Thus knowing $z_t$ may indeed yield more information on $a_t$ and $u_t$ and it is helpful to estimate them (of course if they are independent this is not the case). This information is used as $z_t$ is contained in $c_{t,k}$. The fact that $c_{tk}$ contains $z_t$ can be regarded more as a notation rather than an assumption. This was meant to avoid overloading notation; otherwise having to write $\mathbb{E}[u_t \mid (z_t,c_{tk_t})]$. We will simply define $c_{tk}=(\text{new purchased information}, z_t)$. We will clarify this in line 152.
>
> **Motivating examples and quantities of interests contextualized**:  Here are some examples of the information purchase model: Suppose that an advertiser wants to show ads to a user and is able to see some barebone information through cookies, such as previously visited websites ($z_t$). Based on this information, they can decide whether or not  to buy additional information ($c_{tk}$) from different data brokers that collect user activity. For instance, if they see from the public context that clothing shops were visited, they may buy data that contains purchase information from this website, which enables estimating the user’s gender ($a_t$) based on type of clothing bought. Another example would be fair refugees resettlement to different cities. When the organization in charge of resettlement receives a case ($z_t$), it can either decide to directly assign the refugee to a specific city, or to conduct an additional costly investigation to get more information ($c_{tk}$) on some protected attributes of interest such as wealth or age ($a_t$), which might have been misreported. We will add these examples after the model presentation.
>
> **Contributions and differences with previous literature**: The reviewer is correct in that the main difference compared to Balseiro (2021) and Celli (2022) is considering jointly source selection and fairness penalty, as well as handling uncertainty on $u_t$ and $a_t$. On a technical level, the proofs use the arguments regarding the fairness penalty of these works, and the main differences are the following: A martingale argument (Lemma $3.3$) enables the previous algorithm to be modified to handle uncertainty for a single fixed source by using conditional expectation, yielding one $\lambda_k$ per source $k$. Selecting the optimal source could be done by running a bandit algorithm for each $k$ using the optimal $\lambda_k^*$, but the $\lambda_k^*$ are not known a priori. Instead we instantiate a unique $\lambda_t$ for a combination of sources $\pi_t$, and handle the varying $\pi_t$ through EXP3 and combine its analysis with the $\lambda_t$ dual descent in Balseiro. An important condition in order to use EXP3 on the virtual values is for $\lambda_t$ to be bounded. Hence we optimize over $\Delta_{\delta_t}$ instead of $\Delta$ in Equation (8), which guarantees that $\lambda_t$ will be bounded (Lemma $3.2$) as we show using KKT conditions.  We will highlight in the contributions paragraph (line 57) that the main difference with previous work is the combination of those primal and dual steps, and add the above comments on the technical comparison with previous work near Section 3.4 (Sketch of proof).
>
> **Dimension of the various parameters**: the quantity $a_t$ is of dimension $d$ as defined in line $108$. For the decision making aspect, the dimension of $c_{tk}$ does not matter as it is only present in the conditioning of the expectation: what really matters is the sigma algebra generated by $c_{tk}$. When $\mathcal{Z}$ is finite (line 110) the same argument applies. If we want learn $\mathbb{E}[u_t \mid c_{t,k}]$ (Section 4.2), or deal with $\vert \mathcal{Z} \vert=\infty$ (Section 3.5), additional structure is needed: $c_{tk} \in \mathbb{R}^{q_k}$ (line 350) and $z_t \in \mathbb{R}^r$ (line 310).
>
> **Estimation of $\mathbb{E}[a_t \mid c_{tk}]$**: See reply "Estimates of protected attributes" to reviewer 2Yz9.
>
> **Convexity of the perspective of $R$**: There is indeed a typo, we meant to say that $R(\sum_t a_t x_t/ \sum_t x_t)$ is not convex, and the reviewer is correct that the function $\sum_t x_t R(\sum_t a_t x_t/ \sum_t x_t)$ is convex as the perspective of $R \circ A$ (with $A$ the matrix with columns the $a_t$), We thank the reviewer for catching this mistake! We will correct it, and add the mention of the perspective function as a clear intuition of why this fairness version can also be handled.
>
> **Adversarial inputs**: Yes, it should be possible to use Algorithm 1 for adversarial inputs (adversarial contexts $c_{tk}$) directly without additional modifications under some additional assumptions for $R$ and when there are no public contexts. This would yield a sublinear $\alpha$-regret bound instead of a sublinear regret bound (see Balseiro (2021) Theorem 2 for the assumption). The reason why the current analysis would still hold is that the EXP3 algorithm is able to handle adversarial inputs, and the boundedness of $\lambda_t$ is deterministic. Using in the fashion as Balseiro (2022) the assumption on $R$ to handle adversarial inputs for the fairness penalty, the analysis should then be fairly straightforward. We haven’t discussed the adversarial case, because some stochasticity is still needed for $\mathbb{E}[ u_t \mid c_{t,k}]$ to be well defined, and we found this mixed stochastic-adversarial model to be ambiguous. We can add this comment to the conclusion for completeness.
>
> We hope to have clearly addressed all your interrogations and thank you for the useful clarifying elements that were produced as a response to this review. We are happy to discuss any further concerns you might have during the discussion period.

---

> > ### Comment · Reviewer_Gv3t · 2023-08-15
> >
> > Thank you for your comprehensive response. My main confusion was about the relationship between various pieces of information and I think the motivating examples help clarify this point. I encourage the authors to maintain a running example throughout the paper and explain their model assumptions in the context of the example.

---

> > > ### Author Response · Authors · 2023-08-18
> > >
> > > We thank the reviewer for their reply and relevant suggestion, we will integrate the examples mentioned above throughout the paper to give better intuition for the model.

---

### Official Review · Reviewer_2Yz9 · 2023-07-11

**Soundness:** 3 good
**Presentation:** 3 good
**Contribution:** 3 good
**Rating:** 5
**Confidence:** 2

**Summary:**

This paper studies the problem of online resource allocation with long-term fairness constraints. Different from mainstream literature, the authors assume that the fairness context is not known to the player in advance. However, the player does have the option of paying to buy fairness context information from different sources to make more fair decisions. The algorithms consists of two parts, the first part focuses on source selection for buying protected fairness information and the second part focuses on the allocation decision. The regret of the proposed algorithm is analyzed and it is shown that it is $O(\sqrt(T))$. Last, there are some additional discussions and results regarding some other notions of fairness.

**Strengths:**

++ It is interesting that the authors consider the setting where protected attributes (such as demographics) are unknown to the online decision maker, but the decision maker should still be fair. It makes sense that prior works on fair allocation would only consider the case where these are known, but the reviewer is not familiar enough with the literature to know whether that’s true.

++ The assumption that a user’s protected attributes can be estimated from the online context and/or additional data which is purchased is an interesting addition to the setting.

++   It’s nice that it can model multiple notions of fairness as discussed in the last section of the paper.

**Weaknesses:**

-- The fairness in their setting is a penalty rather than a hard constraint. This should be clarified since the claim of long-term hard fairness constraints is mentioned multiple times in the paper.

-- I was expecting to see more discussion about the trade-off between paying for additional information and paying for fairness — i.e., when fairness doesn’t cost much, how does that impact the allocation? Maybe the regret bounds still hold in either case, but it doesn’t tell us qualitatively how the algorithm actually creates fair allocations. Especially since this “trade-off” is explicitly mentioned in the title of the paper…


-- The assumption that users are drawn i.i.d. from a probability distribution enables this second assumption that the conditional expectation of protected attributes given context is known (learned from past data). However, in practice, past data is often unfair, which means I think these estimates may be inaccurate, particularly for underrepresented groups, and that intuitively should impact fairness here. It seems that there is no discussion of distribution shift or, in general an acknowledgment that the past data might exhibit this property. They do mention in the conclusion that the estimate of protected attributes could be learned online.

**Questions:**

see above.

---

> ### Author Rebuttal · Authors · 2023-08-09
>
> Thank you for this review!
>
> – **Fairness Penalty**: We do mention that we consider a fairness penalty and not a hard constraint in the abstract and in the model (e.g., lines 1,5,10 and 119), but we will make it clearer by also mentioning it in the introduction. We feel that the confusion may have come from the mention of constraints. Here, we mean *soft* constraints, but we see how that can be misunderstood, and will correct this.
>
> – **Trade-off and fair allocations**:  The main goal of the paper is to study online algorithms that can handle the decision trade-off between fairness and the quality of various sources with missing information, and then analyze the value of our algorithm compared to optimal offline value (i.e., the regret). The trade-off is done ``automatically’’ in the algorithm through the use of the virtual value $\varphi$ and dual fairness parameter $\lambda$, which can be thought of as a fairness price. Indeed, the regret bound in our algorithm holds regardless of the magnitude of $R$ (which models the cost of fairness); but note that the regret bound depends on $R$ through its Lipschitz constant $L$---so, for instance, if $R$ is divided by $10$, then so is $L$ and so is the term related to fairness in the regret bound.
>
>  Analyzing how the problem parameters impact the optimal solution for source selection and allocation is definitely another interesting problem, but more of an offline one since the optimal solution comes from the static part of the problem. This was not the focus of our work and a detailed analysis is out of the scope of this paper. Nonetheless, there is a (barebone) example in Figure 1 section F.2 in the Appendix showing an optimal solution. Additionally, we made numerical simulations in a simple case showing how the optimal allocation and other quantities vary as we vary some parameters. We refer the reviewer to the global response and the PDF attached to it for details about this simulation.
>
> – **The last comment of the reviewer contains two points, which we address separately**:\
> **i.i.d inputs assumption**: Concerning the stationarity of the distribution, it is true that there could be a distribution shift but this shift can reasonably be neglected in some applications. This is the case for instance of advertising: ad campaigns are conducted over a short time period (but still for a large number of ads $T$), and consumer habits are unlikely to widely vary. We will add this comment near the i.i.d assumption.
>
> **Estimates of protected attributes**: Concerning the accuracy of the estimates, there are two quantities of interest, $a_t$ and $u_t$. For $u_t$, because this utility is specific to the decision maker (for instance in the advertising example, how much money categories of individuals are likely to spend), it is indeed more likely for this quantity to be biased. For this reason we showed in Section $4.2$ how under some structural assumptions these estimates can be learned in an online fashion. Concerning the protected attributes $a_t$, as it is usually some kind of demographic (age, wealth, gender) not depending on the decision-maker, it is more likely to be correctly captured by historical data and thus can be accurately estimated using this information (see discussion line 379). Still, we showed in Appendix I (indeed mentioned in the conclusion) how under some simplifying assumptions the estimates of $a_t$ can also be learned online. Other structural assumptions such as a logistic or linear structure on the $a_t$ are possible and should lead to a similar analysis as done in Section $4.2$.
>
> However we do agree that there could be decision makers using wrong or biased estimates (that they think are accurate), hence we will highlight more those two assumptions (iid and accurate estimates) in the model section. A very interesting question that the reviewer leads to would be to quantify how a decision maker using a priori biased estimates could impact the fairness of the allocation and the performance of the algorithm, which might be a natural next step. Here we mostly focused in this work on the (online) decision making aspect of using those estimates.
>
>
> We hope to have clearly addressed all your concerns and are happy to discuss further during the discussion period.

---

> > ### Comment · Reviewer_2Yz9 · 2023-08-14
> > **response**
> >
> > Thank you for the detailed response and additional experiments. I have increased my score.

---

### Author Rebuttal · Authors · 2023-08-09

In response to reviewer 2Yz9 we have conducted an additional numerical experiment for a simple example on the impact of varying fairness and information cost over the allocations and other meaningful quantities. The results are presented in the figure of the one-page PDF attached to this global response.

**Model**: For this numerical example, $a_t$ can take values $-1$ or $1$. We consider a fairness penalty $R(x)=r x^2$ with $r\geq 0$ and the following distribution for $(u_t,a_t)$: $\Pr(a_t=1)=\Pr(a_t=-1)=1/2$ and $u_t \sim \mathcal{N}(a_t,1)$. We assume $K=2$ with for source $k=1$, $p_1=p \in [0,1]$ and $c_{t,1}=(a_t,u_t)$, and for source $k=2$, $p_2=0$ and $c_{t,2}=u_t$. Basically we observe the utility $u_t$ in all cases, and before observing $u_t$ we can pay $p$ to additionally observe $a_t$. Note that when $r=0$ then $R=0$ and there is no fairness penalty, while when $p=0$ there is no uncertainty.

We study the asymptotic regime with $T \rightarrow \infty$,  and compute the optimal solution of the offline optimization problem. Figure $1$ represent the quantities associated to the optimal solution for varying $p$ and $r$: (top-left) $\pi^*$ the percentage of times information is bought (i.e., $k=1$ is chosen), (top-right) the probability of group $1$ being selected $\Pr(a_t=1 \mid x_t=1)$, (bottom-left) the fairness cost relative to utility $R(\mathbb{E}[a_t x_t])/\mathbb{E}[u_t x_t]$, and (bottom-right) the information cost relative to utility $p \pi^*/\mathbb{E}[u_t x_t]$.

**Analysis and interpretation**:
- (top-left) Looking at the optimal action in terms of information source selection (to buy the observation of $a_t$ or not), we see that there are three regimes: $\pi^*=1$, $\pi^*=0$ and a transitive regime $\pi^* \in (0,1)$. When $r=0$ there is no reason to care about $a_t$, thus we simply do not buy information $\pi^*=0$ and this remains true as long as the cost of $p$ is large over $r$. When $p=0$ there are no costs to observe $a_t$ and thus $\pi^*=1$ and this remains true as long as the fairness penalty $r$ is large enough over $p$. In between, we may obtain optimal actions which are strictly between $0$ and $1$: in this case the best action is a strict convex combination between the two actions and there is some trade-off between buying costly information and selecting an unfair allocation.
- (top-right) Regarding the proportion of individuals of group $1$ selected, we see that they are always higher than $0.5$, this is because group $1$ correlates to higher utility compared to group $2$ (which corresponds to $a_t=-1$). Therefore it is decreasing in $r$ and goes towards the perfectly fair allocation $0.5$ which would incur no penalty even for high $r$. It is also increasing in $p$ as higher information costs make it less desirable to observe $a_t$, and thus more difficult to accurately make a fair allocation: the support of the utility $u_t$ conditioned on $a_t=1$ and conditioned on $a_t=-1$ are not disjoint, which makes it impossible only by observing $u_t$ to determine whether $a_t=1$ or $-1$.
- (bottom-left) For the relative fairness cost compared to utility, when the information cost $p$ increases the allocation becomes more unfair as it becomes more difficult without access to $a_t$ to choose a fair allocation with good utility, moreover the average utility $u_t$ of the selected groups tend to decrease as the individuals which are more likely to be $a_t=-1$ based on $u_t$ have low utility. Therefore the ratio fairness penalty over expected utility increases. When $r$ increases there are three conflicting effects: high $r$ makes it more appealing to achieve a fair allocation at the cost of expected utility $u_t$ which makes the latter decrease, the expected group allocation becomes more fair as seen previously, and the scaling of $R$ increases. When $r$ becomes too high this leads towards a perfectly fair group allocation and thus a fairness penalty of $0$, and when $r=0$ the fairness penalty is also $0$. Overall for each fixed $p$ there is some maximum in terms of fairness penalty relative to utility.
- (bottom-right) Finally for the information cost relative to utility, it is increasing in $r$: due to large fairness penalties it is better to buy information thus $\pi^*$ increases while $p$ remains fixed, hence the product $p\pi^*$ increases. When $p$ increases, we buy information less often ($\pi^*$ decreases), and because $p\pi^*=0$ for $p=0$ or $p$ large, there is some maximum for each $r$ fixed.

Overall, we see that higher fairness penalties tend to make it more likely to buy information and select a fair allocation even at the cost of utility, while high information cost makes it unattractive to buy this information and can lead to unfairness due to the difficulty of identifying the protected attribute without this information. Given that the focus of our paper is on the online aspects of the problem and not the offline aspects discussed in this response (to address a comment of Reviewer 2Yz9), we plan to put this numerical example (figure and associated comments) in the supplementary material only.

---

### Decision · Program_Chairs · 2023-09-21

**Decision:**

Accept (poster)

**Comment:**

This paper looks at online allocation problems where long-term group unfairness is penalized *and* where the central platform does not directly and immediately observe group membership/sensitive attributes, but instead can pay to receive additional information about group membership.  Reviewers and this AC appreciate that motivation, as (i) many real-world platform allocation/matching problems either are not legally allowed to or otherwise do not collection group membership but (ii) can either pay human labelers to, at cost, get a better estimate of membership or can go through a third-party watchdog service to get some estimate.  The paper addresses a rather general form of group fairness that in this setting captures a few standard group fairness defs like demographic parity.  Reviewers appreciated the model but found issue with some restrictive assumptions (users drawn iid from a distribution that does not change, the mechanism combining public info and paid-for private info, and some lack of bounds.  Still, the reviewers and this AC appreciated the rebuttal and discussion and, when the authors make their rebuttal edits as promised, find the paper a worthy addition to the online allocation/matching + fairness + learning literature.